

# Dirac fermions and topological phases in magnetic topological insulator films

**Kai-Zhi Bai$^{1\star}$, Bo Fu$^{2\dagger}$ and Shun-Qing Shen$^{1\ddagger}$**

**1** Department of Physics, The University of Hong Kong, Pokfulam Road, Hong Kong, China
**2** School of Sciences, Great Bay University, Dongguan, China

$\star$ kzbai@connect.hku.hk , $\dagger$ fubo@gbu.edu.cn , $\ddagger$ sshen@hku.hk

## Abstract

We develop a Dirac fermion theory for topological phases in magnetic topological insulator films. The theory is based on exact solutions of the energies and the wave functions for an effective model of the three-dimensional topological insulator (TI) film. It is found that the TI film consists of a pair of massless or massive Dirac fermions for the surface states, and a series of massive Dirac fermions for the bulk states. The massive Dirac fermion always carries zero or integer quantum Hall conductance when the valence band is fully occupied while the massless Dirac fermion carries a one-half quantum Hall conductance when the chemical potential is located around the Dirac point for a finite range. The magnetic exchange interaction in the magnetic layers in the film can be used to manipulate either the masses or chirality of the Dirac fermions and gives rise to distinct topological phases, which cover the known topological insulating phases, such as the quantum anomalous Hall effect, quantum spin Hall effect and axion effect, and also the novel topological metallic phases, such as the half-quantized Hall effect, half quantum mirror Hall effect, and metallic quantum anomalous Hall effect.

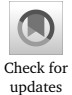

# 1 Introduction

Topological phases, bridging the abstract topological classification [1–4] to the practical electronic phases of matter, have gained an increasing interest and redefined the way people understand and estimate physics in condensed matter systems [5–7]. In contrast to phases described by the Landau-Ginzburg theory and spontaneous symmetry breaking scheme [8, 9], phases termed after topological share no local order parameter, but topological invariants [4, 10–12] defined globally only. These invariants, such as Chern numbers and the $\mathbb{Z}_2$ invariant, exhibit robustness against continuous deformations that do not alter certain preconditions imposed over specified topological classes, like the global gap for an insulator [13–16], and symmetry constraints over the total system [3] or the Fermi surface in a metal [4].

Within the vast topological phase landscape, the three-dimensional topological insulator (3D TI) [17–23] stands out as a unique state of matter, protected by the time-reversal symmetry and characterized by a strong $\mathbb{Z}_2$ index. As a result of the celebrated bulk-boundary correspondence [24–27], the surface of a 3D TI hosts a single gapless Dirac fermion, whose low-energy dispersion is necessarily governed by the massless Dirac equation in 2D, exhibiting spin-momentum locking [28]. Nevertheless, the ever existence of such a gapless Dirac fermion has to be restrained by the no-go Nielsen-Ninomiya theorem [29, 30], and it turns out that the high-energy states of this fermionic band gain a bulk-like mass [31, 32] to reconcile the contradiction. The sign of this restored mass is defined as the chirality [33] for a regulated 2D gapless Dirac fermion, and it is responsible for the half-quantization of its Hall conductance. The emergence of the high-energy mass term due to the lattice regularization essentially breaks the parity symmetry explicitly [34] and evades locality [35] simultaneously.

The gapless behavior of the surface Dirac fermion can be altered through the finite-size effect. When the topological insulator is exfoliated into a film, two gapless Dirac fermions emerge at the top and bottom surfaces. However, as the thickness of the film is further reduced to the ultra-thin limit, by quantum confinement [36–38] the surface states of the two Dirac bands become gapped. The thickness-dependent mass gap exhibits an exponentially decaying and oscillating pattern [39], revealing multiple topological phase transitions. This phenomenon provides a pathway to realize the 2D quantum spin Hall effect [12, 40–42] with an ultra-thin TI film.

The occurrence of spontaneous magnetization can alter the topological property of the TI film. Typically, a pair of gapless Dirac fermions emerge at two surfaces of a TI film, each carrying half-quantized Hall conductance with opposite signs under mirror symmetry, leading to the half quantum mirror Hall effect [33]. The effect shares a similar quantized non-local transport signature with the quantum spin Hall effect [12, 43–45], while being intrinsically a metallic phase. Further gapping out the surface states by an out-of-plane magnetism [46] gives rise to various topologically distinct phases. Within the scheme of magnetic topological insulators, two such phases have been discovered as the Chern insulator [47–49], aka quantum anomalous Hall effect (QAHE) that is characterized by Chern invariant and quantized Hall plateau, and the axion insulator [50, 51], marked by zero Hall plateau and non-vanishing longitudinal conductance. A semi-magnetic topological insulator, on the other hand, bears with the half-quantized quantum anomalous Hall effect (half QAHE) [31, 52, 53] with a half-quantized Hall conductance and unusual bulk-boundary correspondence, signed by the absence of edge state but the appearance of the power-law decaying current from boundary to bulk. In addition, if the magnetization is pushed away from the surfaces and towards the middle of the film with sufficient strength, the metallic quantized anomalous Hall effect (metallic QAHE) [32] can occur, which also exhibits integer Hall conductance but lacks chiral edge states.

Remarkably, the physics underlying the topological phases in the (magnetic) topological insulator films can be all attributed to the topological properties of the emergent two-dimensional Dirac fermions in the system. While certain phases, like QAHE and half QAHE, can be well explained by focusing on the interplay between surface Dirac fermions and magnetism, there exist other phases that essentially involve higher bulk bands, notably the metallic QAHE. These higher bulk bands are identified as a series of massive Dirac fermions, revealing that both gapless and gapped Dirac fermions in the topological insulator film interact with spontaneous magnetism to generate various topological phases. The topological index, or the quantized Hall conductance in each phase, is always given by some gapped or gapless Dirac fermion(s), described by a modified Dirac equation.

In this paper, we will provide a unified framework to discuss and review how emergent Dirac fermions exist and generate various topological phases in magnetic topological insulator films, thus naturally partitioning the paper into two main parts. The first part of the paper will focus on establishing the existence of Dirac fermions in magnetic topological insulator films. This discussion will heavily rely on a newly defined basis derived from an exact solution in 1D. We will thoroughly investigate the Hall conductivity carried by different types of Dirac fermions within this framework, setting the stage for the subsequent discussion of topological phases. In the second part we will delve into the characterization and analysis of topological phases in magnetic topological insulator films. These phases will be classified into weak- and strong-magnetism regimes, providing a comprehensive understanding of how different magnetic strengths influence the emergence of various topological states. In the remainder of this introduction we will give an overview of the main results of this paper following the line.

The TI film is equivalent to a set of Dirac fermions: a pair of massless Dirac fermions for bands that contain the surface states, and a series of massive Dirac fermions consisting of purely bulk states, classified by their momentum-dependent mass terms $m_n(\boldsymbol{k})$. This scenario holds within both its continuum and lattice model versions, and is made clear and exact through an introduced unitary transformation in the whole $k$-space, based on an exact solution in one dimension perpendicular to the film plane. The finite-size effect is briefly discussed here.

The Hall conductivity carried by a massive or gapless Dirac fermion is discussed generally, with additional symmetry constraints imposed on the Fermi surface for the latter one, for both continuum and lattice models. A direct deduction leads to the result that the Hall conductivities associated with the gapless and gapped Dirac fermions in the TI film are $\pm e^2/2h$ and 0, respectively, leading to a half quantum mirror Hall effect by $1/2 - 1/2$, serving as a metallic partner to the insulating quantum spin Hall effect. A brief proof for the half-quantization of a metallic band structure with considered symmetry constraints over the Fermi surface is also presented. Additionally, a field theoretical deduction for the half quantization, and a discussion on handling the Hall conductivity of a gapless Dirac fermion are provided.

The introduced magnetism, characterized by out-of-plane polarization, manifests as two equivalent matrix Higgs fields that collectively couple the Dirac fermions in a TI film, generating and altering their masses. Treated at the mean-field level, the exchange interaction stands as an out-of-plane Zeeman field in TI film, which transforms via the unitary transformation into two momentum-dependent matrix fields $\mathbf{I}_{S/A}(\boldsymbol{k})$. The two fields directly couple different species of Dirac fermions and alter their masses, serving as mass-generating Higgs fields, whose non-vanishing expectation values arise concurrently with the spontaneous establishment of the ferromagnetic order. Depending on the field strength, generally two regimes as weak and strong magnetism are classified. In addition, the forms of other kinds of spin and orbital fields under unitary transformation are discussed.

In the weak Zeeman field regime, the topological phases are characterized by focusing on $n = 1$ matrix elements affecting the two gapless Dirac fermions near the surface. This framework clarifies the underlying physics behind the Chern insulator, axion insulator, and

half QAHE, with symmetric, antisymmetric, or unilateral distribution of Zeeman fields at the surface of the TI film, respectively. The resulting Hall conductance exhibits quantized nature: $1 + 0$, $0 + 0$, and $1/2 + 0$ in units of $e^2/h$. Additionally, the mirror layer Chern number in the Chern insulator with symmetrically distributed magnetism is examined, revealing $(1/4)$–$(1/2)$–$(1/4)$ partition for the non-trivial band and $(c/4)$–$(-c/2)$–$(c/4)$ with $c \approx 1$ for the trivial band.

In the strong Zeeman field regime, the discussion is based on the effective mass picture, involving the gapped series of Dirac fermions through matrix Higgs fields couplings. Another metallic topological phase, the metallic QAHE, is identified where the magnetism is centralized in the middle of the TI film. Despite remaining gapless and lacking chiral edge states, its Hall conductance is quantized into an integer over $e^2/h$. Additionally, higher Chern insulators resulting from sub-band inversion at high-symmetry points are presented under a uniform Zeeman field. Furthermore, the paper discusses topological phases characterized by cooperation between magnetism in the middle and at the surface, based on the framework of gapping out surface states in the metallic QAHE.

The plan of the remainder of this paper is as follows. Beginning with the exact solution of the model Hamiltonian for a topological insulator film in Section 2, we demonstrate that a TI film comprises a pair of gapless Dirac fermions, which contain low-energy surface states, and a series of gapped massive bulk Dirac fermions. Section 3 offers a comprehensive discussion on the Hall conductivity, a critical indicator revealing the presence of topological phases, carried by different species of Dirac fermions. Moving on to the inclusion of magnetism in Section 4, we unveil the role of magnetism as matrix Higgs fields, responsible for generating masses of the Dirac fermions in a TI film. This section also briefly explores other spin and orbital fields possible within the framework. In Section 5, based on the weak magnetism approximation, we identify topological phases processable under the lowest four-band model framework, which stresses surface states with magnetism: half quantum mirror Hall effect, quantum anomalous Hall effect, half-quantized anomalous Hall effect, and axion insulator. We introduce the mirror layer Chern number and illustrate the Hall conductivity distribution in symmetrically magnetized TI film. The Chern and axion insulator phases in interlayer anti-ferromagnetic material $MnBi_2Te_4$ are also discussed under the same framework. In Section 6, we delve into topological phases within relatively strong magnetism regimes, such as high Chern number insulators and the metallic quantized anomalous Hall effect, where bulk Dirac fermions come into play. The paper concludes in Section 7 with a summary and a discussion of future prospects.

## 2 Massless and massive Dirac fermions in a topological insulator film

In this section, by solving the minimal continuum and lattice models of the topological insulator, we show that from the physical aspect, a topological insulator film is composed of a pair of gapless Dirac fermions, whose low-energy parts near Dirac point are composed of massless surface states inside the bulk gap while the high-energy parts away from the Dirac point evolve into bulk states gradually, together with a series of gapped massive Dirac fermions consisting of purely bulk states. Quantitatively, we write

$$H_c(\boldsymbol{k}) = \bigoplus_n \left[ \lambda_\parallel \boldsymbol{k} \cdot \boldsymbol{\sigma} + m_n(\boldsymbol{k}) \tau_z \sigma_z \right], \tag{1a}$$

$$H_l(\boldsymbol{k}) = \bigoplus_{n=1}^{L_z} \left[ \lambda_\parallel \sin(k_x a) \sigma_x + \lambda_\parallel \sin(k_y a) \sigma_y + m_n(\boldsymbol{k}) \tau_z \sigma_z \right], \tag{1b}$$



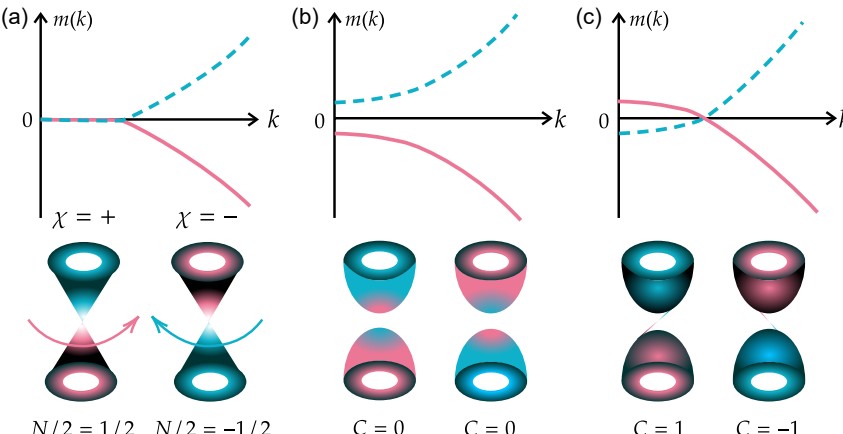

Figure 1: Schematic momentum dependent mass configurations (upper panel) and corresponding band structure of Dirac fermions (lower panel). The quantization of Hall conductivity is denoted by $N$ for half quantization in a metallic band and $C$ for quantization at the bottom of a gapped band. The colors assigned to the Dirac cones represent the sign of Berry curvature with red for positive and blue for negative. (a) On the left panel, two gapless Dirac fermions are shown, whose masses are zero at low-energy near the Dirac point (assumed to be $k = 0$), while non-vanishing at high-energy with opposite signs, which we define as the **chirality** $\chi$ of a 2D gapless Dirac fermion. Such chirality unambiguously determines the sign of the loop integral of Berry connection around the Fermi surface, consequently determining the sign of half-quantized Hall conductivity. (b) In the middle, two trivially gapped massive Dirac fermions are present, with masses being either positive or negative for all $k$, leading to a sign change of Berry curvature and a totally vanishing Hall conductivity labeled by zero Chern number. (c) On the right panel, two non-trivial gapped Dirac cones are displayed, and the corresponding masses exhibit kink configurations with sign change between low and high-energy areas. Such non-trivial mass configuration indicates overall Berry curvature sign convergence, and leads to a non-vanishing Hall conductivity labelled by an integer Chern number. The non-triviality is also addressed by formally drawing states connecting conduction and valence bands, well-known as chiral edge states for a Chern band under open boundary conditions [13, 14].

for the continuum and lattice model, respectively. Here we adopt a homogeneous in-film-plane parameter set with $a$ and $\lambda_{\parallel}$ as the in plane lattice constant and Fermi velocity, and $\boldsymbol{k} = (k_x, k_y)$ is the in-film-plane wavevector. Notice that an infinitely direct summed Dirac fermions exist in the continuum model, while there are only $2L_z$ species with $L_z$ the layer number along opened $z$-direction of the film in the lattice model. For the mainly concerned individual Dirac cone with a single Dirac point at $k = 0$, its topological property is revealed based on a general discussion over the nature of its Hall conductivity quantization, as revealed in the schematic diagram Fig. 1. Especially, in the strong TI film with a single Dirac cone at $\Gamma$, aka $k = 0$ point, the gapless pair of Dirac fermions carry $\pm e^2/2h$, as half-quantized Hall conductivity, while the gapped series are all trivial.

## 2.1 The continuum model

In this subsection, the exact solution of the confined 3D modified Dirac equation, which is the continuum model describing the topological insulator film, is presented. A detailed study can be found in Appendix A.

The continuum model Hamiltonian for the 3D TI reads [27, 54]

$$
\begin{aligned}
H_{\text{TI}}(\boldsymbol{k}, k_z) &= \lambda_\parallel(\boldsymbol{k} \cdot \boldsymbol{\sigma})\tau_x + \lambda_\perp k_z \sigma_z \tau_x + (m_0(\boldsymbol{k}) - t_\perp k_z^2)\sigma_0 \tau_z \\
&= H_{1d}(\boldsymbol{k}, k_z) + H_\parallel(\boldsymbol{k}),
\end{aligned}
\tag{2}
$$

where $H_\parallel(\boldsymbol{k}) = \lambda_\parallel(\boldsymbol{k} \cdot \boldsymbol{\sigma})\tau_x$, $m_0(\boldsymbol{k}) = m_0 - t_\parallel k^2$. This Hamiltonian is isotropic only in $x$-$y$ plane. Substituting $k_z \longmapsto -i\partial_z$ leads to the real-$z$-space description for the 1-D part as $H_{1d}(\boldsymbol{k}, z) = \oplus_{s=\pm} h(s)$, where

$$
h(s) = -is\lambda_\perp \partial_z \tau_x + (m_0(\boldsymbol{k}) + t_\perp \partial_z^2)\tau_z.
\tag{3}
$$

Solving the eigenproblem $h(s)\phi = E\phi$ leads to specifically symmetrized chiral-partner basis [36–38]

$$
\varphi^n(s) = C\begin{pmatrix} -is\lambda_\perp f_+^n \\ t_\perp \eta^n f_-^n \end{pmatrix}, \quad E = m_n,
\tag{4a}
$$

$$
\chi^n(s) = C\begin{pmatrix} t_\perp \eta^n f_-^n \\ is\lambda_\perp f_+^n \end{pmatrix}, \quad E = -m_n,
\tag{4b}
$$

where the dependence on $(\boldsymbol{k}, z)$ is inherited inside even/odd parity functions $f_\pm^n(\boldsymbol{k}, z)$ and real factor $\eta^n(\boldsymbol{k})$, whose definition can be found in Appendix A. The $k$-dependent eigenvalue of $h(s)$ is represented by $\pm m_n(\boldsymbol{k})$, $n = 1, 2, \cdots$, as a mass term, which can be solved in a closed manner through equations

$$
m_n = m_0(\boldsymbol{k}) - t_\perp \frac{\xi_1^2 g(\xi_1) - \xi_2^2 g(\xi_2)}{g(\xi_1) - g(\xi_2)},
\tag{5a}
$$

$$
\xi_\alpha = \sqrt{-\frac{F}{D} + (-1)^{\alpha-1}\frac{\sqrt{R}}{D}}, \quad \alpha = 1, 2,
\tag{5b}
$$

where

$$
\begin{cases}
g(\xi) = \tan(\xi L/2)/\xi, \\
D = 2t_\perp^2, \\
F = -2m_0(\boldsymbol{k})t_\perp + \lambda_\perp^2, \\
R = F^2 - 2D(m_0^2(\boldsymbol{k}) - m_n^2).
\end{cases}
\tag{6}
$$

Projecting TI film Hamiltonian on eigenstates of $H_{1d}$ equals to performing an infinite-dimensional local unitary transformation in $k$-space, which gives a Hamiltonian equivalent to the TI film one as (see Appendix A.)

$$
H(\boldsymbol{k}) = \bigoplus_n \lambda_\parallel \tau_0(\boldsymbol{k} \cdot \boldsymbol{\sigma}) + m_n(\boldsymbol{k})\tau_z \sigma_z,
\tag{7}
$$

as Eq. (1a), where the projection basis is organized as

$$
\Phi_1^n = \begin{pmatrix} \varphi^n(+) \\ 0 \end{pmatrix}, \qquad \Phi_2^n = \begin{pmatrix} 0 \\ \chi^n(-) \end{pmatrix},
$$

$$
\Phi_3^n = \begin{pmatrix} \chi^n(+) \\ 0 \end{pmatrix}, \qquad \Phi_4^n = \begin{pmatrix} 0 \\ \varphi^n(-) \end{pmatrix}.
\tag{8}
$$

We have to emphasize here that although spin is still preserved as $\sigma$ in the transformed Hamiltonian, the degrees of freedom $\tau$ newly appeared here share a different meaning compared with the original TI film Hamiltonian. Notice that $\Phi_{1,4}$ ($\Phi_{2,3}$) are $z$-parity even (odd) states,

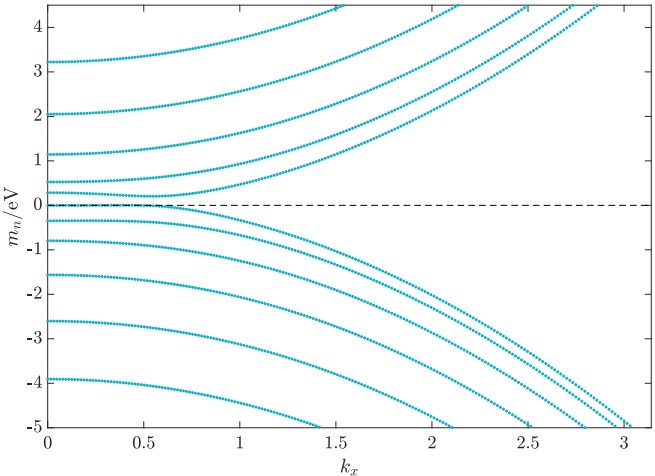

Figure 2: The momentum-dependent mass of Dirac fermions in a TI film as a continuum model. The lowest several momentum-dependent $m_n(\boldsymbol{k})$ along $k_x$ solved from closed equations Eq. (5) are presented, while the homogeneous in-plane nature of the model ensures that the asymptotic behavior of $m_n(\boldsymbol{k})$ is the same as $k \to \infty$. Here the film thickness $L = (L_z + 1)c$ with $L_z = 10$ is chosen here as a TI film with 10 layers. The index $n$ is assigned such that $|m_n|$ increases with $n$. Especially notice the sign-jump behavior that $\text{sgn}(m_n(\infty)) = (-1)^n$. From here on, the model parameters on lattice for numerical calculations and verifications are set as $\lambda_\parallel = 0.41$ eV eV, $\lambda_\perp = 0.44$ eV, $t_\parallel = 0.566$ eV, $t_\perp = 0.4$ eV, $m_0 = 0.28$ eV, $a = b = 1$ nm, $c = 0.5$ nm if with no specific indication [54]. Generically, these parameters can be determined through the first-principle calculations, and the specific choice here is for the sake of illustration. This parameter choice makes the bulk 3D TI a strong one with a single Dirac point at $\Gamma$. And for the continuum model discussed here, the substitution $\lambda_\parallel \to \lambda_\parallel a$ $\lambda_\perp \to \lambda_\perp c$, $t_\parallel \to t_\parallel a^2$, $t_\perp \to t_\perp c^2$ should be recognized.

while $\Phi_{1,2}$ ($\Phi_{3,4}$) are $z$-mirror even (odd) states, which means that under the projection, the unitary matrices related to two operators are transformed into (see Appendix A)

$$P_z = \tau_z \sigma_z, \tag{9a}$$

$$M_z = \tau_z. \tag{9b}$$

Meanwhile, the local unitary matrix in $k$-space that transforms the continuum model Hamiltonian under the original representation is formally written as

$$U^c(\boldsymbol{k}, z) = (\{\{\Phi(\boldsymbol{k}, z)\}_i\}^n), \tag{10}$$

where the double brackets mean that we arrange $i = 1, 2, 3, 4$ index inside each $n = 1, 2, \cdots$, we see that $U^c$ is topologically trivial in $(k_x, k_y)$ space, as it consists of certain arrangement of eigenstates $\Phi_i^n$, which are solved from the separated 1-D Hamiltonian and has a well-defined global representation within the same gauge choice in $(k_x, k_y)$ plane, and is therefore topologically trivial.

Our solution reveals that the 3D topological insulator film is composed of effectively 2D multi-Dirac fermions, differing by their mass terms represented in Fig. 2 only. Notice that for the continuum model, there are in fact an infinite number of $m_n$s as a basic property of bound states in a quantum well, and we just present several lowest branches of the solutions. Also notice that from the solved $m_n$, the mass terms show sign jumping behavior at high-energy (large $k$). Comparing the mass configurations in continuum model with the general

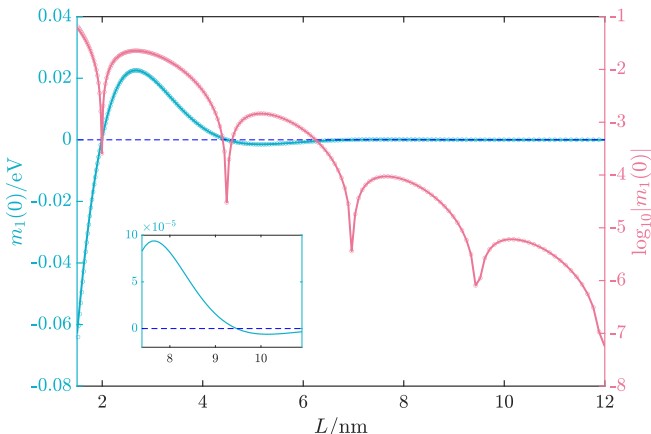

Figure 3: Finite-size effect of an ultra-thin TI film in the continuum model, revealed by the exponentially decaying oscillating mass gap $2m_1(0)$ of surface Dirac cones. Both $m_1(0)$ and its logarithmic absolute value varying with film thickness $L$ are shown. The solid line represents results from Eq. (12), while the circles are obtained from solving the self-consistent equations Eq. (5) directly. The inset shows an amplified area of the $m_1(0) - L$ diagram.

classification in Fig. 1 reveals that while all $n \geq 2$ masses serve as trivial massive Dirac band in the bulk, the lowest states with $n = 1$ are necessarily not, which in the presented case serve as two possible gapless Dirac cones whose low-energy parts are localized $z$-mirror-symmetrically at top and bottom surfaces. Especially, the analytic expression for $m_1(\boldsymbol{k})$, when the film is thick enough, can be written as [33] (also see Appendix A, and here $t_\perp > 0$ is assumed without losing generality)

$$m_1(\boldsymbol{k}) = \Theta(-m_0(\boldsymbol{k}))m_0(\boldsymbol{k}). \tag{11}$$

Notice that the Heaviside Theta function appearing here only reveals physics that, in the low-energy zone near the Dirac point, the surface Dirac cone is massless, preserving both time-reversal and parity symmetry, while for the high-energy part away from the Dirac point, the non-vanishing mass term reveals that the surface Dirac cone has emerged into the bulk state, which breaks both time-reversal and parity symmetry explicitly. The appearance of such non-vanishing high-energy mass term is analogous to the introduced regulator [55–57] in quantum field theory. In this sense, one should not worry about the nonanalytic behavior of the Theta function near $m_0(\boldsymbol{k}) = 0$, as it can always be replaced by its mollifier [31, 58].

For the completeness of discussion here, we note that an ultra-thin TI film bears an exponentially decaying oscillating small gap $m(0)$ with varying film thickness [36–38], which reads upon the lowest order as (for derivation, also see Appendix A)

$$m_1(0) \approx -\frac{4m_0}{\sqrt{4\gamma - 1}} \sin\left(u\sqrt{4\gamma - 1}L\right)e^{-uL}, \tag{12}$$

with $\gamma = m_0 t_\perp/\lambda_\perp^2, u = \lambda_\perp/2t_\perp$. The numerical result is shown in Fig. 3, with excellent agreement between the lowest order approximated gap and that from solving the set of non-linear equations, especially for relatively large $L$. The exponentially decaying tendency is best revealed by the logarithmic absolute value of mass gap at $k = 0$, as its center decreases linearly with thickness, while the oscillating nature is revealed by the dips, which will extend to negative infinity at strict gap closing point, and the mass gap will reverse its sign before and after the dip, as shown directly by the $m_1(0) - L$ diagram and the inner amplified picture. Since $m_1(\infty) = m_0(\infty) < 0$ is certain, we see that the oscillating behavior of $m_1(0)$ with thickness $L$ can drive $m_1(\boldsymbol{k})$ to share configuration that jumps between the one shown in Fig. 1(b) and (c),

i.e., between a trivial band and a band with unit Chern number. Then for an ultra-thin film which owns two copies $\pm m_1(\boldsymbol{k})$ reflected by $\tau_z$ in Eq. (7), the $\mathbb{Z}_2$ topological index shows jumping behavior between $\mathbb{Z}_2 = 0$ and $\mathbb{Z}_2 = 1$, i.e., between a band insulator and a quantum spin Hall insulator [40–42,59,60]. We will not discuss further about this phenomenon except for giving an explicit $\mathbb{Z}_2(L_z)$ oscillating diagram below in the lattice model subsection shown in Fig. 5. We emphasize here that the exponentially decaying gap will not affect physically observable topological phase, either for an insulating or metallic one, for a TI film with enough thickness.

The solution of the continuum model enlightens us to commence with the lattice model of TI film below.

## 2.2 The lattice model

In this subsection, we ask and deal with the same question as above, but in the more realistic lattice model. Details are presented in Appendix B.

The Hamiltonian of a 3D TI with nearest-neighbour hopping on cubic lattice is [17,54]

$$\mathcal{H}_{TI} = \sum_l \Psi_l^\dagger \mathcal{M}_0 \Psi_l + \sum_{l,\mu} \left( \Psi_l^\dagger \mathcal{T}_\mu \Psi_{l+\mu} + \text{h.c.} \right), \tag{13}$$

where energy and hopping matrices are $\mathcal{M}_0 = (m_0 - 2\sum_\mu t_\mu)\beta$, $\mathcal{T}_\mu = t_\mu \beta - i\frac{\lambda_\mu}{2}\alpha_\mu$, with $l$ and $\mu$ denoting site locations and three spatial directions, while $\{\beta, \alpha_\mu\}$ denoting Dirac matrices under standard Dirac representation $\beta = \sigma_0 \tau_z$, $\alpha_\mu = \sigma_\mu \tau_x$, where Pauli matrices $\sigma_\mu$ and $\tau_\mu$ represent different degrees of freedom, respectively. For instance, one could choose them to represent spin and pseudo-spin (like orbital) ones. $\Psi_l$ represents vectorized Fermionic operator at site $l$. Notice that when adopting a full Fourier transformation upon all three spatial dimensions, i.e., an infinite bulk system, the Hamiltonian is transformed into the standard modified Dirac's equation [27] on lattice $\mathcal{H}_{TI} = \sum_{\boldsymbol{k}} \Psi_{\boldsymbol{k}}^\dagger H(\boldsymbol{k})\Psi_{\boldsymbol{k}}$ where

$$H(\boldsymbol{k}) = \sum_\mu \lambda_\mu \sin(k_\mu a_\mu)\alpha_\mu + \left[ m_0 - 4t_\mu \sin^2\left(\frac{k_\mu a_\mu}{2}\right) \right]\beta, \tag{14}$$

whose continuum model is just an anisotropic version of Eq. (2). This model avoids the fermion-doubling problem [29,30] by introducing Wilson terms [34] that break chiral symmetry explicitly for $k \neq 0$.

Consider such a film with $L_z$ number of sites along $z$ direction. The Fourier transformation in $x$-$y$ plane gives

$$\mathcal{H}_{\text{Film}} = \sum_{l_z,\boldsymbol{k}} \left( \Psi_{l_z,\boldsymbol{k}}^\dagger \mathcal{M}_0(\boldsymbol{k})\Psi_{l_z,\boldsymbol{k}} + \Psi_{l_z,\boldsymbol{k}}^\dagger \mathcal{T}_z \Psi_{l_z+1,\boldsymbol{k}} + \text{h.c.} \right) + \sum_{l_z,\boldsymbol{k}} \Psi_{l_z,\boldsymbol{k}}^\dagger H_\parallel \Psi_{l_z,\boldsymbol{k}}, \tag{15}$$

with

$$H_\parallel = \lambda_\parallel [\sin(k_x a)\sigma_x \tau_x + \sin(k_y b)\sigma_y \tau_x], \tag{16}$$

and $\mathcal{M}_0(\boldsymbol{k}) = M_0(\boldsymbol{k})\sigma_0 \tau_z = [m_0(\boldsymbol{k}) - 2t_\perp]\sigma_0 \tau_z$, where

$$m_0(\boldsymbol{k}) = m_0 - 4t_\parallel \left( \sin^2 \frac{k_x a}{2} + \sin^2 \frac{k_y b}{2} \right). \tag{17}$$

Note that we have set $t_x = t_y = t_\parallel$, $t_z = t_\perp$, $\lambda_x = \lambda_y = \lambda_\parallel$, $\lambda_z = \lambda_\perp$, $a = b$.

The solution of lattice model [32] shares much similarity with the continuum one. The details can be found in Appendix B as a repeat. Separating the Hamiltonian at $\boldsymbol{k}$ as

$$\mathcal{H}_{\text{Film}}(\boldsymbol{k}) = \mathcal{H}_{1d}(\boldsymbol{k}) + \mathcal{H}_S(\boldsymbol{k}), \tag{18}$$

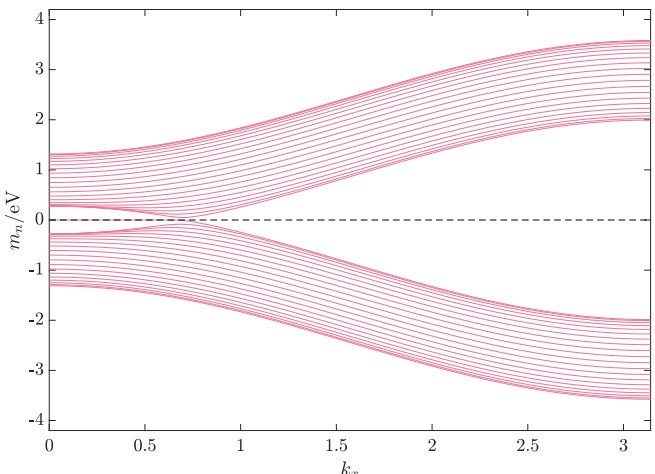

Figure 4: The momentum-dependent mass of Dirac fermions in a TI film on the lattice. Namely, $m_n(\boldsymbol{k})$, $n = 1, 2, \cdots, L_z$ along $k_x$ are solved from closed equations Eq. (20) of lattice model with $L_z = 40$. Again, index $n$ is assigned in the way that $|m_n(\pi, \pi)|$ increases with $n$. Especially notice the sign-jump behavior that $\mathrm{sgn}(m_n(\pi, \pi)) = (-1)^n$.

where

$$\mathcal{H}_{1d}(\boldsymbol{k}) = \sum_{l_z}\left(\Psi^{\dagger}_{l_z,\boldsymbol{k}}\mathcal{M}_0(\boldsymbol{k})\Psi_{l_z,\boldsymbol{k}} + \Psi^{\dagger}_{l_z,\boldsymbol{k}}\mathcal{T}_z\Psi_{l_z+1,\boldsymbol{k}} + \mathrm{h.c.}\right), \tag{19a}$$

$$\mathcal{H}_S(\boldsymbol{k}) = \sum_{l_z}\Psi^{\dagger}_{l_z,\boldsymbol{k}}H_{\parallel}\Psi_{l_z,\boldsymbol{k}}. \tag{19b}$$

The eigenvalues of $\mathcal{H}_{1d}$ can be obtained with a set of simultaneous equations below,

$$m_n = M + 2t_{\perp}\frac{\cos\xi_1 g(\xi_1) - \cos\xi_2 g(\xi_2)}{g(\xi_1) - g(\xi_2)}, \tag{20a}$$

$$\cos\xi_{\alpha} = \frac{-Mt_{\perp} + (-1)^{\alpha-1}\sqrt{M^2 t_{\perp}^2 - (t_{\perp}^2 - \lambda_{\perp}^2/4)(M^2 + \lambda_{\perp}^2 - m_n^2)}}{2(t_{\perp}^2 - \lambda_{\perp}^2/4)}, \tag{20b}$$

where

$$\begin{cases} M = M_0(\boldsymbol{k}), \\ g(\xi) = \dfrac{\tan(\xi(L_z+1))/2}{\sin\xi}, \end{cases} \tag{21}$$

and the sign of $\xi$ is fixed by

$$\sin\xi_{\alpha} = \sqrt{1 - \cos\xi_{\alpha}^2}, \quad \alpha = 1, 2. \tag{22}$$

Now, different from the continuum model, the set of equations give $L_z$ solutions $m_n(\boldsymbol{k})$, $n = 1, 2, \ldots, L_z$ including one surface state and $L_z - 1$ purely trivial bulk states, if within suitable choice of parameters. This is essentially because now the Dirac equation is put on lattice, and the number of solutions is constrained by finite lattice constants. And the other set of $L_z$ masses are just the chiral partners with eigenvalues $-m_n(\boldsymbol{k})$.

The projection basis shares the same form as with the continuum model eigenstates, with only re-defined factor $\eta$ (for details, refer to Appendix B or [32]). And the projection of the TI film model offers an equivalent description as

$$H(\boldsymbol{k}) = \bigoplus_{n=1}^{L_z}\left[\lambda_{\parallel}(\sin(k_x a)\sigma_x + \sin(k_y b)\sigma_y) + m_n(\boldsymbol{k})\tau_z\sigma_z\right], \tag{23}$$

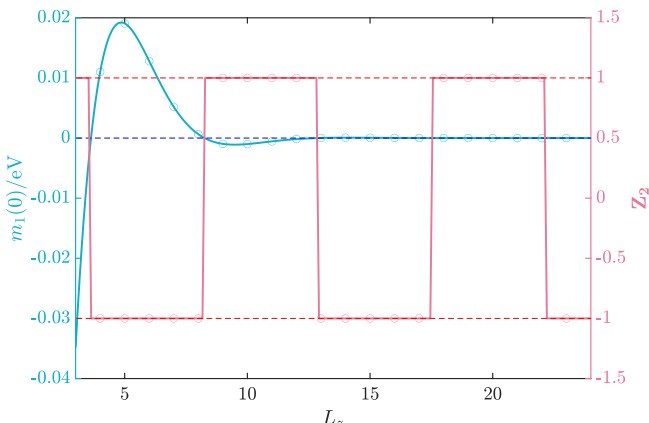

Figure 5: Finite size effect of an ultra-thin strong TI film (Dirac point at $\Gamma$) in the lattice model, revealed by the exponentially decaying oscillating gap $2m_1(0)$ of surface Dirac cones and the oscillating $\mathbb{Z}_2$ index. The solid blue line of $m_1(0)$ represents results from solving the self-consistent equations Eq. (20), while the circles are obtained from diagonalizing the TI film Hamiltonian at $k = 0$ directly. The $\mathbb{Z}_2$ index is calculated from inversion symmetry indicator [61] method, and the solid red line represents index of $n = 1$ block Dirac fermions with solved $m_1(k)$, while circles are indices calculated from TI film Hamiltonian directly.

as Eq. (1b), where $2L_z$ Dirac fermions $H = \oplus_{n,\chi} h_{n,\chi}(\boldsymbol{k})$ emerge as

$$h_{n,\chi}(\boldsymbol{k}) = \lambda_{\parallel}(\sin(k_x a)\sigma_x + \sin(k_y b)\sigma_y) + \chi m_n(\boldsymbol{k})\sigma_z, \tag{24}$$

with $\chi = \pm$ labelling the mirror eigenvalue [33]. An example of $m_n(k)$ with $L_z = 40$ is presented in Fig. 4. Among these Dirac fermions, two of them with $\pm m_1(\boldsymbol{k})$ are gapless Dirac cones with their low-energy states localized at top and bottom surfaces, while emerging into the bulk at their high-energy away from Dirac point, and the remaining fermions are all gapped. Notice that the same arguments about the projection as a trivial local unitary transformation and Heaviside Theta function form of the lowest solution (see below) can be made here, as in the continuum model.

For the strong topological insulator with a single Dirac cone at $\Gamma$ point, as we consider in the article, the lowest mass reads ($t_\perp > 0$ assumed, and the film is thick enough)

$$m_1(\boldsymbol{k}) = \Theta[-m_0(\boldsymbol{k})] m_0(\boldsymbol{k}), \tag{25}$$

which shares the same form with the continuum model.

### 2.2.1 Oscillating $\mathbb{Z}_2$ invariant

As discussed in the continuum model case, in the ultra-thin film limit, the strong TI thin film with a single Dirac cone at $\Gamma$ ($k = 0$) point will show oscillating behavior between a quantum spin Hall insulator and an ordinary insulator. The topological index of this kind is carried out explicitly in Fig. 5, with $\mathbb{Z}_2 = (-1)^\nu$ with $\nu = 0, 1$, and the latter corresponds to a nontrivial 2D quantum spin Hall insulator. The mass oscillation and the index oscillation match perfectly, as $\mathbb{Z}_2 = -1$ ($\nu = 1$) zones correspond to $m_1(0) > 0$, so do their sign transitions (remind that $m_1(\pi, \pi) < 0$ and $m_1(0) > 0$ leads to a nontrivial mass configuration, as to be discussed below). Notice that when attributed to the lowest $n = 1$ block in Eq. (23), there is no constraint to force $L_z$ to be integer from Eq. (20), and in this sense we continue the $n = 1$ block from integer $L_z$ to a positively real one. This is why we can do the calculation above.

Again we emphasize that we will consider thick-enough strong TI film for topological phases hereafter, and the exponentially decaying finite size effect is physically negligible.

# 3 The quantum Hall conductivity of Dirac fermions

As stated, in both the continuum model and lattice model, the strong topological insulator film is composed of two gapless Dirac fermions and countable gapped Dirac fermions. We have also claimed that all the massive fermions inside are trivial, while saying nothing about the massless two. Here in this subsection, we shall complete the basic picture of them. Discussion here is restricted to effectively two-dimensional systems and the zero-temperature limit.

## 3.1 In the continuum model

Our starting point is the continuum model of a two-band Dirac fermion appearing above

$$h_{\text{DF}}^C = \lambda \boldsymbol{k} \cdot \boldsymbol{\sigma} + m(k)\sigma_z, \tag{26}$$

with $\boldsymbol{k} = (k_x, k_y)$ and $\boldsymbol{\sigma} = (\sigma_x, \sigma_y)$. Notice that the mass depends on $k = |\boldsymbol{k}|$ and possesses a topologically trivial infinity behavior. Its Hall conductivity can be carried out by a deformed Kubo formula [27, 62], when the chemical potential $\mu$ lies at the valence band,

$$\sigma_H = -\frac{e^2}{h}\frac{1}{4\pi}\int d^2k\,\Theta(\mu + d)\frac{(\partial_{k_x}\boldsymbol{d} \times \partial_{k_y}\boldsymbol{d}) \cdot \boldsymbol{d}}{d^3}, \tag{27}$$

where $\boldsymbol{d}(\boldsymbol{k}) = (\lambda k_x, \lambda k_y, m(\boldsymbol{k}))$, $d = |\boldsymbol{d}|$, and to reveal possible topological property, we have used the Heaviside Theta function with $\Theta(x > 0) = 1$ and zero otherwise, as the zero-temperature Fermi-Dirac distribution. The Hall conductivity can then be carried out easily by defining

$$\cos\theta = \frac{m}{(\lambda^2 k^2 + m^2)^{1/2}}, \tag{28}$$

and notice that

$$\frac{\sigma_H}{e^2/h} = \frac{1}{2}\int_{k_F}^{+\infty} dk^2 \frac{\partial \cos\theta}{\partial k^2}, \tag{29}$$

which finally leads to

$$\sigma_H = \frac{e^2}{2h}\left[\text{sgn}(m(+\infty)) - \frac{m(k_F)}{d(k_F)}\right], \tag{30}$$

with $k_F$ the Fermi vector determined by $\mu = d(k_F)$, and $\text{sgn}(x)$ the sign function. From this equation, three topological phases are readily to be classified. While we have assumed a path connected Fermi surface, the discussion here should be easily generalized to the Fermi surface composed of concentric circles.

### 3.1.1 Gapless/metallic case

The first case corresponds to a metallic phase with a finite $k_F$. If $m(k_F) = 0$, which leaves a perfect linearized dispersion near the Fermi surface, we obtain a half-quantized Hall conductance as

$$\sigma_H(\mu|d(k_F) = \lambda k_F) = \frac{e^2}{2h}\text{sgn}(m(+\infty)), \tag{31}$$

where the half-quantization is completely determined by the high-energy mass sign which may be recognized as the chirality assigned to the low-energy massless Dirac fermion near the Fermi surface. In our equivalent model, such a case exists for the $n = 1$ bands

$$h_{1,\chi} = \lambda_{\parallel}(\boldsymbol{k} \cdot \boldsymbol{\sigma}) + \chi\Theta(-m_0(k))m_0(k)\sigma_z, \quad \chi = \pm. \tag{32}$$

Since $m_0(\boldsymbol{k}) = m_0 - t_\perp k^2$, then by assuming $m_0 > 0, t_\perp > 0$, we have

$$\sigma_H^{1,\chi}(k_F < k_c) = -\chi\frac{e^2}{2h}, \tag{33}$$

with $k_c = \sqrt{m_0/t_\perp}$ identified. For each gapless Dirac fermion, the exact half-quantization [4,53] comes deeply from the parity 'anomaly' [47,63–68], which manifests itself as an explicit symmetry breaking term at high-energy for a low-energy massless 2D Dirac fermion. To be clearer, the 2D parity symmetry is indeed an in-plane mirror symmetry [31], say about $x$, which forces $(k_x, k_y) \xrightarrow{\mathcal{M}_x} (k_x, -k_y)$, and in our model, the projected spin degrees of freedom make the related unitary transformation to be $U_{\mathcal{M}_x} = \sigma_x$, then the imposed parity symmetry $U_{\mathcal{M}_x}^\dagger h(\boldsymbol{k})U_{\mathcal{M}_x} = h(\mathcal{M}_x\boldsymbol{k})$ stands only when $k < k_c$, which forms a parity invariant regime (PIR) inside which the parity symmetry is respected. The parity invariant regime is recognized as the low-energy zone around the Dirac point with small $k$, and for larger $k > k_c$ recognized as the high-energy zone, the non-vanishing mass term breaks the 2D parity symmetry explicitly, as a consequence of regulating the effective low-energy theory [55].

### 3.1.2 Insulating case

The remaining two phases are insulating with $k_F = 0$ recognized when the chemical potential lies inside the global insulating gap, then simply

$$\sigma_H(|\mu| < d_{\min}) = \frac{e^2}{2h}\left[\text{sgn}(m(+\infty)) - \text{sgn}(m(0))\right], \tag{34}$$

for a Dirac cone, where $d_{\min} = \min(d(k))$ denotes the bound of the global gap. Clearly, $\sigma_H/(e^2/h) = 0, \pm1$ appears, notifying trivial or non-trivial phases depending on the relative signs of low and high-energy masses, with the $\pm1$ cases identified as the Chern insulator or equivalently, the quantum anomalous Hall effect. In our equivalent model, one sees from Fig. 2 that all $n \geq 2$ masses contains the same sign, and the corresponding Dirac cones are all trivial. And we come back to the statement that in a TI film, there are two gapless Dirac fermions with opposite half-quantized Hall conductance, while all other bands form paired trivial massive Dirac fermions. The quantized nature of the Hall conductance in insulating system, $\sigma_H = -Ce^2/h$, is referred to by the famous TKNN theorem [10], with its robustness against continuous non-gap-closing perturbations rooted in the topological nature of $C$ as the Chern invariant [69,70].

## 3.2 In the lattice model

Now we turn to the lattice model with a starting Dirac Hamiltonian defined on the lattice

$$h_{\text{DF}}^L = \lambda(\sin(k_x)\sigma_x + \sin(k_y)\sigma_y) + m(\boldsymbol{k})\sigma_z. \tag{35}$$

Firstly, we notice that when $m \equiv 0$, the remaining part is a naive lattice realization of single Weyl fermion, which is strongly constrained by the Nielsen-Ninomiya theorem [29,30]. There appear to be four connected Dirac points at $\Gamma, X, Y, M$, respectively. Any non-vanishing $m(\boldsymbol{k})$ will serve as a lattice regularization of the theory, with the only difference as its effectiveness

upon in gapping out which Dirac point. Essentially, here the difference with a continuum model appears, say in the latter case there is only a single gapless Dirac cone, and the infinity is usually treated by one-point compactification and the $k$-space is topologically equivalent to a sphere surface $S^2$, while on lattice the Brillouin zone geometry as a torus $T^2$ can contain non-trivial property on its periodic boundary. Such a non-trivial property is exactly reflected by the existence of four Dirac points under naive lattice realization of Dirac operator $k \cdot \sigma$. With an analogical formulation, we write

$$\sigma_H = \frac{e^2}{2h} \left[ S_X + S_Y - S_\Gamma - S_M \right], \tag{36}$$

with $S_k$ as an analogy to $m(k)/d(k)$ appearing in the continuum model. $S_k$ becomes zero when the chemical potential lies in the metallic states around $k$, and over those states certain symmetry constraint is imposed in a finite regime around, such as the parity symmetry which requires $m(\mathcal{M}_x k) = -m(k)$, and essentially, the imposed symmetry should ensure that the net Berry curvature integral contributed from the regime (constrained also by chemical potential) is zero wherever we put the Fermi level inside. On the other hand, we recognize $S_k = \text{sgn}(m(k))$ when Dirac point $k$ is gapped, and the Fermi level lies inside. The formula is further classified into two cases under additional conditions.

### 3.2.1 Gapless/metallic case

The first case corresponds to the existence of gapless Dirac fermion(s) inside a parity invariant regime. Consider an example as a single gapless Dirac fermion at $\Gamma$ point, let the Fermi level lie in the symmetry constrained regime (SCR), and we recognize

$$\sigma_H(k_F \subseteq \text{SCR}) = \frac{e^2}{2h} \left[ \text{sgn}(m(X)) + \text{sgn}(m(Y)) - \text{sgn}(m(M)) \right], \tag{37}$$

which is always half-quantized. Notice that $k_F = \{k | d(k) = \mu\}$ is now a set, representing Fermi surface wavevectors. Also notice that unlike the case in the continuum model where the regulator comes from only at infinity, here on the square lattice, a single gapless Dirac fermion owns **three** regulators. At the same time, if $\text{sgn}(m(X)) = \text{sgn}(m(Y)) = \text{sgn}(m(M))$ is recognized which makes the boundary of the Brillouin zone trivial, we get

$$\sigma_H(k_F \subseteq \text{SCR}) = \frac{e^2}{2h} \text{sgn}(m(M)). \tag{38}$$

In our equivalent model on lattice, the lowest two cones

$$h_{1,\chi}(k) = \lambda_\parallel (\sin(k_x a)\sigma_x + \sin(k_y b)\sigma_y) + \chi m_1(k)\sigma_z, \tag{39}$$

satisfy the condition, with $m_1(k) = \Theta(-m_0(k))m_0(k)$ identified. Since under our model parameter choice, it is easy to verify that $\text{sgn}(m_1(X)) = \text{sgn}(m_1(Y)) = \text{sgn}(m_1(M)) < 0$, and we write

$$\sigma_H^{1,\chi}(m_1(k_F) > 0) = -\chi \frac{e^2}{2h}, \tag{40}$$

inside the symmetry constrained regime which is now the parity invariant regime defined by $m_0(k) > 0$.

### 3.2.2 Insulating case

The second case corresponds to a globally gapped Dirac band. Now by requiring the chemical potential to lie inside the gap, the Chern number reads

$$C = \frac{1}{2} \left[ \text{sgn}(m(\Gamma)) + \text{sgn}(m(M)) - \text{sgn}(m(X)) - \text{sgn}(m(Y)) \right], \tag{41}$$

which ranges among $0, \pm 1, \pm 2$. This formula has two common versions that we will come up with in the following. The first version is the most familiar one with a trivial Brillouin boundary when $\mathrm{sgn}(m(X)) = \mathrm{sgn}(m(Y)) = \mathrm{sgn}(m(M))$ is recognized, and

$$C = \frac{1}{2}\left[\mathrm{sgn}(m(\Gamma)) - \mathrm{sgn}(m(M))\right]. \tag{42}$$

The mass term generating this formula, is usually written as

$$m(\boldsymbol{k}) = m_0 - 4t\left(\sin^2\frac{k_x}{2} + \sin^2\frac{k_y}{2}\right), \tag{43}$$

with a relatively small $|m_0|$ compared to $|t|$, and correspondingly, we have

$$C = \frac{1}{2}[\mathrm{sgn}(m_0) + \mathrm{sgn}(t)], \tag{44}$$

which is non-trivial with unit Chern number when $m_0 t > 0$. And when we relax the value of $m_0$, a better formula for this mass term is

$$C = -\frac{\mathrm{sgn}(m(X))}{2}\left[\mathrm{sgn}(m(\Gamma)) - \mathrm{sgn}(m(M))\right]. \tag{45}$$

In our equivalent model on lattice within our parameter choice as a strong topological insulator with homogeneous in-film-plane parameters, Eq. (42) is enough to describe all $n \geq 2$ massive Dirac fermions; and since from Fig. 4, all $m_{n\geq 2}(k)$ do not change sign at $\Gamma$ and $M$, they are evidently all trivial.

### 3.3 A glance in proof of half-quantization

The proof [4,31] for the half-quantization of a general band structure in 2D comes as follows, with a requirement of parity or time reversal symmetry at the Fermi surface. Without losing generality, we consider a connected Fermi surface. Recognizing the infinity as one point compactifies the $k$-space, then the existence of Fermi surface cuts the curvature integral into two parts with three boundaries where the Stokes' theorem applies

$$\frac{-2\pi\sigma_H}{e^2/h} = \oint_{\mathrm{FS}} \mathrm{d}k \cdot \mathrm{Tr}(A^M) + \oint_{\mathrm{FS}} \mathrm{d}k \cdot \mathrm{Tr}(A^L) + \oint_{\overline{\mathrm{FS}}} \mathrm{d}k \cdot \mathrm{Tr}(\tilde{A}^L), \tag{46}$$

where $A^M$ refers to the non-Abelian Berry connection (convention follows that $\mathcal{A} = i\langle u|\mathrm{d}|u\rangle$) formed by the metallic bands crossed by the Fermi surface with parity or time-reversal symmetry, while $A^L$ refers to connection of bands with lower energy, on the boundary formed by $k_F$. Essentially, the last two terms are phase integrals around one mutual boundary with opposite orientations, which will contribute an integer value [71–73] $2\pi C$. For the first term, requiring the 2D parity (i.e., mirror) symmetry at the Fermi surface leads to a local unitary transformation $U_k^M$ relating states at parity-symmetric points, which leads to

$$A_\mu^M(k) = i(U_k^M)^\dagger \partial_{k_\mu} U_k^M + (U_k^M)^\dagger A_\nu^M(\mathcal{M}k) U_k^M J_{\nu\mu}, \tag{47}$$

where $J_{\nu\mu} = \partial(\mathcal{M}k)_\nu / \partial k_\mu$ is the Jacobian matrix with $\det(J) = -1$. And similarly, requiring time reversal at Fermi surface leads to

$$A_\mu^M(k) = i(U_k^T)^\dagger \partial_{k_\mu} U_k^T - (U_k^T)^\dagger A_\mu^M(-k) U_k^T, \tag{48}$$

where $U_k^T$ is the unitary matrix relating time reversal points satisfying that $U_k^T = -(U_{-k}^T)^T$. Performing Berry phase loop integral of both sides leads to, for both symmetry restricted cases,

$$\oint_{FS} dk \cdot \text{Tr}(A^M) = \frac{i}{2} \oint_{FS} dk \cdot \text{Tr}(U_k^\dagger \nabla_k U_k) = \pi N. \tag{49}$$

Combining three terms gives

$$\sigma_H = -\frac{e^2}{h}\left(C + \frac{N}{2}\right), \tag{50}$$

with both $C$ and $N$ integers. The proof here can be easily generalized to the lattice model, by simply replacing the base manifold with a torus, and to the case when the Fermi surface consists of several separately connected components, with the curvature integral cut into more parts determined by Fermi surface position in $k$-space.

When bands related to $C$ and $N$ are fully separated, the former can be recognized as the Chern number contributed from these fully occupied bands, while the latter reduces to a quantized Fermi surface loop integral over metallic bands [74–77]. We would like to emphasize here that even though reduced to cumulating low-energy (refer to Fermi surface here) quantities, the $N$ index in our analysis has to be determined by the properties of far Fermi sea, i.e., high-energy regime. This is because the application of the Stokes theorem, which turns the Fermi sea volume integral over Berry curvature into Fermi surface line integral over Berry phase, requires a self-consistent gauge choice of the vector field. This gauge choice must not contain any singularities in the integrated volume, in order to ensure the existence of a non-singular gauge field throughout the volume.

## 3.4 View from field theory

The gapless Dirac fermion in a strong topological insulator film can be written as $\mathcal{H}_0(\boldsymbol{k}) = \lambda_\parallel \boldsymbol{\sigma} \cdot (\sin k_x, \sin k_y) + m(\boldsymbol{k})\sigma_z$ with $m(\boldsymbol{k}) = \Theta(-m_0(\boldsymbol{k}))m_0(\boldsymbol{k})$ identified, which is constructed on lattice with finite 2D Brillouin zone. The time-ordered Green function is $\mathcal{G}_0(k) = [\omega - \boldsymbol{d} \cdot \boldsymbol{\sigma}(1 - i\eta)]^{-1}$ where $k_\mu = (\omega, \boldsymbol{k})_\mu$, $\boldsymbol{d}(\boldsymbol{k}) = (\lambda_\parallel \sin k_x, \lambda \sin k_y, m(\boldsymbol{k}))$ and $\eta$ is infinitesimally small quantity. In order to study a linear electromagnetic response in the film system, we include the electromagnetic fields $\mathcal{A}$ which are coupled to the current through the interaction term $\mathcal{H}_{\text{gauge}} = \boldsymbol{j} \cdot \mathcal{A}$. The electric current density operator in the momentum space is given by $\boldsymbol{j} = \nabla_{\boldsymbol{k}} \mathcal{G}_0^{-1}(\boldsymbol{k})$. With the electromagnetic fields, the action reads ($e = \hbar = 1$)

$$S = \int_k \psi_k^\dagger \mathcal{G}_0^{-1}(k)\psi_k + \int_k \int_q \mathcal{A}^\mu(q)\psi_{k+q/2}^\dagger \partial_{k_\mu} \mathcal{G}_0^{-1}(k)\psi_{k-q/2}, \tag{51}$$

where $\int_k = \int \frac{d\omega}{2\pi} \int_{BZ} \frac{d^2\boldsymbol{k}}{(2\pi)^2}$ and the momentum $\boldsymbol{k}$ integral is performed over the whole 2D Brillouin zone. By integrating out the fermions in the action, the effective action for gauge fields $S_{\text{eff}}[\mathcal{A}]$ can be obtained by expanding to the quadratic order

$$\mathcal{S}_{\text{eff}} = \frac{1}{2} \int \frac{d^3q}{(2\pi)^3} \mathcal{A}^\mu(-q)\Pi_{\mu\nu}(q)\mathcal{A}^\nu(q), \tag{52}$$

where $\mu, \nu$ run over the space-time indices $(0, 1, 2)$ with the vacuum polarization operator defined as

$$i\Pi_{\mu\nu}(q) = \int \frac{d^3k}{(2\pi)^3} \text{Tr}[\partial_{k_\mu} \mathcal{G}_0^{-1}(k)\mathcal{G}_0(k+q/2)\partial_{k_\nu} \mathcal{G}_0^{-1}(k)\mathcal{G}_0(k-q/2)]. \tag{53}$$

There is no divergence in $\Pi_{\mu\nu}$ as the momentum integral is performed over a finite Brillouin zone due to the lattice regularization. The antisymmetric terms $\Pi_{\mu\nu}^A(q)$ can be evaluated as follows

$$\Pi_{\mu\nu}^A = \frac{1}{2\pi}\epsilon_{\mu\nu\zeta}q^\zeta C, \tag{54}$$

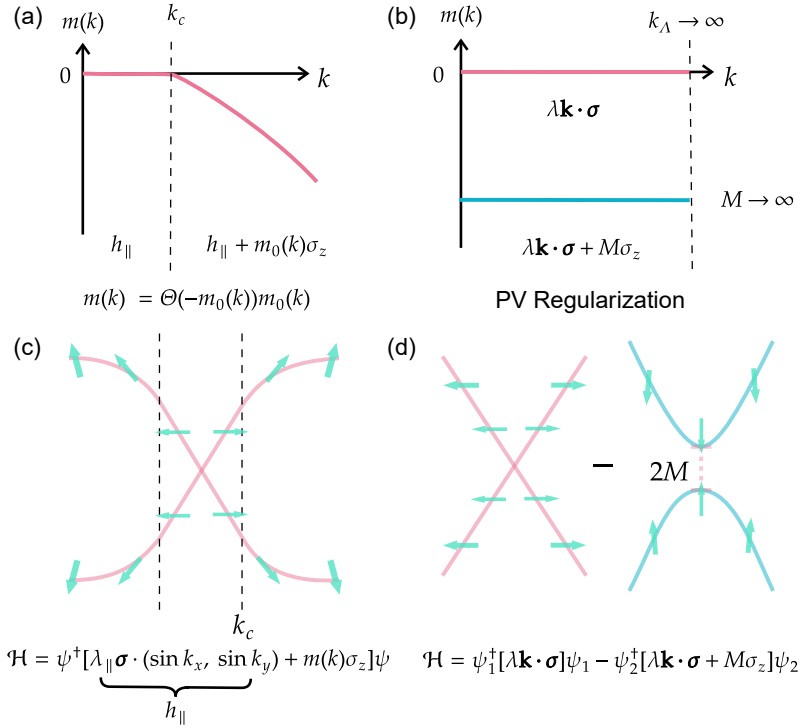

Figure 6: Regulated gapless Dirac fermion on lattice and by Pauli-Villars regularization. (a) Momentum dependent mass of regulated gapless Dirac fermion on lattice, $k_c$ is defined by $m_0(k) = 0$, which splits the mass and the dispersion in (c) of the Dirac fermion into two regions, low-energy part with $k < k_c$ and high-energy part with $k > k_c$. (b) Mass of massless Dirac fermion and of its regulator partner by Pauli-Villars treatment. (c) Dispersion of regulated gapless Dirac fermion on lattice with spin orientation. (d) Dispersion of double Dirac fermions, one massless and one massive under Pauli-Villars regularization, and to obtain convergent result, contributions from two fermions should be subtracted.

with Chern number in the case following definition that

$$C = \int_{BZ} \frac{d^2\boldsymbol{k}}{4\pi} \hat{\boldsymbol{d}} \cdot \partial_{k_x}\hat{\boldsymbol{d}} \times \partial_{k_y}\hat{\boldsymbol{d}} \,, \tag{55}$$

where $\epsilon_{\mu\nu\zeta}$ is Levi-Civita symbol and $\hat{\boldsymbol{d}} = \boldsymbol{d}/|\boldsymbol{d}|$. Finally, we obtain the Chern-Simons theory for $\mathcal{A}_\mu$

$$S_{\text{eff}}[\mathcal{A}] = \frac{C}{2\pi} \int d^3x \epsilon^{\mu\nu\zeta} \mathcal{A}_\mu \partial_\zeta \mathcal{A}_\nu \,. \tag{56}$$

For the lattice Hamiltonian $\mathcal{H}_0(\boldsymbol{k})$, we have $C = -\frac{\text{sgn}(m(\pi,\pi))}{2}$ which is a half-integer with its sign determined by the sign of $m(\pi,\pi)$. Restoring physical units, the Chern-Simons term corresponds a half quantum Hall effect

$$\langle j^\nu \rangle = \frac{\delta S_{\text{eff}}}{\delta \mathcal{A}_\nu} = \frac{\text{sgn}(m(\pi,\pi))}{2} \frac{e^2}{h} \epsilon^{\mu\nu\zeta} \partial_\zeta \mathcal{A}_\mu \,. \tag{57}$$

Notice that upon DC linear response, the result is strict.

If we now focus on the low-energy effective model of the lattice four-band Hamiltonian by neglecting higher energy states ($\propto m(\boldsymbol{k})$), which can be expressed as

$\mathcal{H}_0^{\text{low}}(\boldsymbol{k}) = \lambda_{\parallel}(k_x \sigma_1 + k_y \sigma_2)$. There is a linear ultraviolet divergence in $\Pi_{\mu\nu}(q)$ which should be regularized by Pauli-Villars method in a gauge-invariant way. In the Pauli-Villars regularization approach, we need to introduce a second Dirac field mass $M\sigma_3$. In the limit $(M \to \infty)$, the regulator field decouples from the theory, which removes the divergence in $\Pi_{\mu\nu}$, leaving a finite contribution for the crossed polarization tensor $\Pi_{\mu\nu} = \frac{\text{sgn}(M)}{4\pi} \epsilon_{\mu\nu\zeta} q^{\zeta}$. This also induces a Chern-Simons term and corresponds to a half-quantum Hall effect.

The comparison of mass configuration and band dispersion of two methods is shown in Fig. 6. The advantage of our approach for lattice realization single gapless Dirac fermion lies in its realism, as it appears naturally in a topological insulator film, and also in its conciseness of expressing topological properties with a single analytical mass term. The price here, however, is to introduce symmetry-breaking term at high-energy zone explicitly, and the form of Theta function (or its mollifier) will introduce long-range hopping in real space.

## 3.5 Unexchangeable limits

In the usual context of quantum field theory, a massive $(2+1)$-D Dirac fermion bears half-quantized Hall conductivity when the chemical potential lies inside the gap, even if the mass is infinitesimally small [14, 65, 67], under which one gets in fact a Dirac point. Such a picture relies on the limit sequence that one firstly takes $\mu \to 0$, and then the mass $m \to 0$, while on the other hand, once the sequence is inverted, say at first place, one stays at finite chemical potential $\mu$ and takes $m \to 0$, which leads to zero Hall conductivity, one gets constant zero Hall plateau when pushing $\mu \to 0$. And in this sense one realizes that a gapless Dirac point is singular, and different approaches to reach it will lead to different and even contradictory pictures.

The same thing happens in our model. Consider now a gapless Dirac fermion is perturbed by a small constant mass term

$$h = \lambda_{\parallel}(\boldsymbol{k} \cdot \boldsymbol{\sigma}) + [\delta m + \Theta(-m_0(k))m_0(k)]\sigma_z, \tag{58}$$

where for simplicity we discuss the continuum model here. Given $m_0(k) = m_0 - bk^2$ with $m_0 b > 0$, by Eq. (30) we have

$$\sigma_H = -\frac{e^2}{2h}\left[\text{sgn}(b) + \frac{\delta m}{\sqrt{\lambda_{\parallel}^2 k_F^2 + \delta m^2}}\right], \tag{59}$$

where a small $\mu$ near the Dirac point is assumed. The $k_F$ refers to the Fermi wavevector defined by $\mu = -\sqrt{\lambda_{\parallel}^2 k_F^2 + \delta m^2}$, which lies inside the valence band and satisfies $m_0(k_F) > 0$. Now the two different limits for the Hall conductivity of the gapless Dirac cone in the case read

$$\lim_{\delta m \to 0} \lim_{\mu \to 0} \sigma_H = -\frac{e^2}{2h}[\text{sgn}(b) + \text{sgn}(\delta m)], \tag{60a}$$

$$\lim_{\mu \to 0} \lim_{\delta m \to 0} \sigma_H = -\frac{e^2}{2h}\text{sgn}(b), \tag{60b}$$

i.e., first pushing chemical potential to zero and then pushing $\delta m$ to zero leads to an undefined limit that depends on the limit direction $\delta m$ takes (positive or negative), while an admittedly infinitesimal mass gap will not affect the half-quantization of the gapless Dirac cone by subsequent Fermi level tuning — not only to $\mu \to 0$ but for all possible Fermi wavevectors that lie inside the parity invariant regime [31] defined by $m_0(k) > 0$. The corresponding schematic diagram illustrating the sequential limit-taking processes upon evaluating the Hall conductivity of a regulated gapless Dirac fermion is presented in Fig. 7. In reality, which limit the measured Hall conductance takes has to depend on specific situation of the system, while for the Dirac point emerged in a purely magnetic TI, the second perspective may be deemed more realistic.

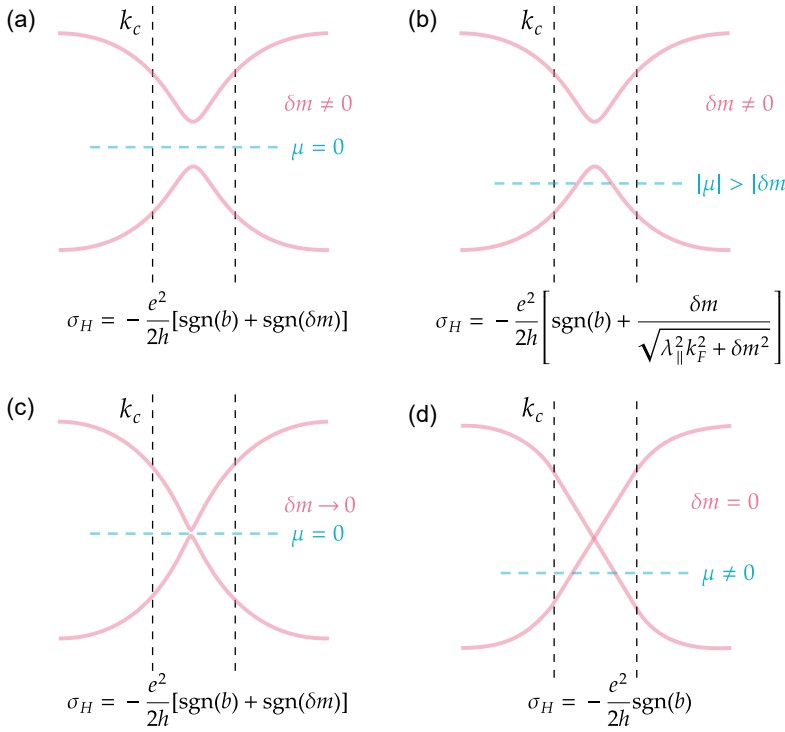

Figure 7: Schematic diagrams illustrating the limits in calculating the Hall conductivity of a regulated gapless Dirac fermion are shown below. In these diagrams, $k_c = \sqrt{m_0/b}$. (a) Initially tuning the chemical potential to $\mu = 0$ leads to integer quantized Hall conductivity. (b) Initially adjusting the chemical potential finite inside the valence band with Fermi wavevector $k_F < k_c$ results in unquantized Hall conductivity asymptotically proportional to $\delta m/k_F$. (c) Continuing from (a), pushing the small gap $\delta m \to 0$ while pinning the chemical potential at $\mu = 0$ leaves the integer of the quantized Hall conductivity invariant. (d) Continuing from (b), pushing the small gap $\delta m \to 0$ while keeping the finite chemical potential inside the valence band with $k_F < k_c$ leads to half-quantized Hall conductivity of a gapless Dirac fermion, with the sign of the Hall conductivity determined by its chirality or equivalently its high-energy mass sign.

## 4  Magnetic and orbital fields in topological insulator films

In this section we consider additional elements, such as exchange interaction, gate-voltage and orbital orders, to play their roles in the topological insulator film at the mean-field level. We identify the mean field to be $V(\boldsymbol{k}, l_z)\sigma_\mu \tau_\nu$, with single in plane wavevector and out of plane position dependence, and transform the field into the Dirac fermion representation. For instance, an induced $z$-Zeeman field $V_z(l_z)\sigma_z \tau_0$ with solely $z$-dependence and intrinsic spin-orbital coupling $H_\parallel(\boldsymbol{k})$ that only depends on $\boldsymbol{k}$ are two special cases under the formulation. For our interest, we will mainly consider magnetic exchange interaction that has been approximated to affect as an effectively mean-field Zeeman field [78] along $z$ direction, and transformation over other spin and orbital related fields are discussed and summarized later.

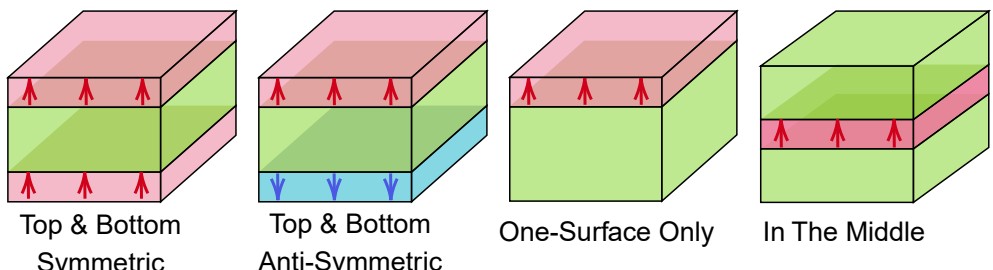

Figure 8: Several basic representative magnetic topological insulator heterostructures. From left to right: Zeeman field at top and bottom surfaces with parallel and antiparallel polarizations, at top surface and in the middle only, corresponding to basic topological phases in magnetic topological insulator film as Chern insulator, axion insulator, half-quantized anomalous Hall effect, and metallic quantized anomalous Hall effect, respectively. We use color and its gradation to emphasize the direction and strength of the Zeeman field.

## 4.1 Magnetism polarized along $z$ direction

The stated mean $z$-Zeeman field is assumed to be uniform intralayer while varies with $l_z$, and that is to say [32],

$$\mathcal{V}_Z(\boldsymbol{k}) = \sum_{l_z, \boldsymbol{k}} \Psi^\dagger_{l_z, \boldsymbol{k}} V_Z(l_z) \Psi_{l_z, \boldsymbol{k}}\,, \tag{61}$$

where

$$V_Z(l_z) \equiv V_z(l_z)\sigma_z\tau_0\,, \tag{62}$$

which acts on spin $z$. For several schematic examples with different Zeeman configurations, see Fig. 8. Its equivalent action by projection $\langle \Phi^n_m | V_Z | \Phi^{n'}_{m'}\rangle$ ($m, m' = 1, 2, 3, 4$; $n, n' = 1, \cdots, L_z$) reads

$$V(\boldsymbol{k}) = \big(\mathbf{I}_S(\boldsymbol{k})\tau_0 - \mathbf{I}_A(\boldsymbol{k})\tau_y\big)\sigma_z\,. \tag{63}$$

In the expression, two projected Hermitian matrices $\mathbf{I}_{S/A}(\boldsymbol{k})$ have been defined with elements

$$I^{nn'}_S = |C_n C_{n'}| \sum_{l_z} V_z(l_z)[\lambda^2_\perp (f^n_+)^* f^{n'}_+ + t^2_\perp \eta^n \eta^{n'} (f^n_-)^* f^{n'}_-] = (I^{n'n}_S)^*\,, \tag{64a}$$

$$iI^{nn'}_A = i|C_n C_{n'}| \sum_{l_z} V_z(l_z)\lambda_\perp t_\perp [\eta^{n'}(f^n_+)^* f^{n'}_- + \eta^n (f^n_-)^* f^{n'}_+] = -i(I^{n'n}_A)^*\,, \tag{64b}$$

where $n, n' = 1, \ldots, L_z$. Notice that $I_{S/A}$ is non-vanishing only when the symmetric/antisymmetric component of $V_z$ is non-zero. Our formula then illustrates that the Zeeman field in a TI film is brought into two classes by the discrete parity or mirror symmetry, with $S(A)$ labelling the part respects (disrespects) this symmetry. Bring the transformed Zeeman term into multi-Dirac fermions representation, and we obtain

$$H^V = \bigoplus_{n=1}^{L_z} \big[\lambda_\parallel(\sin(k_x a)\sigma_x + \sin(k_y b)\sigma_y) + m_n(\boldsymbol{k})\tau_z\sigma_z\big] + \big(\mathbf{I}_S(\boldsymbol{k})\tau_0 - \mathbf{I}_A(\boldsymbol{k})\tau_y\big)\sigma_z\,. \tag{65}$$

Under the local unitary transformation, the Zeeman field in TI film undergoes a transformation into the $\mathbf{I}$ matrices, which act as generalized Higgs fields in matrix form, generating mass through the Yukawa-like couplings among Dirac fermions in the film [55, 79]. This phenomenon occurs precisely due to the fact that the projected Zeeman terms still act on spin-$z$ component, similar to how masses affect the system. The emergence of a non-vanishing Higgs

expectation value is closely associated with the establishment of the magnetic order in the system, either by intrinsic spontaneous magnetization or a proximate magnetic field.

A closer look then classifies this action into three aspects. Firstly, the intra-Dirac cone elements $I_S^{nn}$ tell how the Zeeman field directly modifies the mass term $m_n$, and due to the trace invariance under unitary transformation, such a direct modification is significant in understanding the impact of the Zeeman field on the overall mass generation process. Secondly, the intra-block inter-Dirac cone elements $I_A^{nn}$ couple the two mirror-symmetric Dirac fermions with the same $n$-label together, and force them to recombine into two new Dirac fermions that break the mirror symmetry. Finally, the general inter-block elements $I_{S/A}^{nn'}(n \neq n')$ couple Dirac cones with different $n$-labels. Nevertheless, since the linear winding part of Dirac fermions in our equivalent TI film model (see Eq. (65)) is identity in subspace spanned by $n$ and $\tau$, the total effect of the projected Zeeman term is to modify the mass terms, i.e.,

$$\mathbf{M}(\mathbf{k})\sigma_z = \left[ \bigoplus_{n=1}^{L_z} m_n \tau_z + \mathbf{I}_S \tau_0 - \mathbf{I}_A \tau_y \right](\mathbf{k})\sigma_z, \tag{66}$$

and further diagonalization of this total mass part will give another set of $2L_z$ mass terms without affecting the winding part, i.e.,

$$\mathbf{M}(\mathbf{k}) \xrightarrow{\text{diagonalization}} \bigoplus_{n=1}^{2L_z} \tilde{m}_n(\mathbf{k}), \tag{67}$$

and accordingly, we can write down the Dirac fermion Hamiltonian under Zeeman field as

$$\tilde{H}(\mathbf{k}) = \bigoplus_{n=1}^{2L_z} \left[ \lambda_\| (\sin(k_x a)\sigma_x + \sin(k_y b)\sigma_y) + \tilde{m}_n(\mathbf{k})\sigma_z \right], \tag{68}$$

which describes the $2L_z$ Dirac fermions in a magnetic topological insulator film. Notice that the Zeeman term alters the masses of Dirac fermions thus their topological properties, which is the origin of the fruitful magnetic topological phases in the system.

The formula and discussion above are general and applies for any $z$-varying Zeeman configurations. For our consideration here, we separately discuss main cases.

### 4.1.1 Uniform field strength

In this case $V_z(l_z) \equiv V$ for any $l_z$, and it is easy to check out that

$$I_S^{nn'} = V\delta_{nn'}, \tag{69a}$$

$$I_A^{nn'} = 0, \tag{69b}$$

which offers us with an exact projection without further diagonalization as

$$H^V(\mathbf{k}) = \bigoplus_{n=1}^{L_z} \left[ \lambda_\| (\sin(k_x a)\sigma_x + \sin(k_y b)\sigma_y) + (m_n(\mathbf{k})\tau_z + V\tau_0)\sigma_z \right] = h_{n,\chi}^V(\mathbf{k}), \tag{70}$$

where each sub-block

$$h_{n,\chi}^V(\mathbf{k}) = \lambda_\| (\sin(k_x a)\sigma_x + \sin(k_y b)\sigma_y) + (\chi m_n(\mathbf{k}) + V)\sigma_z, \tag{71}$$

describes a Dirac fermion of TI film modified by a uniform Zeeman splitting $V$. This formula serves as a clear physical picture to illustrate the formation of higher Chern number in TI film, with multi-sub-bands inversion [80] generated by the direct Higgs coupling $V$, as we shall illustrate in the section thereafter.

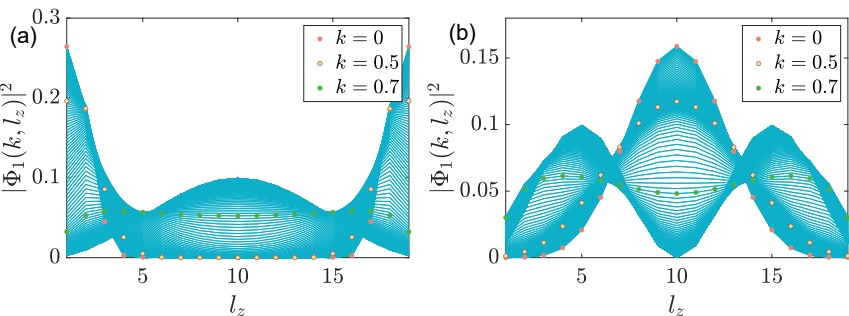

Figure 9: Basis wavefunction distribution along $z$ for (a) $n = 1$ and (b) $n = 2$, varying from $k_x = 0$ to $k_x = \pi$ with $k_y = 0$. Dots in purple light, yellow and green represent wavefunction at $k_x = 0$, $k_x = 0.5$ and $k_x = 0.7$, respectively. Total layer number $L_z = 19$.

### 4.1.2 Weak Zeeman field

When a weak Zeeman field, whose strength is comparably small to major parameters in topological insulator, especially, the bulk gap $m_0$, is applied to the topological insulator film system, its effective Hamiltonian can be obtained by considering only $n = n' = 1$ elements in the projected matrix as a cut-off approximation. The reason why we can do this lies in the basis wavefunction distribution along $z$-direction. As revealed in Fig. 9, where we have presented $n = 1$ basis wavefunction distribution for the strong topological insulator with a single Dirac cone at $\Gamma$, together with $n = 2$ basis wavefunction distribution as a representative for higher states, the surface state and higher states have little overlap in the low-energy zone (near Dirac cone, in our case the parity-invariant regime [31] around $\Gamma$ point, i.e., small $k$ area), which makes the overlap integral $I_{S/A}^{1,n \geq 2}$ approach zero in the regime. This tells that the low-energy behavior of the system under weak Zeeman field is dominated by only $I_{S/A}^{1,1}$ terms. And when we turn to high-energy part, the effective Hamiltonian for $n = 1$ is dominated by the non-vanishing mass term $m_1(k)$ since Zeeman integrals are all perturbative quantities in the case. What is more, since $n \geq 2$ bands are naturally gapped with minimal gap $m_0$, weak Zeeman field has no prominent influence to them. Based on the picture above, it suffices that we only consider $n = 1$ block with $m_1(k)$ and preserve $I_{S/A}^{1,1}$ as the influence (mass-)source at low energy. This procedure is equivalent to a cut-off approximation. Notice that since low-energy surface states distribute mainly at two surfaces, Zeeman field at these two zones should play the major role.

Now we ignore $n = 1$ index and write

$$\begin{cases} I_S(\boldsymbol{k}) = \langle \Phi_1(\boldsymbol{k})|V_Z|\Phi_1(\boldsymbol{k})\rangle \,, \\ iI_A(\boldsymbol{k}) = \langle \Phi_1(\boldsymbol{k})|V_Z|\Phi_3(\boldsymbol{k})\rangle \,, \end{cases} \tag{72}$$

which varies with wavevector $\boldsymbol{k}$, then by utilizing basis solutions above we have

$$I_S = |C|^2 \sum_{l_z} V_S(l_z)[\lambda_\perp^2 |f_+|^2 + t_\perp^2 \eta^2 |f_-|^2], \tag{73a}$$

$$iI_A = i|C|^2 \sum_{l_z} V_A(l_z)\lambda_\perp t_\perp 2\eta \, \mathrm{Re}[(f_+)^* f_-], \tag{73b}$$

respecting (anti-)symmetric part projection of Zeeman field to $z$ as

$$V_{S/A}(l_z) = \frac{V_z(l_z) \pm V_z(-l_z)}{2} \,. \tag{74}$$

Note that $I_{S/A}$ are real. The effective Hamiltonian for Zeeman term then reads

$$V_{\text{EFF}}(\boldsymbol{k}) = \left( I_S(\boldsymbol{k})\tau_0 - I_A(\boldsymbol{k})\tau_y \right)\sigma_z. \tag{75}$$

Adding this term to the lowest four-band model leads to

$$H_{\text{EFF}} = \lambda_\parallel(\sin(k_x a)\sigma_x + \sin(k_y a)\sigma_y) + m(\boldsymbol{k})\tau_z\sigma_z + I_S(\boldsymbol{k})\tau_0\sigma_z - I_A(\boldsymbol{k})\tau_y\sigma_z, \tag{76}$$

where $m(\boldsymbol{k}) = \Theta(-m_0(\boldsymbol{k}))m_0(\boldsymbol{k})$ for thick-enough film, while $I_{S/A}(\boldsymbol{k})$ are $z$-Zeeman-related integrals dependent on $\boldsymbol{k}$. This effective Hamiltonian serves as the starting point for analyzing magnetic phases in a topological insulator film within weak Zeeman regime, and we should confine the Zeeman distribution to mainly stay at the top and bottom surfaces to make the best use of it.

Notice that this Hamiltonian for the lowest surface bands is written under the (mirror) symmetric basis, and it is actually equivalent to a generalization of the commonly utilized four-band surface state Hamiltonian [48,81], which treats the top and bottom surfaces as two fundamental degrees of freedom. To show this, we introduce a two-step unitary transformation $U = U_1 U_2$ with $U_1 = e^{i\pi\tau_y\sigma_z/4}$ and $U_2 = e^{-i\pi\tau_x/4}$, which combines the mirror symmetric basis and transforms the Hamiltonian into the 'surface state representation' as

$$\begin{aligned}H_S &= U^\dagger H_{\text{EFF}} U \\ &= \begin{pmatrix} -\lambda_\parallel(\sin(k_x a)\sigma_y - \sin(k_y a)\sigma_x) + I_+(\boldsymbol{k})\sigma_z & m(\boldsymbol{k}) \\ m(\boldsymbol{k}) & \lambda_\parallel(\sin(k_x a)\sigma_y - \sin(k_y a)\sigma_x) + I_-(\boldsymbol{k})\sigma_z \end{pmatrix},\end{aligned} \tag{77}$$

with $I_\pm = I_S \pm I_A$. When $k < k_c$ with the projecting basis composing of surface states, $I_+$ and $I_-$ can be recognized approximately as the Zeeman field strengths at top and bottom, respectively. This Hamiltonian utilizes the same Dirac matrices as the usual four-band surface state Hamiltonian, but includes $k$-dependent projected Zeeman terms $I_{S/A}(\boldsymbol{k})$ and mass term $m(\boldsymbol{k})$. The low-energy form $m(0)$ is commonly referred to as the finite-size coupling between the top and bottom surface states in an ultra-thin film. For a sufficiently thick film, the $k$-dependence of this mass term becomes crucial, since at low energies it is zero and offers us two well-separated surface states localized at the top and bottom surfaces, while at high energies it tells us that the surface bands will inevitably mix together, rendering the 'top' and 'bottom' labels ineffective as quantum numbers in this much broader zone. This observation is consistent with the wavefunction distribution in Fig. 9(a), where the low-energy surface states are predominantly localized on the top and bottom surfaces, whereas the high-energy states spread into the bulk. As a result, a well-defined Chern number cannot be assigned to a single surface but must instead involve contributions from bulk states. Furthermore, as we have discussed, the Theta function form of the lowest mass term $m(\boldsymbol{k})$ differs from the conventional approach, which assumes a mass term of the form $\tilde{m}_0 + bk^2$ similar to the bulk band. The usual choice only restores parity symmetry near $k = 0$ as $\tilde{m}_0$ goes to zero, and fails to fully capture the topological nature of the surface gapless Dirac fermion that contains parity symmetry in a finite but much larger area by $k < k_c$.

**Effective mass treatment**  Diagonalization of the mass part in the weak Zeeman field case shows much less complexity than that in Eq. (67), and is accessible analytically. A careful look on Eq. (76) tells that we can treat all the latter-three terms as mass terms, since by $\tau$-space diagonalization

$$U_M^\dagger \left[ m\tau_z + I_S\tau_0 - I_A\tau_y \right] U_M = \begin{pmatrix} \tilde{m}_+ & \\ & \tilde{m}_- \end{pmatrix}, \tag{78}$$

where the defined unitary matrix reads

$$U_M = \frac{1}{\sqrt{2}} \begin{pmatrix} i\,\text{sgn}(I_A)\sqrt{1 + \frac{m}{M}} & \sqrt{1 - \frac{m}{M}} \\ \sqrt{1 - \frac{m}{M}} & i\,\text{sgn}(I_A)\sqrt{1 + \frac{m}{M}} \end{pmatrix}, \tag{79}$$

with $M(\boldsymbol{k}) = \sqrt{m^2(\boldsymbol{k}) + I_A^2(\boldsymbol{k})}$, we can write $\tilde{H}_{\mathrm{EFF}} = \oplus_{\chi=\pm}\tilde{H}_\chi$ with

$$\tilde{H}_\chi = \lambda_\parallel(\sin(k_x a)\sigma_x + \sin(k_y a)\sigma_y) + \tilde{m}_\chi(\boldsymbol{k})\sigma_z, \tag{80}$$

where the effective mass is defined as

$$\tilde{m}_\chi(\boldsymbol{k}) \equiv I_S(\boldsymbol{k}) + \chi\sqrt{m^2(\boldsymbol{k}) + I_A^2(\boldsymbol{k})}. \tag{81}$$

This equation illustrates minimally the mass generation brought by the matrix form Higgs fields, which are reduced into merely two components $I_{S/A}(k)$ here. The ultimate effect given by the Zeeman field action to the system is reduced to a correction of the Dirac mass term, which is responsible for the possible non-trivial topology of the system. The treatment here relies on the sign invariance of $I_A$ inside the parity invariant regime, which ensures the global gauge consistency for the transformation.

Notice that the gap is now determined by

$$\Delta_\chi = 2|\tilde{m}_\chi(0)| = 2|I_S(0) + \chi|I_A(0)||, \tag{82}$$

which is non-zero (gapped) as long as $|I_S(0)| \neq |I_A(0)|$. The $\chi$-Chern number, according to Eq. (42), for each gapped surface state is written as

$$C_\chi = \frac{1}{2}[\mathrm{sgn}(\tilde{m}_\chi(0)) - \mathrm{sgn}(\tilde{m}_\chi(\pi,\pi))], \tag{83}$$

which, by utilizing the fact that $m(0) = 0$ and Zeeman field is added perturbatively so that $m(\boldsymbol{k})$ dominates at $(\pi,\pi)$, we obtain that

$$C_\chi = \frac{1}{2}[\mathrm{sgn}(I_S(0) + \chi|I_A(0)|) - \chi] = -\chi\Theta(-|I_A(0)| - \chi I_S(0)). \tag{84}$$

This formula works in the chosen parameter regime $0 < m_0 < 4t_\perp$ within weak Zeeman treatment.

### 4.1.3 Strong Zeeman field

For a general strong Zeeman field whose strength is comparably large enough relative to the system parameters (mainly bulk gap $m_0$) or even stronger, with arbitrary configuration along $z$ direction, both the uniform and the weak criteria fail, and in this case, we usually have to adopt the most general formula from Eq. (65), whose topological property is revealed after a further diagonalization of mass terms given by Eq. (67), which turns the total Hamiltonian again into a direct sum of a series of Dirac fermions shown in Eq. (68). Then based on our discussion in 3, the Hall conductivity of each single Dirac fermion is determined, from which we can analyze the topological property of the system.

## 4.2 Other fields

In the subsection, we present more examples of spin and orbital fields other than the $z$-Zeeman field discussed above, and the results are listed in Table 1. The signals appearing here only apply in the subsection. The list of results reveals the power of our general procedure, and is enlightening for discovering more topological phases driven by diverse physical origins.

For a given field $V(\boldsymbol{k}, l_z)\sigma_\mu\tau_\nu$, the transformation follows similarly by organizing the projected elements $\sum_{l_z} V(\boldsymbol{k}, l_z)\langle\Phi_m^n(\boldsymbol{k}, l_z)|\sigma_\mu\tau_\nu|\Phi_{m'}^{n'}(\boldsymbol{k}, l_z)\rangle$ $(m, m' = 1, 2, 3, 4;\ n, n' = 1, \cdots, L_z)$ aligned with the sequence of the basis. The form of field after transformation will always be two $L_z \times L_z$ matrix fields differing by $z$-parity symmetry labels, with $S$ counting for symmetric distribution and $A$ for the opposite. Each matrix field will also be attached with a new $4 \times 4$ Dirac matrix.

Table 1: Different fields and their forms under the transformation.

| Name of field | Original field expression | Field after transformation | Kernel |
|---|---|---|---|
| Spin-orbital coupling | $\lambda_{\parallel}[\sin(k_x a)\sigma_x\tau_x + \sin(k_y b)\sigma_y\tau_x]$ | $\bigoplus_{n=1}^{L_z}\lambda_{\parallel}\tau_0(\sin(k_x a)\sigma_x + \sin(k_y b)\sigma_y)$ | $F_{S+}$ |
| Zeeman field | $Z_z(l_z)\sigma_z\tau_0$ | $\left(\mathbb{I}_S^z(\boldsymbol{k})\tau_0 - \mathbb{I}_A^z(\boldsymbol{k})\tau_y\right)\sigma_z$ | $F_{S+}$, $F_{A+}$ |
| | $Z_x(l_z)\sigma_x\tau_0$ | $\left(\mathbb{I}_S^x(\boldsymbol{k})\tau_x - \mathbb{I}_A^x(\boldsymbol{k})\tau_z\right)\sigma_x$ | $F_{S-}$, $F_{A-}$ |
| | $Z_y(l_z)\sigma_y\tau_0$ | $\left(\mathbb{I}_S^y(\boldsymbol{k})\tau_x - \mathbb{I}_A^y(\boldsymbol{k})\tau_z\right)\sigma_y$ | $F_{S-}$, $F_{A-}$ |
| Gate-voltage | $G(l_z)\sigma_0\tau_0$ | $\left(\mathbf{G}_S(\boldsymbol{k})\tau_0 - \mathbf{G}_A(\boldsymbol{k})\tau_y\right)\sigma_0$ | $F_{S+}$, $F_{A+}$ |
| Oribital field | $O_y(l_z)\sigma_0\tau_y$ | $\left(\mathbf{O}_A^y(\boldsymbol{k})\tau_0 - \mathbf{O}_S^y(\boldsymbol{k})\tau_y\right)\sigma_z$ | $F_{S+}$, $F_{A+}$ |
| | $O_x(l_z)\sigma_0\tau_x$ | $\left(\mathbf{O}_A^x(\boldsymbol{k})\tau_z - \mathbf{O}_S^x(\boldsymbol{k})\tau_x\right)\sigma_0$ | $-F_{S-}$, $-F_{A-}$ |
| | $O_z(l_z)\sigma_0\tau_z$ | $\left(\mathbf{O}_A^z(\boldsymbol{k})\tau_x - \mathbf{O}_S^z(\boldsymbol{k})\tau_z\right)\sigma_z$ | $F_{S-}$, $-F_{A-}$ |

To express matrix quantities $\mathbf{I}, \mathbf{G}, \mathbf{O}$ in Table 1, we introduce the momentum-dependent matrix-form acting functional $\mathcal{F}_k$ over $V$ field that generates projected matrix component like

$$\mathcal{F}_k^{nn'}[V] = \sum_{l_z} V(\boldsymbol{k}, l_z) F_V^{nn'}(\boldsymbol{k}, l_z) = (\mathcal{F}_k^{n'n}[V])^*, \tag{85}$$

where the summation kernel $F_V^{nn'}(\boldsymbol{k}, l_z)$ depends on different Dirac matrix the untransformed field carries. However, in practice, we find that the non-vanishing components in the transformed field matrix are only generated by four kinds of summation kernels,

$$F_{S+}^{nn'}(\boldsymbol{k}, l_z) = |C_n C_{n'}|[\lambda_{\perp}^2 (f_+^n)^* f_+^{n'} + t_{\perp}^2 \eta^n \eta^{n'} (f_-^n)^* f_-^{n'}], \tag{86a}$$

$$F_{A+}^{nn'}(\boldsymbol{k}, l_z) = |C_n C_{n'}|\lambda_{\perp} t_{\perp}[\eta^{n'} (f_+^n)^* f_-^{n'} + \eta^n (f_-^n)^* f_+^{n'}], \tag{86b}$$

$$F_{S-}^{nn'}(\boldsymbol{k}, l_z) = |C_n C_{n'}|[-\lambda_{\perp}^2 (f_+^n)^* f_+^{n'} + t_{\perp}^2 \eta^n \eta^{n'} (f_-^n)^* f_-^{n'}], \tag{86c}$$

$$F_{A-}^{nn'}(\boldsymbol{k}, l_z) = |C_n C_{n'}|(-i)\lambda_{\perp} t_{\perp}[\eta^{n'} (f_+^n)^* f_-^{n'} - \eta^n (f_-^n)^* f_+^{n'}], \tag{86d}$$

different by symmetry requirement and an inner sign. In the table the symmetry labels between the transformed fields and the summation kernels are in one-to-one correspondence.

The table can be longer once one considers more kinds of Dirac matrices. This procedure above is general, powerful while easy to understand. Despite the easiness of the transformation, the non-trivial difficult part is to endow physical meaning to the attached fields, both before transformation and after. For instance, the spin-orbital coupling remains its meaning after the transformation, while being block-diagonal in the Dirac fermion representation; the $z$-Zeeman field, as discussed above, is transformed into two matrix form Higgs fields, which stand as the effective mass generators.

**Spin-orbital duality** Interestingly, we see that the $y$-orbital order is transformed to attach the same Dirac matrices as the transformed $z$-Zeeman field, but with symmetry indices of matrix quantities exchanged. This relation tells that, as long as some topological phase is discovered with $z$-Zeeman field $Z_z = Z_{z,S} + Z_{z,A}$, another phase with the same topological index can immediately be identified with $y$-orbital order satisfying that $O_{y,A} = Z_{z,S}, O_{y,S} = Z_{z,A}$. For instance, we show the dual phases formed by $\sigma_z$ and $\tau_y$ orders in Table 2, the Chern insulator, aka quantum anomalous Hall effect (QAHE), the axion insulator, the half QAHE and the metallic QAHE as several typical phases in magnetic topological insulators as we will discuss below. Here one has to notice that for the metallic QAHE [32], which requires a relatively strong magnetism in the middle of a topological insulator film, the corresponding $\tau_y$ orbital order induced metallic QAHE requires a higher threshold for the antisymmetric field strength $O_{y,A}^m$, due to the odd function nature which forces $O_{y,A}^m(L_z/2) = 0$.

Table 2: Duality of typical topological phases induced by spin order $\sigma_z$ and orbital order $\tau_y$. $t, b, m$ for top, bottom, middle and $S, A$ for symmetric, antisymmetric distribution of fields, respectively.

| Name of phase | $\sigma_z$ configuration | $\tau_y$ configuration |
|---|---|---|
| Chern insulator | $Z_z^t = Z_z^b \neq 0$ | $O_y^t = -O_y^b \neq 0$ |
| Axion insulator | $Z_z^t = -Z_z^b \neq 0$ | $O_y^t = O_y^b \neq 0$ |
| Half QAHE | $Z_z^t \neq 0, Z_z^b = 0$ | $O_y^t \neq 0, O_y^b = 0$ |
| Metallic QAHE | $Z_{z,S}^m$ strong | $O_{y,A}^m$ strong |

Following the effective mass treatment above, we can furthermore construct quantitative model unifying the two orders. There are now totally five mass terms that read

$$\mathbf{M}(\boldsymbol{k}) = \left[ \bigoplus_{n=1}^{L_z} m_n \tau_z + (\mathbf{I}_S^z + \mathbf{O}_A^y)\tau_0 - (\mathbf{I}_A^z + \mathbf{O}_S^y)\tau_y \right](\boldsymbol{k}), \tag{87}$$

and a similar diagonalization leads to the effective masses

$$\mathbf{M}(\boldsymbol{k}) \overset{\text{diagonalization}}{\longrightarrow} \bigoplus_{n=1}^{2L_z} \tilde{m}_n(\boldsymbol{k}), \tag{88}$$

without affecting the spin-orbital coupling field (the linear winding part). On the other hand, in the context of weak field, we only preserve $n = n' = 1$ components and write down mass terms for $n = 1$ block as

$$\left[ m\tau_z + (I_S^z + O_A^y)\tau_0 - (I_A^z + O_S^y)\tau_y \right](\boldsymbol{k}), \tag{89}$$

with $n = 1$ label ignored. Here merely a substitution $I_S \to I_S^z + O_A^y$, $I_A \to I_A^z + O_S^y$ happened compare with Eq. (76), and a similar diagonalization leads to two effective masses for the surface Dirac bands as

$$\tilde{m}_\chi(\boldsymbol{k}) = (I_S^z + O_A^y)(\boldsymbol{k}) + \chi \sqrt{m^2(\boldsymbol{k}) + (I_A^z + O_S^y)^2(\boldsymbol{k})}, \tag{90}$$

from which the synergistic and competing relations between $\sigma_z$ and $\tau_y$ orders are shown more explicitly.

# 5 Topological phases with weak field

Counting on the mean strength of the magnetic exchange interaction, our exploration can be further divided into two main branches as weak and strong Zeeman fields. The division follows simply from the criterion whether the phase can be described within the $n = 1$ frame, or equivalently, whether Eq. (76) from weak Zeeman field approximation is applicable. If it is the case, we identify the phase to lie inside the weak field regime, as we shall discuss here.

## 5.1 Half quantum mirror Hall effect: A non-magnetic film with mirror symmetry

The topological insulator film itself without adding any external ingredients or interactions, but with an intrinsic mirror symmetry, possibly like rhombohedral 3D TI $Bi_2(Se, Te)_3$ along

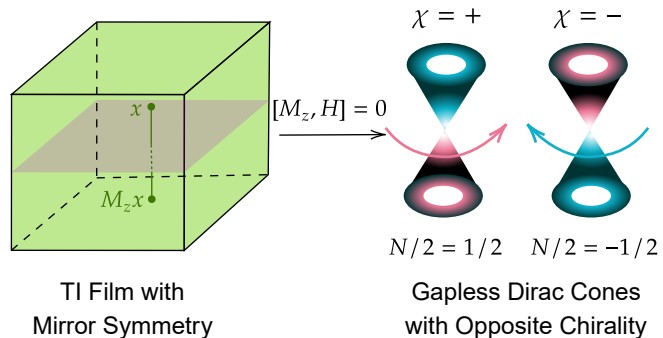

TI Film with
Mirror Symmetry

Gapless Dirac Cones
with Opposite Chirality

Figure 10: Schematic diagram of the half quantum mirror Hall effect. The lowest four bands of a topological insulator film with mirror symmetry (left) are classified into two gapless Dirac cones with opposite chiralities labelled by the eigenvalues of $z$-mirror operator.

the $[100]$ direction or with a mirror twin-boundary along $[001]$ direction, is already interesting enough and exhibits a novel topological phase [33], namely, the half quantum mirror Hall effect shown in Fig. 10, which reveals measurable parity anomaly physics. A general film Hamiltonian reads $\mathcal{H} = \sum_{l_z, l'_z, \mathbf{k}} \Psi^{\dagger}_{l_z, \mathbf{k}} H(l_z, l'_z, \mathbf{k}) \Psi_{l'_z, \mathbf{k}}$ with $\mathbf{k} = (k_x, k_y)$, and the out of film plane mirror symmetry $\mathcal{M}_z$ emerges as a combination of inversion and $C_{2z}$ rotation that reads $\mathcal{M}_z \Psi_{l_z, \mathbf{k}} \mathcal{M}_z^{-1} = U_z \Psi_{-l_z, \mathbf{k}}$, where $U_z$ is a unitary matrix. Requiring such a symmetry over the system Hamiltonian leads to the condition $U_z^{\dagger} H(l_z, l'_z, \mathbf{k},) U_z = H(-l_z, -l'_z, \mathbf{k})$. It is then possible to write down the mirror operator under $\{\Psi_{\mathbf{k}, l_z}\}$ as $M_z = C_{2z} P$, with $U_z$ as its anti-diagonal elements, and the Hamiltonian can be projected into decoupled mirror-labelled parts as

$$H_{\chi} = P^{M_z}_{\chi} H, \qquad P^{M_z}_{\chi} = \frac{1 + i\chi M_z}{2}, \tag{91}$$

with $\chi$ labelling the eigenvalue of the mirror operator. Each $H_{\chi}$ is yet again a complete system whose non-trivial property is revealed by the (zero-temperature, ignored below) mirror Hall conductivity

$$\sigma^{\chi}_H = \frac{e^2}{h} \frac{\mathrm{Im}}{\pi} \left[ \sum_{E^{\chi}_n < \mu < E^{\chi}_m} \int \mathrm{d}^2 k \frac{\bar{v}^{mn,\chi}_x \bar{v}^{nm,\chi}_y}{(E^{\chi}_n - E^{\chi}_m)^2} \right], \tag{92}$$

where $\bar{v}^{mn,\chi}_i = \langle n^{\chi} | \partial_{k_i} H^{\chi} | m^{\chi} \rangle$ is the expectation value of the mirror velocity operator evaluated over eigenstates of the mirror-projected Hamiltonian. Clearly, this is just the usual Kubo formula [62] evaluated over the projected Hamiltonian $H_{\chi}$, and thanks to the imposed mirror symmetry, two parts with mirror label $\chi = \pm$ do not communicate with each other and are totally decoupled.

The gapless pair of Dirac fermions in a topological insulator film causes the half quantum mirror Hall effect. Here in the concrete model the anti-diagonal elements of mirror operator read $U_z = -i\sigma_z \tau_z$, which is projected into $\tau_z$ under multi-Dirac fermions representation (see Appendix B.), indicated by $\chi = \pm$ as its eigenvalue in the effective Hamiltonian. The gapless $n = 1$ Dirac fermions in the TI film read

$$H_{n=1} = H_{\mathrm{surf},+} \oplus H_{\mathrm{surf},=}, \tag{93}$$

where each block with mirror label reads

$$H_{\mathrm{surf},\chi} = \lambda_{\parallel}(\sin(k_x a)\sigma_x + \sin(k_y a)\sigma_y) + \chi m(\mathbf{k})\sigma_z, \tag{94}$$

with $m(\mathbf{k}) = \Theta(-m_0(\mathbf{k})) m_0(\mathbf{k})$ identified. To show the nature of the half quantum mirror

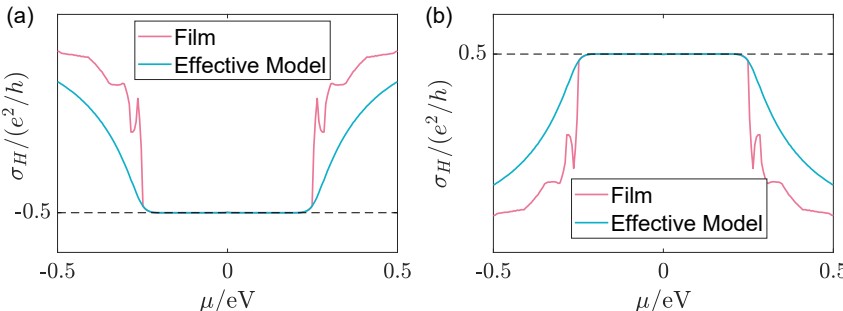

Figure 11: Half-quantized mirror Hall metal: total layer number $L_z = 19$. Results are presented for (a) $\chi = +$, and (b) $\chi = -$. Both results from direct calculation with TI film model and effective model with $h_{n=1,\chi}$ sub-blocks are shown.

Hall effect, we calculate the Hall conductivity of $H_\chi$ obtained from the mirror-projected TI film Hamiltonian, and of the split Dirac fermion $H_{\text{surf},\chi}$. The results are shown in Fig. 11, where the half-quantized transverse conductivity nature is shown for each $\chi$ part with inverse signs, indicating quantum spin Hall like physics [12, 43–45, 82–86], while the topological origin of the half-quantized mirror Hall conductivity is bound with the metallic gapless Dirac fermions [33]. Their massless low-energy parts distribute mirror-(anti-)symmetrically at both top and bottom surfaces of the TI film as a result from the bulk-boundary correspondence of 3D strong topological insulator [17], corresponding to states with mass $\pm\Theta(-m_0(\boldsymbol{k}))m_0(\boldsymbol{k})$ term at $m_0(\boldsymbol{k}) > 0$. Here, the symmetry statement is traced back to our basis, which is chosen to distribute along $z$ either mirror symmetrically or anti-symmetrically (see Appendix B). As a complete band, the surface Dirac cone does not end at a finite wavevector, but gradually emerges into the bulk with a regulated non-zero mass term represented by $\Theta(-m_0(\boldsymbol{k}))m_0(\boldsymbol{k})$ at $m_0(\boldsymbol{k}) < 0$, and it is this non-vanishing high-energy part that ultimately gives rise to the half-quantized Hall conductivity, as discussed in Section 3, which finally reads by Eq. (40) as $\sigma_H^\chi = -\chi e^2/2h$, when the Fermi surface satisfies that $m_0(k_F) > 0$.

The physically observable effect generated by the phase is embedded in the mirror Hall conductivity [33], which is defined as

$$\sigma_H^{\text{Mirror}} = \sum_\chi \chi \sigma_H^\chi, \tag{95}$$

and equals to quantum unit $-e^2/h$ in the case. The quantity reveals that, though, by opposite Hall conductivity, the charge current by a transverse electrical field vanishes as $\sigma_H = \sum_\chi \sigma_H^\chi = 0$, the 'mirror' current does not, similar to that in quantum spin Hall effect. Nevertheless, a better way of looking at the half quantum mirror Hall effect may start from treating it as an intrinsic 'spin' Hall effect in metal, while the effect shows quantization with its transverse 'spin' Hall conductivity that shares a topological origin deeply related to the parity anomaly, and replacing 'spin' with 'mirror' leads to the observation that in different mirror sectors, the mirror current and the charge current will be either parallel or anti-parallel with the same quantized magnitude. Such a way of narration also lies in the lineage of induced dissipationless mirror current and dissipative longitudinal current, as they are both generated by metallic gapless Dirac fermions. To detect the mirror current, non-local electrical transport signals [87–89] are needed, while to reveal the quantized nature, one needs to perform a series of measurements to fully separate the dissipationless and dissipative currents [33], by changing the sample width and noticing the scale invariance of the Hall conductance.

## 5.2 Quantum anomalous Hall effect: Chern Insulators

The Chern insulator is identified as an insulating phase which hosts the quantum Hall effect [90] with quantized Hall conductance, while without the need of applying an external magnetic field to form Landau levels [91]. The key ingredient lies in the breaking of time-reversal symmetry, which makes the non-vanishing Hall conductivity possible, as studied extensively in the anomalous Hall effect [92]. The quantization nature, on the other hand, is determined by the Berry phase flux integral over the Brillouin zone, which is an integer known as the first Chern number [10, 47, 69, 93–95]. An insulator with a non-zero Chern number is known to host gapless chiral edge modes [24] that circulate around the system dissipationlessly without backscattering [96]. Essentially, the number of these modes is equal to the Chern invariant, as a physical realization of the index theorem by bulk-boundary correspondence [13, 25, 26, 97]. It is usually argued that to realize a Chern insulator in a realistic material, relatively strong spin-orbital coupling together with internal magnetism are needed [98].

With confined geometry, the topological insulator film is predicted [48, 50, 99] to host the quantum anomalous Hall effect (QAHE) with proper magnetism, either by magnetic doping approach [49, 100–104] like Cr and V doped $(Bi,Sb)_2Te_3$, magnetic proximity effect [105] in the sandwich heterostructures of $(Zn, Cr)Te/(Bi, Sb)_2Te_3/(Zn, Cr)Te$ or establishing intrinsic magnetic order [106–108] in materials like $MnBi_2Te_4$ with an odd layer number. In this sense three typical cases realizing the Chern insulating phase are presented in Fig. 12, with uniform Zeeman field (to make consistency with discussion here, the Zeeman strength here is still chosen to be weak, while the uniformly strong strength case is left to be discussed in the higher Chern number case later on), symmetric top and bottom surface Zeeman fields configuration and an asymmetric configuration which does not break the holistic polarization, by which we mean that the symmetric ingredient in the configuration overwhelms the asymmetric one. The common feature these realizations share is the parallel polarization of the top and bottom surface-magnetism vertical to the TI film plane, effectively as the Zeeman field directions that point to both up or down.

The verification of the three cases is brought out by numerical calculations with both TI film and weak Zeeman effective four-band models, as revealed in Fig. 13, Fig. 14 and Fig. 15, respectively. Besides the bands in (a) that all show Zeeman-gapped feature with perfect correspondence between two methods, the Hall conductivity in (c) pictures captures the essence of a Chern insulator with an integer Chern number quantifying the quantized Hall plateau magnitude. What is more, the calculated $I_{S/A}$ in (b) and Hall conductivity in (d) for $\tilde{H}_\chi$ reveal more about physics behind the phenomenon. Below, based on the symmetric or asymmetric Zeeman configurations, we further divide the discussion into two classes.

### 5.2.1 Symmetric magnetic structure

In this class,

$$\begin{cases} I_S \neq 0, \\ I_A = 0, \end{cases} \tag{96}$$

and the given first two cases satisfy the condition. In case I and II, the symmetric Zeeman distribution leads to a vanishing $I_A$, and the effective mass, according to Eq. (81), is written as

$$\tilde{m}_\chi(\boldsymbol{k}) = I_S(\boldsymbol{k}) + \chi|m(\boldsymbol{k})|, \quad I_S > 0, \tag{97}$$

it is thus clear that under the circumstance, $\chi = -$ branch will contain a mass sign change from Dirac point $\Gamma = (0,0)$ to high-energy point $M = (\pi, \pi)$, and is topologically non-trivial with unit Chern number given by Eq. (83), while $\chi = +$ mass remains positive and leads to a trivially gapped surface band. And this composes of the explanation of the $\chi$-dependent Hall conductivity for the first two cases.

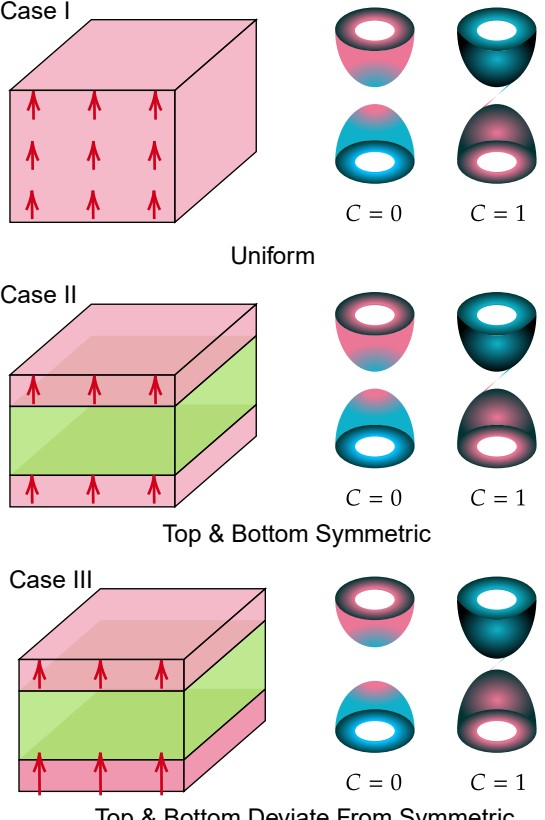

Figure 12: Schematic diagram of three typical Chern insulator cases with pairing gapped lowest Dirac cones responsible for the phase. From top to bottom: Case I: Chern insulator with uniform magnetism whose polarization contains a non-vanishing component vertical to the TI film; Case II: Chern insulator with symmetric top and bottom magnetism; Case III: Chern insulator with top and bottom magnetism that deviates from symmetric distribution, but the polarization direction remains the same. In all three cases, two gapped Dirac cones, where the gap comes from the gapped surface states, are present with one trivial cone and one cone with a unit Chern number. In the third case we deliberately tune the gap in the diagram to emphasize that it is the cone with a smaller gap that is non-trivial.

### 5.2.2 Asymmetric magnetic structure

In this case,

$$
\begin{cases}
I_S \neq 0, \\
I_A \neq 0, \\
|I_S| > |I_A|,
\end{cases}
\tag{98}
$$

i.e., an imbalance between top and bottom Zeeman strength appears, while their directions remain parallel so that the symmetric component overwhelms, as reflected by the case III. Now we observe that in Fig. 15 (d) the $\chi = -$ branch is non-trivial with unit quantized Hall plateau, and $\chi = +$ branch is trivial with a broader zero-Hall plateau, this means that the non-trivial $\chi = -$ band has a smaller gap than the $\chi = +$ band, as revealed in Fig. 15 (a). Lifting this to some principle, we claim that *the surface band with a smaller magnetic gap is non-trivial for a Chern insulator film*. To gain insight from the phenomenon, notice that in this case, both $I_S$ and $I_A$ are non-vanishing, but generally $I_S > |I_A| > 0$ since the Zeeman configuration is closer to the symmetric case, i.e. $V_S > |V_A| > 0$ near two surfaces in this case. The above observation

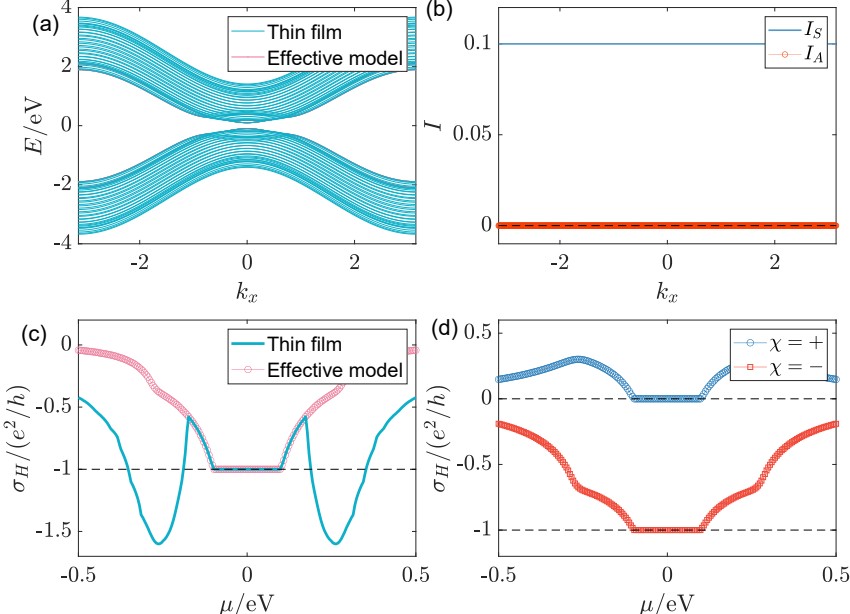

Figure 13: Chern insulator case I: total layer number $L_z = 19$ with uniform Zeeman field $V_z \equiv 0.1$ eV. (a) Comparison of band structure from TI film model and effective four-band Hamiltonian. (b) Calculated $I_S(\boldsymbol{k})$ and $I_A(\boldsymbol{k})$. (c) Calculated Hall conductivity from TI film model and effective four-band Hamiltonian. (d) Hall conductivity for $\chi = \pm$.

leads to

$$\begin{cases} \tilde{m}_\chi(0) = I_S(0) + \chi |I_A(0)| > 0, \\ \tilde{m}_\chi(M) = I_S(M) + \chi \sqrt{m^2(M) + I_A^2(M)} \sim \chi |m(M)|, \end{cases} \tag{99}$$

and since non-trivial topology requires mass inversion, we conclude that $\tilde{m}_-$ is non-trivial with unit Chern number while $\chi = +$ is trivial, and clearly the gap $\Delta = 2|\tilde{m}(0)|$ tells that $\Delta_- < \Delta_+$.

Pictures and discussions above complete the case study for the Chern insulator phase here. Notice that in the typical cases given above, the Zeeman field directs along $z$-positive axis, and it is always the $\chi = -$ band that has $-e^2/h$ Hall conductivity while the $\chi = +$ band is trivial with zero Hall contribution, i.e., it is a $1 + 0$ combination with the sign of Hall conductivity determined by the polarization direction of the Zeeman field, as we shall illustrate further below.

Generalization of the picture above about the Chern insulator phase in TI film to arbitrary weak Zeeman configuration that varies layer by layer is presented here. According to Eq. (84), the non-trivial condition is satisfied whenever $|I_S| > |I_A|$, i.e., symmetric Zeeman distribution overwhelms asymmetric configuration, and especially there exists a $\chi$ for which it holds that

$$-\chi I_S(0) > |I_A(0)|, \tag{100}$$

and correspondingly we have

$$C_\chi = -\chi, \qquad C_{\bar\chi} = 0, \tag{101}$$

with $\bar\chi = -\chi$ identified. This tells us that while one of the two gapped surface Dirac fermions becomes topologically non-trivial, carrying non-vanishing Chern index of unit, the other gapped cone is driven into a topologically trivial band. Then totally the system owns unit Chern number and quantized Hall conductivity. Meanwhile, by definition of $I_S$ in Eq. (73a), one deduces that when $I_S(0) > 0$ which corresponds to a general $z$-up $V_S$ configuration, it is

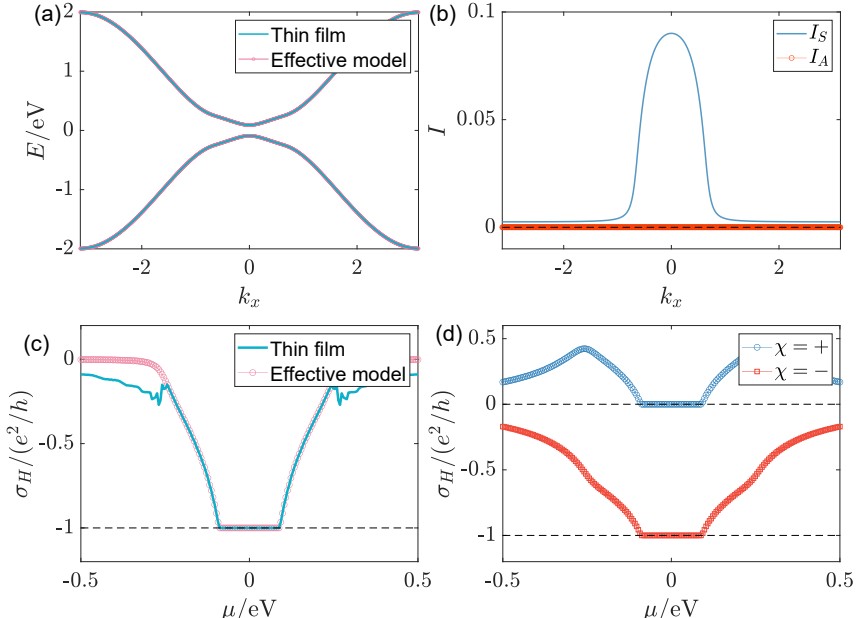

Figure 14: Chern insulator case II: total layer number $L_z = 19$ with symmetric Zeeman field $V_z(l_z) = 0.1$ eV at top and bottom 2 layers. (a) Comparison of band structure from the lowest four bands of TI film model and effective four-band Hamiltonian. (b) Calculated $I_S(\boldsymbol{k})$ and $I_A(\boldsymbol{k})$. (c) Calculated Hall conductivity from TI film model and effective four-band Hamiltonian. (d) Hall conductivity for $\chi = \pm$.

$\chi = -$ that satisfies the condition, vice versa, which allows us to write

$$\begin{cases} C_- = 1, \ C_+ = 0, & \text{for } I_S(0) > 0, \\ C_+ = -1, \ C_- = 0, & \text{for } I_S(0) < 0, \end{cases} \tag{102}$$

with $I_S(0)$ contributed mainly from surfaces. There is indeed no threshold for the Zeeman strength to realize Chern insulator counting the gapless feature of surface states as long as Eq. (100) is satisfied.

We have seen that for the topological insulator based Chern insulator, there are always one trivially gapped Dirac cone and one with unit Chern number, and a natural question emerges as which cone is non-trivial? In the symmetric case, gaps of two Dirac fermions are the same, and we have to rely on $\chi$ labelled mirror symmetry together with magnetization direction to decide which cone is non-trivial. However, for the slightly asymmetric case, a quick answer to the question can be made: the one with smaller gap is. To see why, we can consider the gap equation Eq. (82) which can be rewritten as

$$\Delta_\chi = 2|(-\chi I_S(0)) - |I_A(0)||, \tag{103}$$

we find that for the asymmetric Chern insulator case $-\chi I_S(0) > |I_A(0)| \geq 0$, and it always holds that

$$\Delta_\chi < \Delta_{\bar{\chi}}, \tag{104}$$

then combined with Eq. (101), we arrive at the conclusion that it is always the cone with smaller gap which becomes topologically non-trivial carrying unit Chern number, while the cone with a larger Zeeman gap becomes just trivial.

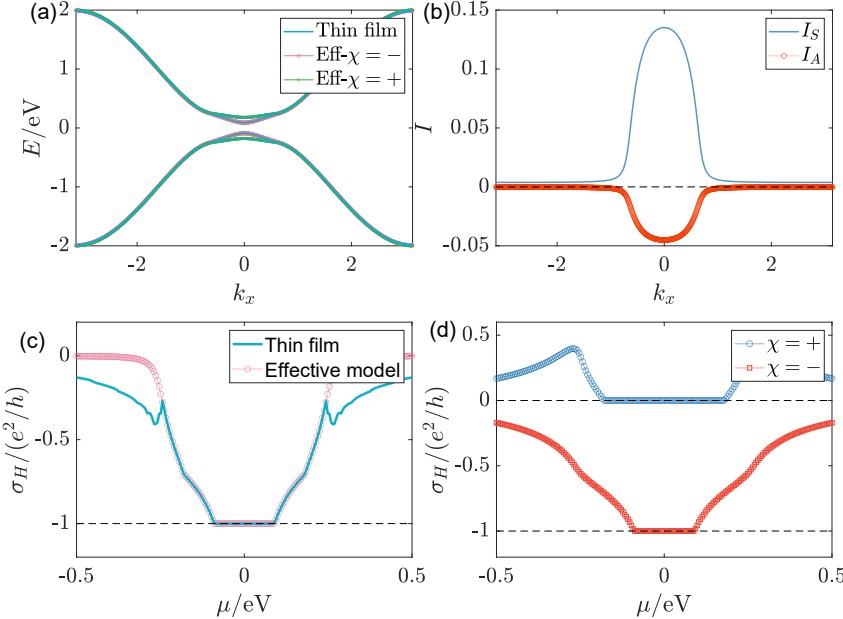

Figure 15: Chern insulator case III: total layer number $L_z = 19$ with top-2-layer Zeeman field $V_z^t = 0.1$ eV and bottom-2-layer field $V_z^b = 0.2$ eV. (a) Comparison of band structure from the lowest four bands of TI film model and effective four-band Hamiltonian. (b) Calculated $I_S(\mathbf{k})$ and $I_A(\mathbf{k})$. (c) Calculated Hall conductivity from TI film model and effective four-band Hamiltonian. (d) Hall conductivity for $\chi = \pm$.

### 5.2.3 Mirror layer Chern number

Notice that there exists a fully mirror symmetric case where $V_A = 0$, and in this special case, a quantity proposed as *mirror layer Chern number* can be defined. Again, the mirror-symmetric Hamiltonian including the Zeeman term can be projected into decoupled mirror-labelled parts as

$$H_\chi = P_\chi^{M_z} H, \qquad P_\chi^{M_z} = \frac{1 + i\chi M_z}{2}, \tag{91}$$

with $M_z$ the represented mirror operator, and its anti-diagonal elements are recognized to be $U_z$, which relates quantity at $\pm l_z$ (see section 5.1 for more about mirror symmetry).

Due to the film geometry, it is natural to introduce the so-called layer Hall conductivity [109–113] by considering layer-dependent eigenstates

$$\sigma_H(l) = \frac{e^2}{h} \frac{\text{Im}}{\pi} \sum_{E_n < \mu < E_m} \sum_{l'} \int d^2k \, \frac{\bar{v}_x^{nm}(l)\bar{v}_y^{mn}(l')}{(E_n - E_m)^2}, \tag{105}$$

where in the usual case, the expectation value of velocity operator is $\bar{v}_i^{mn}(l) = \langle m(l)|\partial_{k_i} H|n(l)\rangle$ with only diagonal elements, which, however, fails for the mirror projected Hamiltonian. The key observation lies in the fact that by projection $\partial_{k_i} H_\chi$ contains not only diagonal elements but off-diagonal part, which induces additional non-local transition contribution from exactly mirror symmetrized layers. Work the effect out and one obtains the mirror layer Hall conductivity

$$\sigma_H^\chi(l) = \frac{e^2}{h} \frac{\text{Im}}{\pi} \left[ \sum_{E_n^\chi < \mu < E_m^\chi} \sum_{l'} \int d^2k \, \frac{\bar{v}_{\chi,k_x}^{nm}(l)\bar{v}_{\chi,k_y}^{mn}(l')}{(E_n^\chi - E_m^\chi)^2} \right], \tag{106}$$

with

$$\bar{v}_{\chi,k_i}^{nm}(l) = \frac{1}{2} \langle n^\chi(l)| \left( v_{k_i}(l)|m^\chi(l)\rangle + U_z v_{k_i}(-l)|m^\chi(-l)\rangle \right), \tag{107}$$

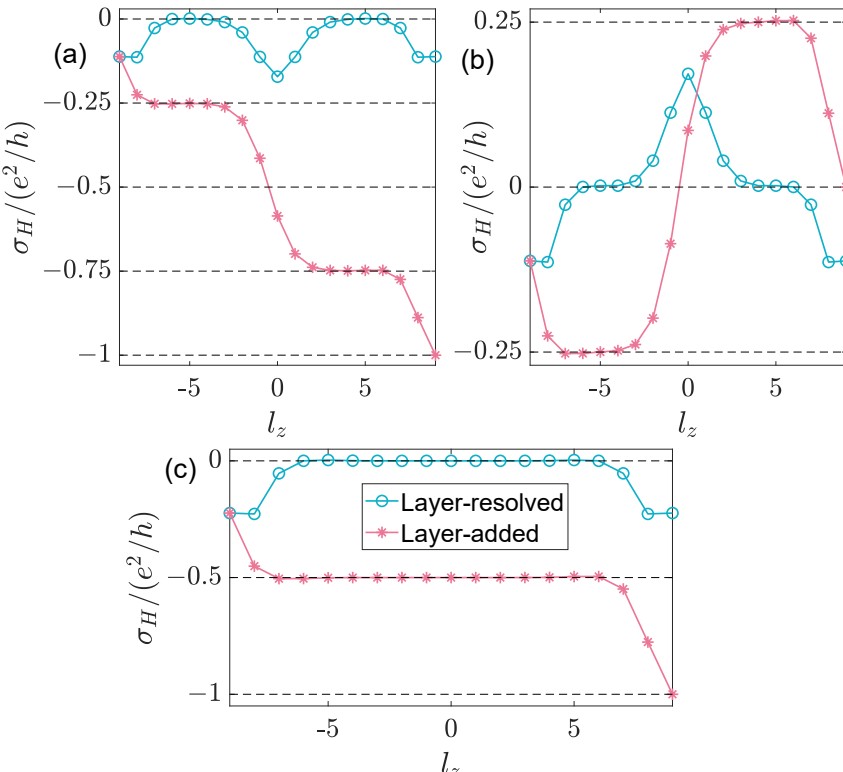

Figure 16: Mirror layer Hall conductivity for topological insulator film with symmetric $z$-Zeeman field immersed at top 2 and bottom 2 layers upon total 19 layers with strength $V_z = 0.1$eV, and chemical potential is chosen to be $\mu = 2.5$meV, where (a) for $\chi = -$, (b) for $\chi = +$ and (c) for the total result by adding two mirror parts together, respectively. Both layer-resolved and layer-added Hall conductivity are presented. To respect the mirror symmetry we put the TI film at origin and the layer index becomes $l_z = -\frac{L_z-1}{2}, -\frac{L_z-3}{2}, \cdots, \frac{L_z-1}{2}$ with total layer number $L_z = 19$.

where the appeared velocity operator is defined through the original Hamiltonian and is assumed to contain only diagonal element $v_{k_i}(l) = (\partial_{k_i} H)(l)$.

Now we turn to our special case. As stated in half quantum mirror Hall effect, the bare Hamiltonian without external field contains mirror symmetry, while the same symmetry constraint imposed on the Zeeman field distribution leads to the restriction that $V_z(l_z) = V_z(-l_z)$, which is equivalent to the requirement that $V_A(l_z) = 0$. Thus, Chern insulator generated by TI film with symmetric Zeeman field owns mirror symmetry, and the corresponding $\sigma_H^\chi(l_z)$ could be carried out, so does its layer-cumulated version $\sigma_{H,c}^\chi(l_z) = \sum_{l=-(L_z-1)/2}^{l_z} \sigma_H^\chi(l)$, as presented in Fig. 16. The anti-diagonal elements of mirror operator read $U_z = -i\sigma_z\tau_z$ for the TI film.

The layer dependent Hall conductivity serves us a new insight to understand the phenomenon. Treating the system as a whole, its layer-resolved Hall conductivity, as presented in Fig. 16(c), becomes non-zero mainly near the top and bottom surfaces where time-reversal symmetry is broken explicitly under the Zeeman field. And the cumulated Hall conductivity gains approximately half quantum Hall conductivity near two surfaces. On the other hand, as shown in Fig. 16(a), (b), when we split the system by mirror symmetry, the layer-resolved mirror Hall conductivity shows similar top and bottom distribution as the whole system, but with only half the amplitude by mirror splitting, while the Hall conductivity distribution around mirror plane shows opposite-sign peaks inherited from the time-reversal unbroken bulk property like that in the half quantum mirror Hall effect. Once the Hall conductivity contribution

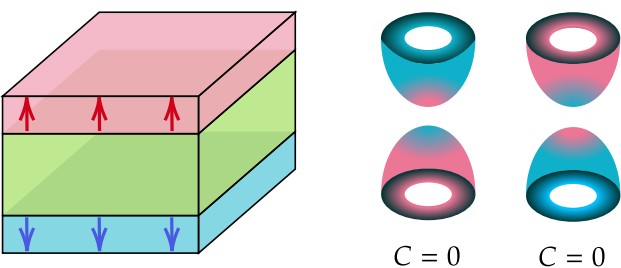

Top & Bottom Anti-Symmetric

Figure 17: Schematic diagram of the axion insulator. On the left, the magnetic heterostructure of the TI film is presented, with top and bottom surface magnetism containing opposite polarization components vertical to the film. On the right, a pair of trivially gapped Dirac cones is presented, both with zero Chern number. The gap comes from the gapped surface states.

is added layer by layer, we immediately see the tri-section configuration: for the non-trivial $C_- = 1$ part, there exist two Hall-plateaus separating the surface and bulk, then following the top-middle-bottom section cut, we see a contribution rather close to $(-1/4)$–$(-1/2)$–$(-1/4)$ from each section; and for the trivial $C_+ = 0$ part, the section separation is not that apparent, and we only roughly write $(-c/4)$–$(c/2)$–$(-c/4)$ with $c$ approximately one to represent the observed distribution.

## 5.3 Axion insulator: An antisymmetric magnetic structure

Along with the special (3+1)-D space-time dimension, the Maxwell electrodynamics is allowed to be decorated with an extra $\theta$ term, which generates the axion electrodynamics [114, 115] to the space-time dependent $\theta$ axion field that couples with the ordinary electromagnetic field. On a practical level, based on the picture of surface Hall effect [61, 116] and analogical mathematical structure between Hall current and magnetization current, people generalize and propose the topological field theory [50], where a $\theta$ term is introduced to describe the magnetoelectric effect [109–111, 117–122] in a topological insulator medium, where the axion field is forced to gain a magnitude of $\pi$ [123] by symmetry and topological requirement.

Realistically, an anti-ferromagnetic TI represents an example of the axion insulator [109]. The axion field, proportional to the space-time volume integral field product $\boldsymbol{E} \cdot \boldsymbol{B}$ or equivalently the Chern-Simons form [50], is odd under time reversal/inversion. In a system with such symmetry, the $\theta$ field matters only for its absolute value and is defined only modulo $2\pi$, which is essential for its $\pi$ magnitude [81]. The anti-ferromagnetic TI certainly breaks these two symmetries, however, as a 3D system, its $\theta$ quantization is protected by an effective time-reversal symmetry as a combination of time reversal and translation [124].

The magnetic configuration in TI film closest to the proposed axion insulator is the one in Fig. 17, which shows a zero-Hall plateau and accompanied non-vanishing longitudinal conductance as an experimental signature [51, 107, 125], also in Cr, V doped $(\text{Bi, Sb})_2\text{Te}_3$ and $\text{MnBi}_2\text{Te}_4$ systems with an even layer number. Here then, based on the effective mass picture, we show that the two Dirac cones with gapped surface states are both trivial, once high-energy parts are involved. Now the fully antisymmetric magnetic configuration leads to $I_S = 0$ for all $k$, and the only left Zeeman quantity is $I_A$, as shown in Fig. 18(b). Then upon weak Zeeman approximation, the two effective masses become, according to Eq. (81),

$$\tilde{m}_\chi(\boldsymbol{k}) = \chi \sqrt{m^2(\boldsymbol{k}) + I_A^2(\boldsymbol{k})}, \tag{108}$$

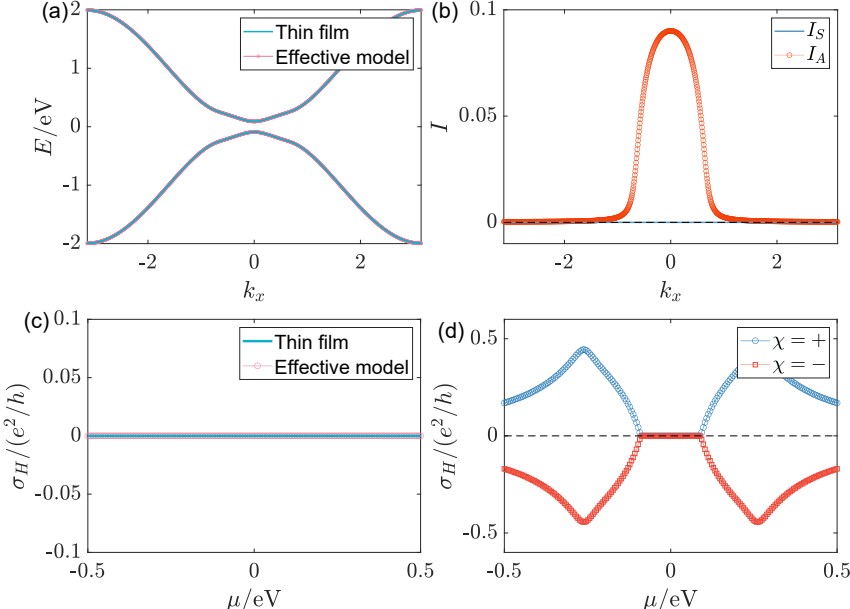

Figure 18: Axion insulator: total layer number $L_z = 19$ with top-2-layer Zeeman field $V_z^t = 0.1$ eV and bottom-2-layer field $V_z^b = -0.1$ eV. (a) Comparison of band structure from the lowest four bands of TI film model and effective four-band Hamiltonian. (b) Calculated $I_S(\boldsymbol{k})$ and $I_A(\boldsymbol{k})$. (c) Calculated Hall conductivity from TI film model and effective four-band Hamiltonian. (d) Hall conductivity for $\chi = \pm$.

which do not show sign reversal in whole Brillouin zone for both $\chi$ and are thus trivial. Numerical results for the Hall conductivities related to two masses are shown in Fig. 18 (d), where they cancel each other exactly at any chemical potential. Especially the zero-plateaus for both $\chi$ bands, which correspond to the situation with the chemical potential lying inside the Zeeman gap, reveal that both bands are trivial with zero Chern number.

We can also generalize this case. Generally for the axion insulator we need $|I_S(0)| < |I_A(0)|$, i.e., asymmetric Zeeman distribution overwhelms symmetric configuration at surfaces, then from Eq. (84) we have

$$C_+ = C_- = 0\,, \tag{109}$$

which in fact leads to a trivially insulating phase viewed from the effective 2D model. The phase is termed as the axion insulator (AI) phase, since the totally asymmetric magnetic polarization leads to, if one switches a surface-state representation, a sign difference of low-energy mass of top and bottom surface states, which gives rise to non-vanishing Berry curvature at low-energy thus surface Hall contribution, with opposite sign for two surfaces. However, the Chern number as we have shown for each complete surface band is zero, which reveals an overall cancellation of transverse transport signals to the linear order, and the Hall conductivity contributed from the gapped surface states is not protected to be half-quantized. Furthermore, counting on the zero Chern number nature for each individual band, the absence of chiral edge state for an $x$-$y$ opened TI film stands firmly, and the non-vanishing longitudinal conductance measured has to be induced by the side-surface states of a topological insulator, and the signal becomes non-zero only when the chemical potential is fine-tuned to avoid falling in the finite-size gap $\sim \lambda_\parallel/L_z$ of the side surface.

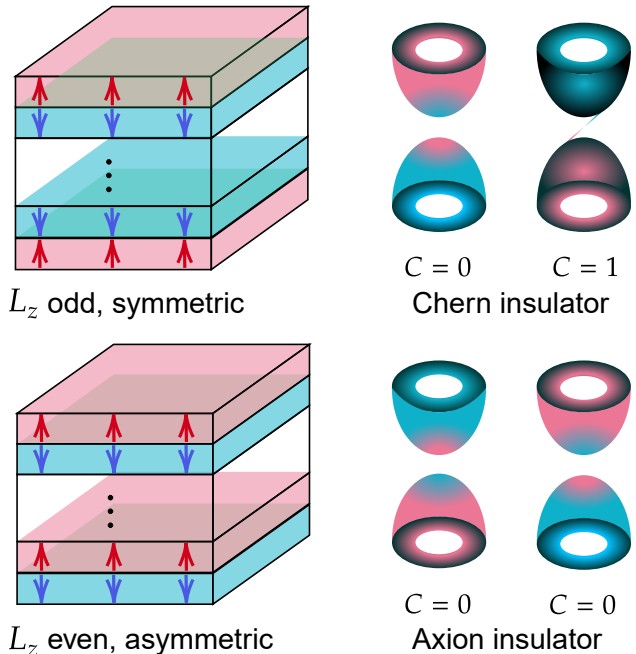

Figure 19: Schematic diagram of the anti-ferromagnetic topological insulator films MnBi$_2$Te$_4$ with the magnetic moments along the $z$ axis. Up: Odd layer number film with net ferromagnetism and symmetric Zeeman distribution, which corresponds to a non-trivial Chern insulator; Down: Even layer number film without net ferromagnetism and antisymmetric Zeeman distribution, which corresponds to the axion insulator with two trivially gapped Dirac cones.

## 5.4 MnBi$_2$Te$_4$ film: Even and odd number of magnetic layers

The first intrinsic antiferromagnetic topological insulator [109], MnBi$_2$Te$_4$ (Te-Bi-Te-Mn-Te-Bi-Te) [126–128], is composed of septuple layers (SLs), with out-of-plane intralayer ferromagnetism and interlayer anti-ferromagnetism, known as the A-type AFM state. It is predicted and shown that with odd or even SL layer numbers, the material will exhibit quantum anomalous Hall effect [106, 107, 129–131] or the axion insulating phase [107, 132], respectively. Here, based on the lowest four-band model and the discussed Chern and axion insulator pictures, we can explain these two phenomena in a simple and elegant way.

The combination of layer-number-odevity determined (anti-)symmetric Zeeman distribution and the localized nature of surface states leads to two qualitatively distinct physical pictures. As revealed in the schematic diagram Fig. 19, when the layer number $L_z$ is odd, the Zeeman distribution is symmetric with parallel polarization of the outermost top and bottom Zeeman field direction, and vice versa. Based on the symmetry analysis, two cases are identified.

### 5.4.1 Odd layer: Chern insulator

In this case

$$\begin{cases} I_S > 0\,, \\ I_A = 0\,, \end{cases} \qquad L_z \mod 2 = 1\,, \tag{110}$$

with the maximum value of $I_S$ centralized around $\Gamma$ as shown in Fig. 20(a), and its sign is controlled by the outermost layer Zeeman field direction, given by the fact that the low energy states around $\Gamma$ are localized near two surfaces. $I_S$ almost vanishes for large $k$ since the high

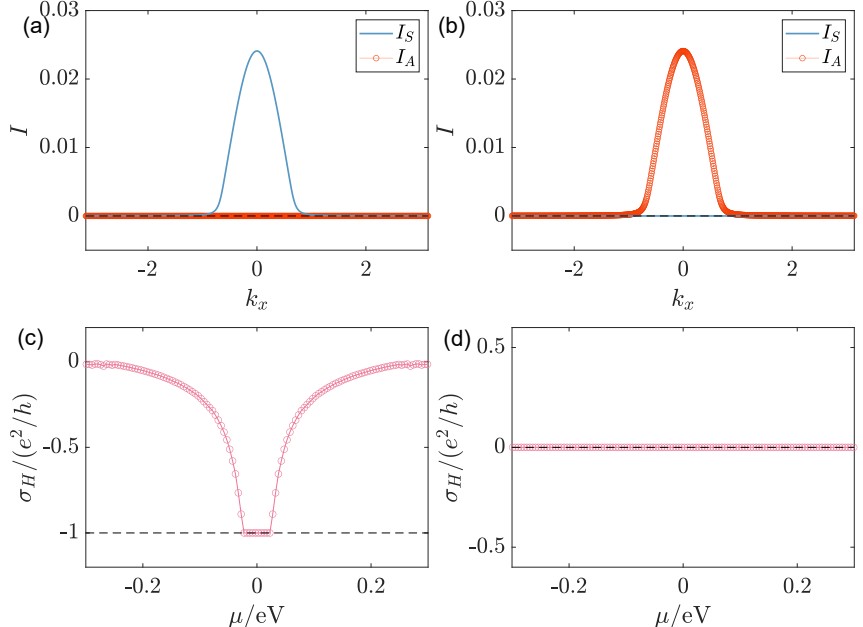

Figure 20: Left (right) pictures are for $L_z = 19$ (18) anti-ferromagnetic TI film as an odd (even) one. The Zeeman strength is chosen to be $|V_z| = 0.1$ eV. (a) (b) Calculated $I_{S/A}(\boldsymbol{k})$ for the effective model. (c) (d) Calculated Hall conductance from magnetic TI film Hamiltonian.

energy states emerge into bulk and distribute diffusely, which leads to the cancellation of $I_S$ integral counting on the interlayer antiferromagnetism. Discussion above classifies the odd SL MnBi$_2$Te$_4$ films into Chern insulator phase, as now $\tilde{m}_\chi = I_S + \chi|m|$ following Eq. (97), with $\mathrm{sgn}(\tilde{m}_\chi(\Gamma)) = \mathrm{sgn}(I_S) > 0$, $\mathrm{sgn}(\tilde{m}_\chi(M)) = \chi$, and $\tilde{m}_-$ changes signs at $\Gamma$ and $M$ which gives rise to a unit Chern number, while $\tilde{m}_+$ is trivially gapped. Totally, the odd-layer MnBi$_2$Te$_4$ stands as a Chern insulator with unit Hall plateau, as revealed in Fig. 20(c), where the relatively narrow quantized Hall plateau for the quantum anomalous Hall insulator phase is due to the second-outermost-layer Zeeman field which owns an inverse polarization direction compared with the outermost field by the interlayer anti-ferromagnetic nature, and thus weakens the $I_S$ integral at the $\Gamma$ point, whose amplitude is recognized as the band gap which measures the width of the quantized plateau when the chemical potential shifts.

### 5.4.2 Even layer: Axion insulator

In this case

$$\begin{cases} I_S = 0, \\ I_A > 0, \end{cases} \qquad L_z \mod 2 = 0, \tag{111}$$

with the maximum value of $I_A$ centralized around $\Gamma$ as shown in Fig. 20(b), which classifies the even SL MnBi$_2$Te$_4$ films into axion insulator phase, as now $\tilde{m}_\chi = \chi\sqrt{m^2 + I_A^2}$ following Eq. (108), with $\mathrm{sgn}(\tilde{m}_\chi(\Gamma)) = \mathrm{sgn}(\tilde{m}_\chi(M)) = \chi$, and both become trivial since they do not change signs. Totally, the even-layer MnBi$_2$Te$_4$ shares zero Hall plateau revealed in Fig. 20(d).

### 5.5 Half-quantized anomalous Hall effect: A semi-magnetic film

From a model point of view, there should exist a search for the phase characterized by a domain-wall separating the axion insulator $(|I_A(0)| > |I_S(0)|)$ and Chern insulator $(|I_A(0)| < |I_S(0)|)$, and that comes to the celebrated half-quantized anomalous Hall phase

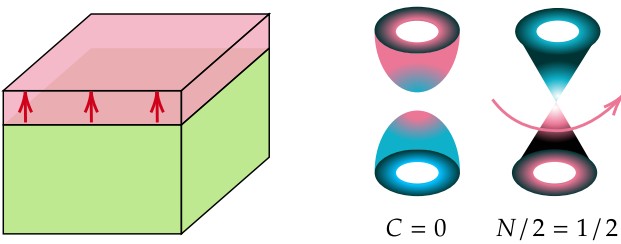

Magnetism at One-Surface Only

Figure 21: Schematic diagram of the half-quantized anomalous Hall effect. In this case only one side of the TI film is immersed with magnetism. The topological property is revealed by one trivially gapped Dirac cone and a gapless Dirac fermion that carries half-quantized Hall conductivity.

[31,52,53] with condition $|I_S| = |I_A|$ inside the parity-invariant regime. Configurationally, this corresponds to a semi-magnetic TI with a Zeeman field applied on only one side, as illustrated in Fig. 21. The corresponding numerical results are presented in Fig. 22.

Another motivation for searching such a phase lies deeply in the lattice realization of a single Dirac fermion, which serves as a basis for the lattice gauge theory [133, 134]. The Nielsen-Ninomiya theorem [29, 30], however, imposes strong constraints on this realization. Tremendous approaches have been proposed like the Wilson fermion [4,34], the SLAC fermion [35, 135, 136], the Tan fermion [137, 138], etc. These realizations either break one or more conditions required by the fermion-doubling theorem, such as symmetry or locality, or evade the physical requirements like existence of first order derivative of wavefunction and finite bandwidth on lattice.

In this context, by introducing magnetism to gap out surface states of one Dirac cone through magnetism, the remaining gapless Dirac cone, as depicted in Fig. 22(a), essentially serves as one lattice realization of a single Dirac fermion. As stated, the gapless Dirac cone on lattice has to boil one or more conditions required by the fermion-doubling problem, and it is the 2D parity symmetry together with the locality that are broken. To avoid doubling caused by periodicity of Brillouin zone, the mass term of this gapless Dirac fermion has to contain non-vanishing bulk-like high-energy part, as captured by Eq. (11), which breaks the parity symmetry explicitly, while the vanishing low energy mass preserves the symmetry. Such a low-energy symmetry-preserving while high-energy symmetry-breaking term shares similarity with the 'quantum anomaly' [47, 63–68] in field theory, specifically the parity anomaly in this case. However, the gapless Dirac fermion appeared here manifests itself as a regularized complete condensed matter system with explicit symmetry breaking at high-energy, which should be distinguished from the spontaneous symmetry breaking case under the frame of quantum anomaly. The locality principle is violated by the massless to massive transition.

The gapless Dirac fermion, identified as the band with gapless surface states contributes a half-quantized Hall conductance. From Fig. 22 (d), the $\chi = +$ band is trivial with zero-Hall plateau inside the Zeeman gap, i.e., the Zeeman gapped band is trivial, while the $\chi = -$ band contains a relatively large Hall plateau quantized to $-e^2/2h$, which is bounded by the TI bulk gap and corresponds to the Hall conductance contributed from the high-energy part of the gapless Dirac band [31]. To explain this behavior, it is important to note that we now have $I \equiv I_S = I_A > 0$ around the Dirac point revealed in Fig. 22 (b) (valid in the parity invariant regime bounded by $k_c$), and the effective masses become, according to Eq. (81),

$$\tilde{m}_\chi = I(\boldsymbol{k}) + \chi \sqrt{m^2(\boldsymbol{k}) + I^2(\boldsymbol{k})}, \tag{112}$$

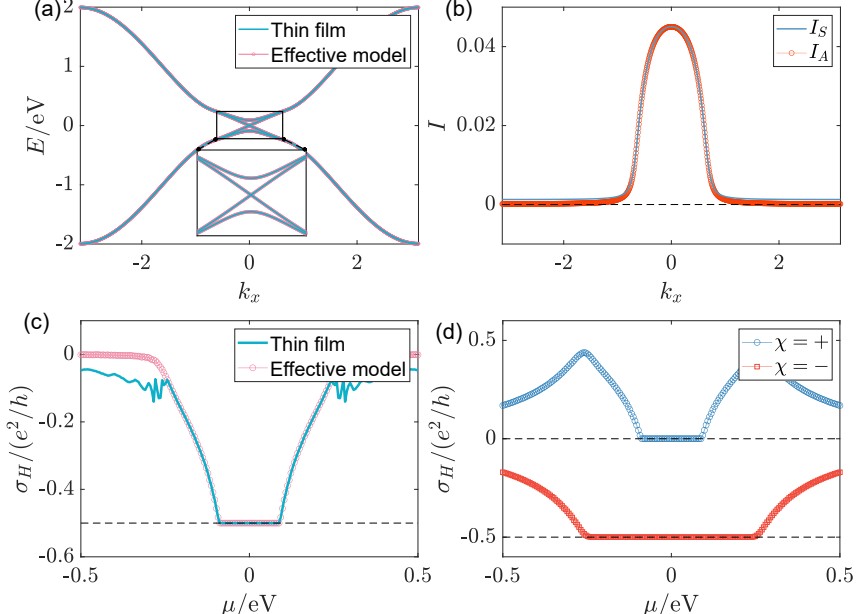

Figure 22: Half-quantized anomalous Hall metal: total layer number $L_z = 19$ with top-2-layer Zeeman field $V_z^t = 0.1$ eV. (a) Comparison of band structure from the lowest four bands of TI film model and effective four-band Hamiltonian. (b) Calculated $I_S(\boldsymbol{k})$ and $I_A(\boldsymbol{k})$. (c) Calculated Hall conductance from TI film model and effective four-band Hamiltonian. (d) Hall conductance for $\chi = \pm$.

from which we see that $\tilde{m}_+ > 0$ holds for any $k$ and is trivial, while

$$\tilde{m}_- = \begin{cases} 0, & k < k_c, \\ I - \sqrt{m^2 + I^2} \sim -|m(\boldsymbol{k})|, & k > k_c, \end{cases} \tag{113}$$

which is nontrivial and offers us with a half-quantized Hall conductance within the regime $k < k_c$, as read from Eq. (38).

To realize this phase generally, we need $|I_S(k)| = |I_A(k)|$ when $k < k_c$. Under the situation, one specifies the $\chi$ such satisfying that

$$-\chi I_S(k < k_c) = |I_A(k < k_c)|, \tag{114}$$

which gives the gaps according to Eq. (82) that $\Delta_\chi = 0$ while $\Delta_{\bar{\chi}} = 4|I_A(0)|$, i.e., one gapless band plus one gapped band. For the gapped band, the Chern number description still works and gives

$$C_{\bar{\chi}} = -\bar{\chi}\Theta(-2|I_A(0)|) = 0, \tag{115}$$

while for the gapless band, we can not use Chern number to define its topology in principle, since it describes a metallic phase with a non-vanishing Fermi surface. Nevertheless, the effective masses now have property

$$\begin{cases} \tilde{m}_\chi(k < k_c) = 0, \\ \tilde{m}_{\bar{\chi}}(k < k_c) = 2\bar{\chi}|I_A(k)|, \end{cases} \tag{116}$$

then combined with the high-energy condition $\tilde{m}_\chi(\pi, \pi) \sim \chi|m(\pi, \pi)|$, one obtains that

$$\begin{cases} \sigma_H^\chi = \dfrac{\chi}{2}\dfrac{e^2}{h}, & |\mu| < 2|I_A(0)|, \\ \sigma_H^{\bar{\chi}} = 0, \end{cases} \tag{117}$$

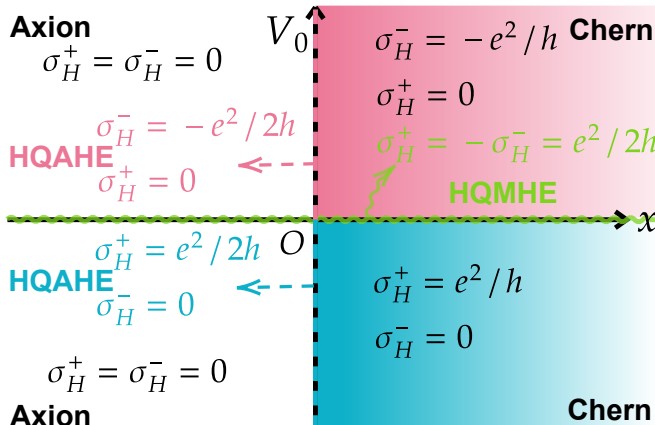

Figure 23: Phase diagram of topological phases with weak field. Four distinct phases have been labelled as Chern insulator phase in the first and fourth quadrants differed by sign of Hall conductance, Axion insulator phase in the second and third quadrants, the half quantum mirror Hall effect (HQMHE) along the $x$-axis (indicated by the green wave line), and the half-quantized anomalous Hall effect (HQAHE) along $V_0^+$ and $V_0^-$ rays (indicated by red or blue dashed lines) differed by sign of Hall conductance. The effectiveness of the phase diagram should be confirmed for chemical potential lying in both the parity invariant regime and (smaller) Zeeman gap of surface states, and the Zeeman strength should be constrained to be relatively weak compared with the bulk gap, while playing its role mainly at top and bottom surfaces under the discussed frame.

in line with Eq. (38), i.e., the gapless Dirac cone provides half-quantized Hall conductance, accompanied by a trivially gapped cone. This phenomenon is known as the half-quantized anomalous Hall effect [31, 52, 53], and is experimentally observed in Cr-doped $(Bi, Sb)_2Te_3$ system. It is important to note that the chemical potential should lie within the magnetic gap to avoid non-quantized contributions from the trivial $\bar{\chi}$ band. Additionally, the weak Zeeman field presumption ensures that the Zeeman gap, which is smaller than the bulk gap, does not exceed the energy limit of the parity-invariant regime. The metallic nature of the non-trivial gapless Dirac fermion indicates that the system stays inside a metallic topological phase. Notice that the non-trivial gapless band requirement Eq. (114) gives $\chi = -\mathrm{sgn}(I) = -\mathrm{sgn}(V)$ with $I = I_S(0)$ and $V = V_S^{\mathrm{top}}$, and we can write down the asymptotic Hamiltonian for this band as

$$H_{\mathrm{half}} \sim \lambda_{\parallel}(\sin(k_x a)\sigma_x + \sin(k_y b)\sigma_y) + \mathrm{sgn}(V)m(\boldsymbol{k})\sigma_z, \qquad (118)$$

counting on the fact that $m(\boldsymbol{k}) \leq 0$. This effective Hamiltonian offers with half-quantized Hall conductance $-\mathrm{sgn}(V)e^2/2h$, which does not depend on whether the magnetism is put at the top or bottom of TI film, but only on its polarization direction. Under an external magnetic field, such a single gapless Dirac fermion will step into the quantum Hall regime [52, 90] and exhibits quantized Hall conductance whenever an integer number of Landau levels become fully filled [139]. Especially, the 'anomaly' contribution will manifest itself to compensate the half quantization contributed from the lowest Landau level, so as to keep the integer value of Chern invariant for this gapped Landau level system.

## 5.6 Phase diagram

To appreciate the details of the phases mentioned, especially regarding the phase transitions among, we go back to the effective model and assume that the immersed depth of top and

bottom Zeeman field, if exists, is relatively longer than the characteristic exponentially decaying length of surface states while being much smaller than the film thickness, with uniform strength for the top or bottom field. Then we can adopt the substitution

$$I_{S/A} \to V_{S/A} = V_{S/A}^{\text{top}}.$$ (119)

And the effective model reads

$$\begin{cases} \tilde{H}_\chi = \lambda_\| (\sin(k_x a)\sigma_x + \sin(k_y a)\sigma_y) + \tilde{m}_\chi(\boldsymbol{k})\sigma_z, \\ \tilde{m}_\chi(\boldsymbol{k}) = V_S + \chi \sqrt{m^2(\boldsymbol{k}) + V_A^2}, \\ m(\boldsymbol{k}) = \Theta(-m_0(\boldsymbol{k}))m_0(\boldsymbol{k}), \\ m_0(\boldsymbol{k}) = m_0 - 4t_\| \left( \sin^2 \frac{k_x a}{2} + \sin^2 \frac{k_y b}{2} \right), \end{cases}$$ (120)

from which one reads the Hall conductance from Eq. (83) as (in the Zeeman gap or the parity invariant regime)

$$\sigma_H^\chi = \frac{e^2}{h} \frac{1}{2} \left[ \chi - \text{sgn}(V_S + \chi |V_A|) \right].$$ (121)

Now let us introduce the top Zeeman strength $V_z^{\text{top}} = V_0$, and the bottom Zeeman strength $V_z^{\text{bottom}} = x V_0$ described by the collaboration between $V_z^{\text{top}}$ and a phenomenological parameter $x$ characterizing their relative strength. Then accordingly we have

$$\begin{cases} V_S = V_0 \dfrac{1+x}{2}, \\ V_A = V_0 \dfrac{1-x}{2}, \end{cases}$$ (122)

which gives further the Hall conductance

$$\sigma_H^\chi = \frac{e^2}{h} \frac{1}{2} \left[ \chi - \text{sgn}(V_0)\text{sgn}\left( \frac{1+x}{2} + \chi \, \text{sgn}(V_0) \left| \frac{1-x}{2} \right| \right) \right],$$ (123)

whose dependence on parameters $(x, V_0)$ are presented in Fig. 23 as a phase diagram emphasizing the role the relative strength $x$ plays here. Notice that we have defined $\text{sgn}(0) = 0$ here, corresponding to realistic physical phenomenon when $V_0 = 0$. From the diagram, except for $V_0 = 0$ line, which represents a pure topological insulator film with half quantum mirror Hall effect, it is always $x \geq 0$ side that gives rise to phases with non-vanishing Hall conductance, belonging to either Chern insulator or half-quantized anomalous Hall metal phase, while the $x < 0$ side termed as axion insulator phase always contains two trivially gapped Dirac cones/fermions.

Focusing on the phase transition, we observe that a phase characterized by an anomalous half-quantized index always emerges upon the integer index phase transition. This phenomenon echoes transitions observed in integer quantum Hall systems [100, 140], where the renormalization group flow diagram exhibits a generic fixed point with half-quantized Hall conductance and finite longitudinal conductance, suggesting a phase transition in 2D from a field theoretical point of view. However, the physics here should differ, as the robustness of the gapless surface state is protected by the bulk and corresponding surface time-reversal symmetry as an intrinsic feature of 3D strong topological insulators [17]. Put the statement differently, the additional dimension in our system exhibits robust topological/geometric effects, making it plausible that phases characterized by half-integers here are more likely to be symmetry-protected metallic topological phases, while this protection only occurs in a finite regime over the whole Brillouin zone. Especially, the half QAHE here is protected by a parity invariant regime, and is different from a critical quantum Hall transition phase without protection from any non-conformal symmetries.

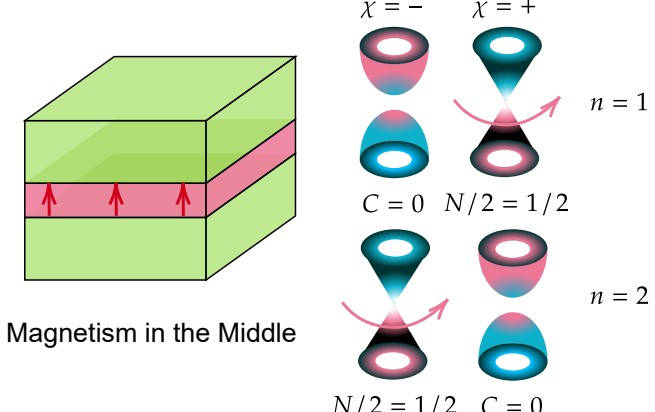

Figure 24: Schematic diagram of the metallic quantized anomalous Hall effect. In the case a relatively strong out-of-plane ordered magnetism exists in the middle of the film. The topological property of the system is reflected by a pair of gapless Dirac cones with the same high-energy mass sign, each carrying half-quantized Hall conductivity.

In the phase diagram we draw, the line of half quantum mirror Hall effect is crossed when transitioning between two Chern insulator phases characterized by opposite Chern numbers, since such a phase transition relies on changing of Zeeman polarization direction, thus crossing $V_0 = 0$ where half quantum mirror Hall effect happens. A similar thing happens for the transition between axion insulator phases differed by Zeeman direction. On the other hand, lines representing half QAHE are crossed when stepping between the Chern insulator and axion insulator phases, with the sign of Hall conductance determined by Zeeman direction.

# 6 Topological phases with strong field

A more extensive and complex regime exists beyond the weak Zeeman field approximation, and the criterion tells that the topological phase appearing here can not be simply described under $n = 1$ framework. In this scenario, we step into the strong field regime, where the appearance of $n \geq 2$ cones is unavoidable. Surprisingly, the inter-Dirac-cone interaction can sometimes play the ultimate role deciding the topological property of the system. It is in such situations that our effective mass picture from Eq. (66) and Eq. (67) serves as the ultimate criterion for the topological property in the system.

## 6.1 Metallic quantized anomalous Hall effect: A film with a magnetic sandwich structure

One other novel metallic topological phase bearing a pair of gapless Dirac fermions has been recently proposed [32], which shows a quantized Hall conductivity of unit that originates from two metallic bands, each with one-half quantum. To further enhance our understanding of magnetic topological phases, the key findings related to this phase are summarized below.

The schematic diagram is shown in Fig. 24. We set total layer number $L_z = 22$ which is even, and the $z$-symmetric site positions read

$$l_z = \pm\frac{1}{2}, \dots, \pm\frac{L_z - 1}{2}. \tag{124}$$

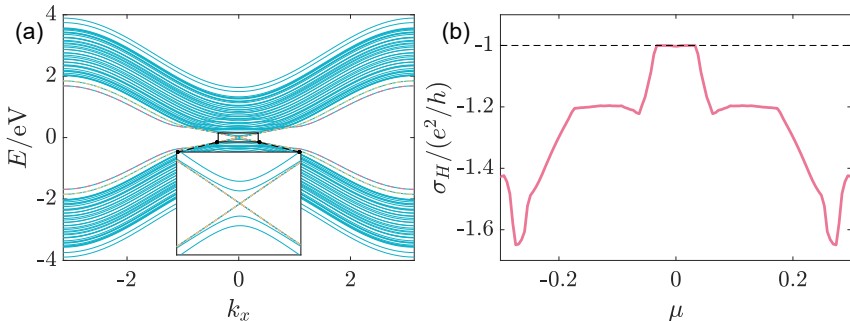

Figure 25: (a) The band structure near the $\Gamma$ point with $k_y = 0$ with the presence of magnetic doping ($\alpha = 0.9$). The gapless dispersions for the surface states are doubly degenerate, as shown by the red and yellow lines. (b) Corresponding Hall conductivity as a function of the chemical potential $\mu$ at $\alpha = 0.9$. The thickness $L_z = 22$ and the magnetic layers $m_z = 6$.

Accordingly, $z$-symmetric Zeeman field in magnetically doped layers at the middle of the TI film is set as

$$V_z(l_z) = \begin{cases} \alpha t_\perp, & l_z = \pm 1/2, \ldots, \pm(m_z - 1)/2, \\ 0, & \text{otherwise}, \end{cases} \tag{125}$$

with magnetic layer number $m_z = 6$. By $z$-symmetric $V_S(l_z) = V_z(l_z)$, $V_A(l_z) = 0$, the projection only contains $\mathbf{I}_S$ term proportional to $\alpha$. Then we bring $\alpha$ to the front explicitly as

$$\mathbf{I}_S(\alpha, \mathbf{k})\tau_0\sigma_z \mapsto \alpha\mathbf{I}_S(\mathbf{k})\tau_0\sigma_z, \tag{126}$$

with $\mathbf{I}_S(\alpha = 1, \mathbf{k}) \mapsto \mathbf{I}_S(\mathbf{k})$ as a re-definition.

The metallic feature and quantized Hall conductivity nature are revealed in Fig. 25. The band structure of the film is shown in the presence of strong enough magnetism ($\alpha = 0.9$), with a pair of massless Dirac fermions. The pairing nature is reflected by the double degeneracy of band dispersion near the $\Gamma$ point, as labelled by the red and yellow lines. The unbroken surface states picture is possible due to the localized nature of the surface states inside the bulk-gap, which is not affected by the far-away magnetism in the middle of the film. Meanwhile, a quantized Hall conductivity is observed, when the chemical potential lies inside both the bulk and magnetic gap. And as we shall see later, essentially the quantization comes from the two gapless Dirac fermions, each sharing a half-quantized Hall conductivity with the same sign, based on which we further identify that the effect is not only superficially metallic, but originates from such metallic bands. And it is in this circumstance that we term this new phase as the 'metallic quantized anomalous Hall effect' (metallic QAHE), indicating that it differs from the conventional QAHE, aka the Chern insulator in an insulating phase.

Attributed to the mass exchange mechanism over the effective mass picture presented in Section 4, such a topological phase transition with the increasing of $\alpha$ as Zeeman strength in the middle can be explained. Absorbing the $\alpha$-dependent Zeeman term into the one-dimensional Hamiltonian separated from the TI film leads to an $\alpha$-dependent 1-D Hamiltonian $H_{1d}(\alpha)$, with $H_{1d}(\alpha = 0)$ coming back to the 1-D Hamiltonian extracted from TI film and solved exactly before (see section 2.2). Projecting $H_{1d}(\alpha)$ over solutions of $H_{1d}(0)$ leads to $\left(\bigoplus_{n=1}^{L_z} m_n \tau_z + \alpha\mathbf{I}_S(\mathbf{k})\tau_0\right)\sigma_z$, and further diagonalizing this provides a bijection which maps the projected Hamiltonian form into the mass term $\bigoplus_{n=1,\chi=\pm}^{L_z} \tilde{m}_{n,\chi}(\mathbf{k}, \alpha)\sigma_z$ (see section 4.1). Notice that both $\sigma_z$ and $\tau_z$ here are good quantum numbers, as spin and mirror indices ($\chi = \pm$), respectively. Confining to the subspace with $\sigma_z = +$, we could then track the evolution and interaction of the mass terms $\tilde{m}_{n,\chi}$ between $n = 1$ and $n = 2$ blocks with increasing

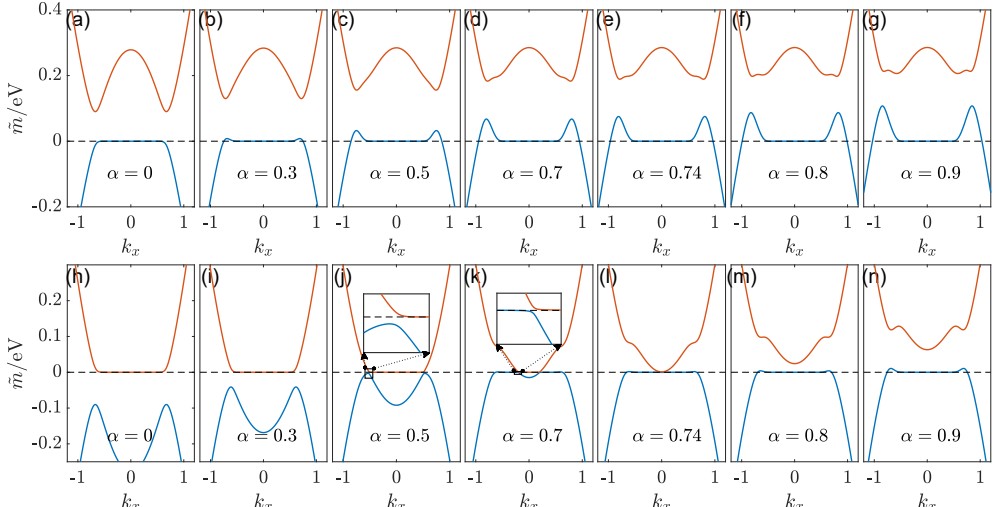

Figure 26: The evolution of the effective mass $\tilde{m}_{n,\chi}(k_x, k_y = 0)$ ($n = 1, 2$). (a)∼(g) and (h)∼(n) show lowest-two **effective masses** varying with changing Zeeman field strength $\alpha$ belonging to set $[0, 0.3, 0.5, 0.7, 0.74, 0.8, 0.9]$ for $\chi = +$ and $\chi = -$, respectively. (a) (g) (h) (n) have already been shown as Fig. 3 in the main text but with a finer structure here. Adapted from [32].

$\alpha$ for given $\chi$. As shown in Fig. 26, $\tilde{m}_{n,+}(n = 1, 2)$ maintain their shapes increasing $\alpha$, while $\tilde{m}_{n,-}(n = 1, 2)$ have effectively exchanged their high-energy parts through the low-energy mass exchange, which leads to the high-energy mass sign change of the gapless Dirac cone, and alters its Hall conductivity from $e^2/2h$ to $-e^2/2h$, when Fermi surface lies inside the parity invariant regime. Then combined with the unaltered $-e^2/2h$ from $\tilde{m}_{1,+}$, totally a topological phase transition happens, driving the system from zero Hall conductivity to quantized Hall conductivity, with Hall contribution coming from two metallic bands, which makes the system a metallic topological phase. We can identify

$$\alpha_c \approx 0.74, \tag{127}$$

in this case to indicate $0 \to -1$ plateau transition. Notice that although we have explicitly exploited the $z$-mirror symmetry to separate our effective masses into two groups, this symmetry consideration is not necessary here and the metallic QAHE is not protected by the symmetry. For example, from Eq. (66), Eq. (67) we see clearly that a general Zeeman field configuration can still generate $2L_z$ independent Dirac masses, and if we place a strong enough Zeeman field in the middle of the film deviating from the symmetric case, still we can see the effect with unit Hall plateau.

The key difference between our metallic QAHE and the conventional QAHE or equivalently the Chern insulator lies in the unconventional bulk boundary correspondence. As discussed in [31], the half-quantized Hall conductivity bears no chiral edge states, while its corresponding boundary physics lies in the existence of the chiral current, which is indeed a bulk states contribution and decays algebraically along the metallic surface, starting from the middle magnetic zone where time-reversal symmetry is broken most severely.

### 6.1.1 A qualitatively model with $n = 1, 2$

A qualitative understanding of the phenomenon within a cut-off approximation based on the $n = 1, 2$ blocks can be deduced. In the mass exchange picture above, we have used the fully diagonalized $\tilde{m}_{1,2}$ to illustrate the physics behind, while the picture with only $n = 1$ involved

based on the weak Zeeman field approximation breaks down. This is essentially because, the weak field approximation heavily relies on effect the magnetism has upon the surface states, which is not the case here since the magnetism in the middle will not directly affect the surface states, and were there to be any physics effects, they must be conducted through the bulk states, whose wavefunction has maximal overlap with the magnetic areas. Here, the metallic QAHE is just the first non-trivial case of such kind, where the inter-$n$ blocks interaction conducted through magnetism is deterministic, and luckily, we have found a way to directly observe the overall effect by a second diagonalization, yielding the effective masses $\tilde{m}_n$. While the process and the results are straightforward and conclusive, it will be more satisfying if a simplified model exists and grasps the core of physics even qualitatively. Interestingly, a model incorporating the $n = 1, 2$ blocks plays a crucial role in achieving this.

For simplicity, we consider the symmetric Zeeman field in the middle, and by preserving $n = 1, 2$, the mass terms read

$$M(\alpha) = \begin{pmatrix} m_1 & \\ & m_2 \end{pmatrix} \tau_z + \alpha \begin{pmatrix} I_S^{11} & I_S^{12} \\ I_S^{21} & I_S^{22} \end{pmatrix} \tau_0, \tag{128}$$

with $k$-dependence in $m_n$ and $I_S$ terms. The Hamiltonian for $n = 1, 2$ reads

$$H^{n=1,2}(\boldsymbol{k}) = \lambda_\parallel \rho_0 \tau_0 (\sin(k_x a)\sigma_x + \sin(k_y b)\sigma_y) + M\sigma_z, \tag{129}$$

with $\rho$ another pseudo-spin degrees of freedom for two blocks.

Following the effective mass treatment, we further block-diagonalize $H^{1,2}$ into $2 \times 2$ sub-blocks. Notice that again the projected mirror operator $\tau_z$ in $M$ serves as a good quantum number due to the chosen symmetric Zeeman distribution, then a split $M = \oplus_\chi M_\chi (\chi = \pm)$ can be made, so does that for the Hamiltonian $H^{1,2} = \oplus_\chi H^{1,2}_\chi$, where

$$\begin{aligned} H^{1,2}_\chi &= \lambda_\parallel \rho_0 (\sin(k_x a)\sigma_x + \sin(k_y b)\sigma_y) + \alpha \operatorname{Re}(I_S^{12})(\boldsymbol{k})\rho_x\sigma_z - \alpha \operatorname{Im}(I_S^{12})(\boldsymbol{k})\rho_y\sigma_z \\ &\quad + E_\chi(\boldsymbol{k})\rho_0\sigma_z + \Delta_\chi(\boldsymbol{k})\rho_z\sigma_z, \end{aligned} \tag{130}$$

with

$$\begin{cases} E_\chi = [\chi(m_1 + m_2) + \alpha(I_S^{11} + I_S^{22})]/2, \\ \Delta_\chi = [\chi(m_1 - m_2) + \alpha(I_S^{11} - I_S^{22})]/2. \end{cases} \tag{131}$$

Clearly, diagonalization in $\rho$-space is accessible without altering the linear part, which leads to

$$\tilde{H}^{1,2}_{\chi\zeta} = \lambda_\parallel (\sin(k_x a)\sigma_x + \sin(k_y b)\sigma_y) + \tilde{m}_{\chi,\zeta}\sigma_z, \tag{132}$$

where

$$\tilde{m}_{\chi,\zeta} = (E_\chi(\boldsymbol{k}) + \zeta\Lambda_\chi(\boldsymbol{k}))\sigma_z, \quad \chi, \zeta = \pm, \tag{133}$$

with $\Lambda_\chi = \sqrt{\Delta_\chi^2 + \alpha^2 |I_S^{12}|^2}$ defined. This is reached by a unitary transformation $U_\chi = U_2^\chi U_1^\chi$ for each $\chi$, where $U_2^\chi = e^{i\rho_x \theta_2^\chi}$, $U_1^\chi = e^{i\rho_y \theta_1^\chi}$, with definitions $\tan 2\theta_1^\chi = \alpha \operatorname{Re}(I_S^{12})/\Delta_\chi$, $\tan 2\theta_2^\chi = \alpha \operatorname{Im}(I_S^{12})/\delta_\chi$, $\delta_\chi = \sqrt{\alpha^2 \operatorname{Re}(I_S^{12})^2 + \Delta_\chi^2}$.

Now we choose case $\alpha > 0$ so that $\alpha I_S^{nn} > 0$ to illustrate the physics. Topological phase transition happens when $\alpha I_S^{22} > m_2(0) > 0$ (for $m_2(0) > 0$ see Fig. 4) with the help of $I_S^{12}$. In the case now, we identify the Hall conductivity for each sub-block as

$$\sigma_H^{\chi\zeta} = \frac{e^2}{2h}\left[\operatorname{sgn}(\tilde{m}_{\chi,\zeta}(M)) - \operatorname{sgn}(\tilde{m}_{\chi,\zeta}(k_F^{\chi,\zeta}))\right], \tag{134}$$

with $\tilde{m}_{\chi,\zeta}(k_F^{\chi,\zeta})$ recognized as $\tilde{m}_{\chi,\zeta}$ at Fermi surface of the band, and for an insulating band with Fermi level inside the gap, it is $\tilde{m}_{\chi,\zeta}(0)$. For unification and simplicity, we will always

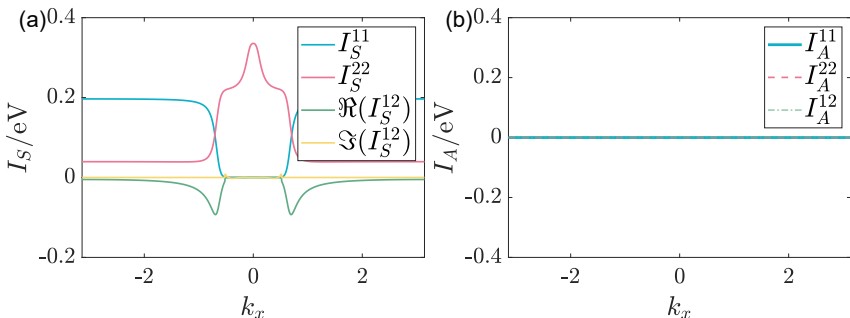

Figure 27: $I_{S/A}^{11}$, $I_{S/A}^{22}$ and $I_{S/A}^{12}$ calculated with total layer number $L_z = 22$ and middle Zeeman layer number $m_z = 6$. Re-plotted from [32].

assume Fermi level to lie inside insulating gap and the parity invariant regime of a gapless band near $\Gamma$ point so to always recognize $k_F = 0$, and those worrying about the singular gapless Dirac point for the metallic case can always take the unambiguous second limit in Section 3.5. Then by treating

$$\begin{cases} m_1(0) = 0, \ m_2(0) > 0, \\ -m_1(M) \approx m_2(M) \gg \alpha |I_S(M)| > 0, \\ I_S^{11}(0) = I_S^{12}(0) = 0, \ I_S^{22}(0) > 0, \\ I_A = 0, \end{cases}$$

where quantities $I_{S/A}$ can be read from Fig. 27, we can write

$$\begin{cases} \tilde{m}_{\chi,\zeta}(0) = \dfrac{\chi m_2(0) + \alpha I_S^{22}(0)}{2} + \zeta \left| \dfrac{\chi m_2(0) + \alpha I_S^{22}(0)}{2} \right|, \\ \tilde{m}_{\chi,\zeta}(M) \approx \zeta m_2(M). \end{cases} \tag{135}$$

Clearly, $\tilde{m}_{\chi,\zeta}(M)$ are almost unchanged since the projected Zeeman field is not that strong here, and the Hall conductivity formula is reduced into

$$\sigma_H^{\chi\zeta} = \frac{e^2}{2h} \left[ \zeta - \text{sgn}(\tilde{m}_{\chi,\zeta}(0)) \right]. \tag{136}$$

For $\tilde{m}_{\chi,\zeta}(0)$ two cases should be distinguished. When $\alpha I_S^{22}(0) < m_2(0)$,

$$\begin{cases} \tilde{m}_{++}(0) = m_2(0) + \alpha I_S^{22}(0) > 0, \\ \tilde{m}_{+-}(0) = 0, \\ \tilde{m}_{-+}(0) = 0, \\ \tilde{m}_{--}(0) = -m_2(0) + \alpha I_S^{22}(0) < 0, \end{cases} \tag{137}$$

and we obtain

$$\begin{cases} \sigma_H^{++} = 0, \\ \sigma_H^{+-} = -e^2/2h, \\ \sigma_H^{-+} = e^2/2h, \\ \sigma_H^{--} = 0, \end{cases} \tag{138}$$

with total Hall conductivity zero. Interestingly, in this case the symmetric magnetism in the middle does not even quantitatively change the half quantum mirror Hall phase. On the other

hand, for $\alpha I_S^{22}(0) > m_2(0)$,

$$
\begin{cases}
\tilde{m}_{++}(0) = m_2(0) + \alpha I_S^{22}(0) > 0\,, \\
\tilde{m}_{+-}(0) = 0\,, \\
\tilde{m}_{-+}(0) = -m_2(0) + \alpha I_S^{22}(0) > 0\,, \\
\tilde{m}_{--}(0) = 0\,,
\end{cases}
\tag{139}
$$

and we obtain

$$
\begin{cases}
\sigma_H^{++} = 0\,, \\
\sigma_H^{+-} = -e^2/2h\,, \\
\sigma_H^{-+} = 0\,, \\
\sigma_H^{--} = -e^2/2h\,,
\end{cases}
\tag{140}
$$

with total Hall conductivity unit upon $e^2/h$. This unit is fundamentally different the $C = 1$ as Chern insulator case, since here $1 = 1/2 + 1/2$, with non-vanishing contribution coming from two metallic bands describing gapless Dirac fermions. It is recognized that the phase transition happens only within $\chi = -$ sub-blocks, where $\zeta = \pm$ Dirac fermions exchange their low-energy mass when crossing the qualitative phase transition point $I_S^{22}(0) = m_2(0)$, and by treating approximately $I_S^{22} \approx \alpha t_\perp$, $m_2(0) \approx m_0$, we see the qualitative critical point as

$$
\alpha_c^{\text{quali}} = \frac{m_0}{t_\perp} \approx 0.7\,,
\tag{141}
$$

which is close to the numerical result.

$I_S^{12}$ as inter-$n$ Dirac fermions coupling plays an important role here. Without this term, $n = 1$ and $n = 2$ Dirac fermions will totally be decoupled from Eq. (128), which makes the mass exchange between $\zeta$-Dirac fermions with $\chi = -$ impossible. With this term, which serves as an avoid-crossing source between $\zeta$-Dirac fermions masses, and obtains its maximum nearly after surface to bulk transition of $n = 1$ gapless Dirac fermions, we see that the crossing behavior of $\tilde{m}_{-,\zeta}$ at $\Delta_-(k_{\text{cross}}) = 0$ is prohibited by a non-zero $I_S^{12}(k_{\text{cross}})$, and the two bands are forced to exchange masses before and after $k_{\text{cross}}$. This is possible since $\Delta_-(k) = 0$ requires that $I_S^{11}(k) > I_S^{22}(k)$, which can happen only when $n = 1$ surface states emerge into the bulk at $k > k_c$, where $I_S^{12}(k)$ is also non-zero.

### 6.1.2 Lower threshold by decreasing the mass in the middle

It was pointed out [141, 142] that magnetic doping can reduce and even drive the bulk band gap $m_0$ of TI into a trivial one, and this effect plays a positive role in realizing the metallic QAHE indeed. To illustrate this, consider a simplified scenario where the bulk mass of TI, initially $m_0 = 0.28$ eV, is reduced to $\tilde{m}_0 = 0.08$ eV in the magnetically doped region. Then by comparing the detailed effective mass evolution in Fig. 28 with the original case in Fig. 26, we observe that the critical point $\alpha_c$ decreases to approximately $\alpha_c \approx 0.435$. Such a reduction is beneficial for the experimental realization of the metallic QAHE. Moreover, since the decrease in the critical $\alpha$ is positively correlated with the reduction of doped middle layer mass, while and this mass reduction itself is also positively correlated with the increase in concentration of magnetic doping, it is expected that the metallic QAHE can be achieved with a significantly lower threshold of magnetic doping concentration in practice. Note that if the middle layers are driven to a trivial state with a negative bulk mass $\tilde{m}_0 < 0$, and are simultaneously considered nonconductive, the system effectively splits into two semi-magnetic TI films, a trivial metallic QAHE comprising two non-communicative half QAHEs with the same half-quantized Hall conductance is obtained. It is important to recognize that the above calculation assumes an oversimplified relationship between doping concentration and the reduced

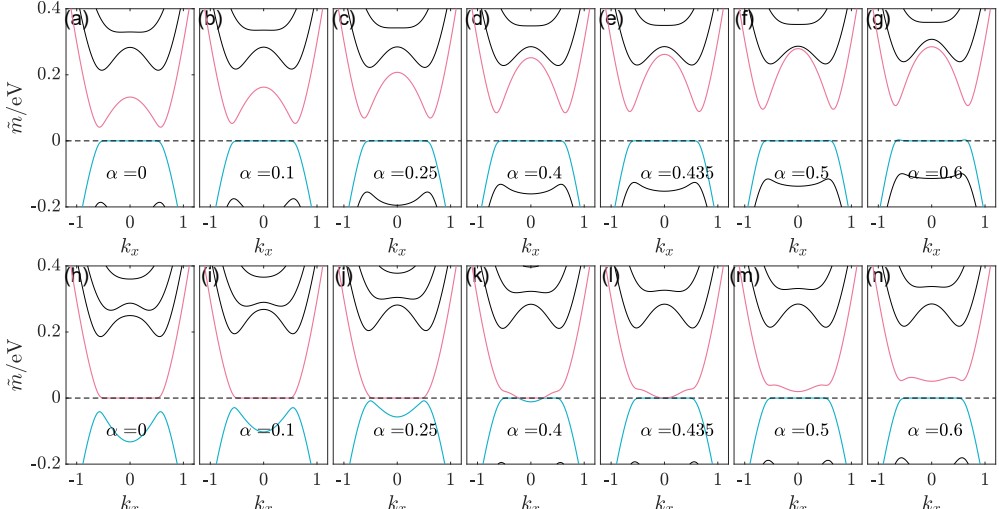

Figure 28: The evolution of the effective mass $\tilde{m}_{n,\chi}(k_x, k_y = 0)$. (a)~(g) and (h)~(n) emphasize lowest-two ($n = 1, 2$) **effective masses** in red and blue colors, varying with changing Zeeman field strength $\alpha$ belonging to set $[0, 0.1, 0.25, 0.4, 0.435, 0.5, 0.6]$ for $\chi = +$ and $\chi = -$, respectively. Still, we take total layer number $L_z = 22$ and middle Zeeman layer number $m_z = 6$, while the difference with Fig. 26 is that here the middle-layer mass is reduced to $\tilde{m}_0 = 0.08$ eV.

mass. A more accurate determination of the modified critical point requires a realistic model and a self-consistent calculation.

### 6.1.3 Stronger field in the middle

Encouraged by the mass exchange series diagrams, a natural question to ask is what happens when we increase Zeeman strength in the middle further. A first step answer to the ask is we will meet a system with higher Hall conductance. For instance, after increasing Zeeman field strength to $\alpha = 1.2$, we see from Fig. 30(a) that the Hall conductivity of the system becomes $-2e^2/h$ now. For the reason behind, we again look on the effective masses presented in Fig. 30(b), where a pair of gapless Dirac cones and one non-trivial gapped Dirac cone with mass sign reversal emerge, and essentially, from Eq. 38 and Eq. 42, they contribute synergistically to the Hall conductivity, i.e., $1/2 + 1/2 + 1 = 2$ units over $-e^2/h$. A careful trace over the effective mass evolution upon increasing $\alpha$ reveals that, at this time, $n = 3$ band of $\chi = +$ closes and reopens the gap, during which an avoid crossing happens and forces it to exchange low energy mass with $n = 1$ band of $\chi = +$, which leads to the result above.

## 6.2 Higher Chern number insulator

Based on magnetic TI film, several proposals to realize higher Chern number have been provided [80, 131, 142], among which one theoretical proposal [80] utilizes one-by-one sub-band inversion to illustrate the increasing Chern number process. Here the physics behind is brought out in a more strict way with a similar picture.

Still we firstly present an example shown in Fig. 31(a) as the Chern numbers of a uniformly magnetized TI film with total layer number $L_z = 8$. The algorithm follows [143]. With increasing the uniform Zeeman strength $V$, the change of Chern numbers experiences three stages: For the relatively weak Zeeman field, the Chern number plateau increases step by step, from 0 to 8, as revealed by red dots; For the Zeeman field with medium strength, the Chern number plateau drops from 8 to $-8$ with 2 as a step, illustrated by blue dots; Finally for the

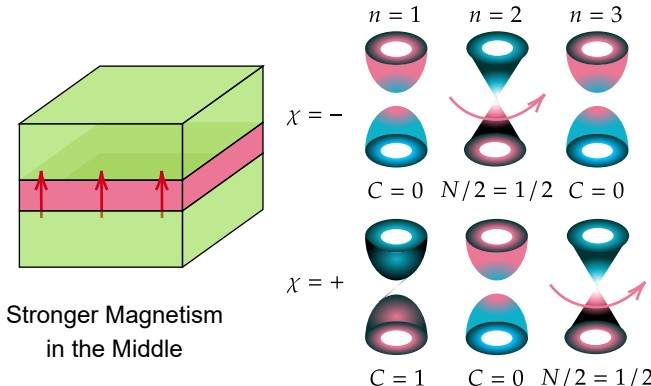

Figure 29: Schematic diagram of a stronger magnetism in the middle of the topological insulator film. The system now contains a pair of gapless Dirac fermions with the same high-energy mass signs, together with one non-trivial gapped Dirac cone with unit Chern number. The contributions of these Dirac fermions are synergistic.

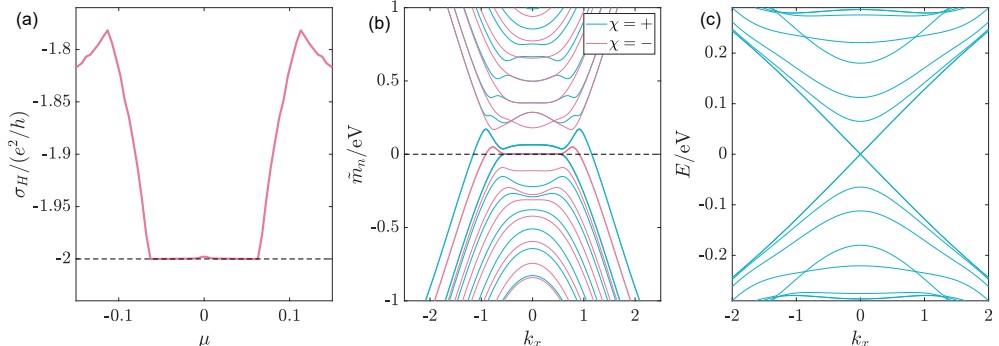

Figure 30: (a) Hall conductance of a metallic QAHE with a stronger magnetic field in the middle. (b) Momentum-dependent effective masses of Dirac fermions in Eq. (68). The masses for non-trivial bands have been stressed in the same color. (c) Band dispersion for the system, where the gapless bands at $\Gamma$ are doubly degenerate. Specifically, here the total layer number of TI film is $L_z = 22$, the magnetic layer number is the middle is 6, and the Zeeman strength is $V = \alpha t_\perp$ with $\alpha = 1.2$.

relatively strong Zeeman field, the Chern number plateau again increases from $-8$ to $0$ one-by-one, shown by purple dots. Notice that under our parameter choice we have $m_1(\pi,0) \sim 2$ eV and $m_1(\pi,\pi) \sim 4.3$ eV.

The Hamiltonian Eq. (70) now best suits to describe the phenomenon, where the uniform Zeeman field makes it exact to preserve diagonal blocks only. However, due to the largely adjustable magnitude of the Zeeman field, Eq. (83) becomes inapplicable here, and a more general formula following Eq. (45) is written as [27, 50, 98]

$$C_\chi = -\frac{\text{sgn}(\tilde{m}_\chi(X))}{2}[\text{sgn}(\tilde{m}_\chi(\Gamma)) - \text{sgn}(\tilde{m}_\chi(M))], \tag{142}$$

i.e., it accounts for the mass sign-change induced topological phase transition at $X = (\pi, 0)$. In this case, the $\chi$-Chern number for each $n = 1, \cdots, L_z$ is written as

$$C_\chi^n = -\frac{\text{sgn}(V + \chi m_n(X))}{2}[\text{sgn}(V + \chi m_n(\Gamma)) - \text{sgn}(V + \chi m_n(M))]. \tag{143}$$

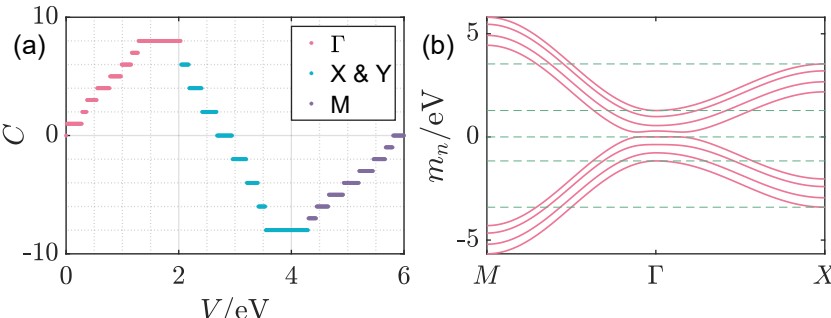

Figure 31: (a) Chern numbers of magnetic TI film with varying uniform Zeeman field strength $V$. Red, blue and purple dots represent Chern numbers caused by $\Gamma$, $X/Y$ and $M$ mass inversions, respectively. (b) Calculated mass $m_n(\boldsymbol{k})$ along $M{-}\Gamma{-}X$ high symmetry line. Green guidance lines have been imposed to reveal either zero-energy surface state plateau or relative magnitude of masses among high symmetry points. Total layer number $L_z = 8$.

In our case, $|m_n(\Gamma)| < |m_n(X)| < |m_n(M)|$, and admittedly, all bulk bands $n \geq 2$ are trivial by which we mean $m_n(\Gamma/X/M)$ share the same sign, then focusing on one band and increasing $V$ from zero, we see that when $V$ just crosses $|m_n(\Gamma)|$, the band with $\chi m_n < 0$ increases its Chern number from zero to one; continuing to increase $V$ so that it is bigger that $m_n(X)$, the corresponding Chern number reverses its sign from 1 to $-1$; and finally when $V$ goes beyond the bandwidth $|m_n(M)|$, the band goes back to its trivial phase with zero Chern number. Notice that under our assumption $V > 0$, the band $\bar{\chi} m_n > 0$ is always trivial.

It is now clear that the sub-band mass-inversion at $\Gamma$, $X$ and $M$ points are responsible for the change of Chern numbers, or equivalently the anomalous Hall plateaus with quantum units of conductance revealed in Fig. 31(a). As presented in Fig. 31(b), the masses $m_n(\boldsymbol{k})$ now share the property that $\max[m_n(\Gamma)] < \min[m_l(X)]$, $\max[m_n(X)] < \min[m_l(M)]$, as revealed by the green guidance lines. Then the Chern number change can be divided into three regions with increasing Zeeman field $V$ labelled in Fig. 31(a), i.e., the $\Gamma$-mass inverse region, the $X(Y)$-mass inverse region and the $M$-mass inverse region, without crossing among distinct regions. The physics happening in each region is exactly $L_z = 8$ copies illustrated above with increasing $V$, i.e., the Chern number increases one-by-one in the $\Gamma$-region each time Zeeman field $V$ crosses some $|m_n(\Gamma)|$ and makes the band non-trivial, until it reaches its maximum $C_{\max} = L_z = 8$, then decreases two-by-two in the $X$-region once $V$ gets bigger than some $|m_n(X)|$, where topological phase transition happens with both sides non-trivial, until bottom touching $C_{\min} = L_z - 2L_z = -8$, and finally the Chern number goes back to zero step-by-step in the $M$-region as long as $V$ becomes bigger than some bandwidth $|m_n(M)|$ and makes corresponding band trivial again. The inverse process happens for an opposite Zeeman field, with Chern number reversing its sign.

## 6.3 Cooperation between middle and surfaces

Similar to the approach of gapping out surface(s) of a topological insulator film, we can gap out the surface states in metallic QAHE with surface magnetism polarized along $z$ direction. In this sense we explore the cooperation between magnetism in the middle and at surface(s).

The surface magnetism is chosen to be weak compared to the smallest gap in metallic QAHE, and it can thus be treated again as a perturbation. This is simply because gapping out the gapless surface needs no threshold over surface magnetic strength. Based on such a picture, the physics beneath comes from perturbating two gapless Dirac fermions with the same high-energy mass signs in metallic QAHE, whose simplified model Hamiltonian reads

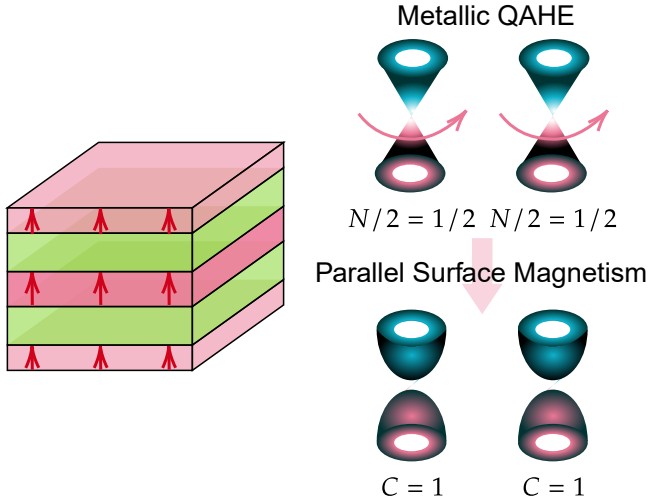

Figure 32: Schematic diagram of the metallic quantized anomalous Hall effect with top and bottom symmetric magnetism parallel to that in the middle. In the case a relatively strong Zeeman field exists in the middle of the film, while top and bottom states are gapped out by a weak Zeeman field. The system is now an insulator again, and contains a pair of gapped Dirac cones, each carrying Chern number one.

$H_{\text{MQAHE}} = h \oplus h$ with single Dirac cone Hamiltonian

$$h(\boldsymbol{k}) = \lambda_\parallel(\sin(k_x a)\sigma_x + \sin(k_y b)\sigma_y) + \text{sgn}(V^{\text{mid}})\tilde{m}(\boldsymbol{k})\sigma_z, \tag{144}$$

with $\tilde{m}(\boldsymbol{k}) = \Theta(-m_0(\boldsymbol{k}))m_0(\boldsymbol{k})$ identified. Considering now in metallic QAHE, the middle Zeeman field does not affect the gapless surface states, then the projection of top and bottom Zeeman fields onto the mirror-symmetric surface states can still be written as $I_S(\boldsymbol{k})\tau_0\sigma_z - I_A(\boldsymbol{k})\tau_y\sigma_z$. And by approximation, we recognize $I_S \equiv V_S^{\text{top}}$, $I_A \equiv V_A^{\text{top}}$ so that the phenomenological mass terms read

$$\text{sgn}(V^{\text{mid}})\tilde{m}(\boldsymbol{k})\tau_0 + V_S^{\text{top}}\tau_0 + V_A^{\text{top}}\tau_y, \tag{145}$$

which can be diagonalized without affecting linear term as

$$\tilde{m}_\zeta(\boldsymbol{k}) = \text{sgn}(V^{\text{mid}})\tilde{m}(\boldsymbol{k}) + V_S^{\text{top}} + \zeta V_A^{\text{top}}, \tag{146}$$

with $\zeta = \pm$. Attributing to Eq. (42), we have for a gapped Dirac cone with $V_S^{\text{top}} + \zeta V_A^{\text{top}} \neq 0$,

$$C^\zeta = \frac{1}{2}\left[\text{sgn}(V_S^{\text{top}} + \zeta V_A^{\text{top}}) + \text{sgn}(V^{\text{mid}})\right], \tag{147}$$

while for a gapless Dirac cone with $V_S^{\text{top}} + \zeta V_A^{\text{top}} = 0$, according to Eq. (38) we have

$$N^\zeta = \text{sgn}(V^{\text{mid}}), \tag{148}$$

and the corresponding Hall conductivity reads $\sigma_H^\zeta = -Ce^2/h$ or $\sigma_H^\zeta = -Ne^2/2h$ depending on gapped or gapless nature, which serves as the starting point for analyzing phases below.

For an instance, adding gap opening $z$-Zeeman field at both top and bottom surfaces parallel to magnetic polarization in the metallic QAHE system leads to $C = 2$ state, composed of a pair of non-trivial gapped Dirac fermions each carrying unit Chern number, as represented in Fig. 32. Such $C = 2$ state has been observed [142] in a similar magnetic structure with an alternate explanation based on the assumption that magnetic layers dividing topological insulator film do not hold side surface states, which then turns the magnetic insulator-topological

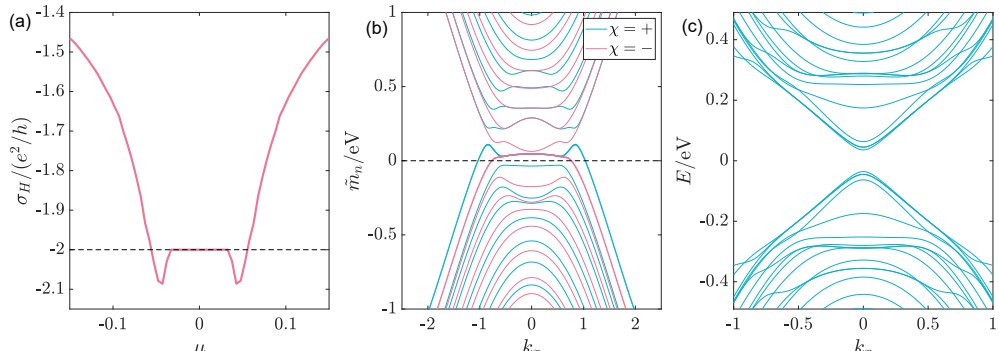

Figure 33: (a) Hall conductivity of a metallic QAHE with its top and bottom surface states also gapped by magnetism, whose polarization direction is parallel to the field in the middle. (b) Momentum-dependent effective masses of Dirac fermions in Eq. (68). Due to the symmetric Zeeman configurations, masses are again divided into mirror classes by $\chi = \pm$. The masses for gapped surface states have been stressed in the same color. (c) The band structure of the system. Specifically, here the total layer number of TI film is $L_z = 22$, the magnetic layer numbers at top, middle and bottom are 2, 6, 2, with mean Zeeman strengths chosen to be $V^{\text{top}} = 0.05$ eV, $V^{\text{mid}} = \alpha t_{\perp}$ with $\alpha = 0.9$, and $V^{\text{bot}} = 0.05$ eV, respectively.

insulator multilayer structure into individual $C = 1$ insulators, each of which can be explained by the discussion over Chern insulator in the weak Zeeman field section. Here instead we assume that the magnetism does not alter the bulk gap $m_0$ very much, so that the side surface state goes throughout the zone with magnetism. The calculated Hall conductivity for one configuration following the assumption is shown in Fig. 33(a), where a $C = 2$ plateau is presented inside the top/bottom Zeeman gap for surface states. The system is thus identified as a Chern insulator by the gapped band structure shown in Fig. 33(c). For simplicity, we have chosen a symmetric surface Zeeman distribution with $V^{\text{top}}_A = 0$. Now since $V^{\text{top}}_S > 0$, $V^{\text{mid}} > 0$, we have mass sign changes at $\Gamma$ and $M$ for both surface states as revealed by mass configurations in Fig. 33(b), and by Eq. (147)

$$C^+ = C^- = 1\,, \tag{149}$$

which leads to totally a $C = 2$ state.

Now let us switch down $V^{\text{top}}$, which makes $V^{\text{top}}_S = -V^{\text{top}}_A > 0$, accordingly we have $N^+ = 1$, $C^- = 1$, which corresponds to a system with Hall conductivity $3e^2/2h$. Further we re-add $V^{\text{top}} = -V^{\text{bottom}} < 0$, which leads to $V^{\text{top}}_S = 0$, $V^{\text{top}}_A > 0$, and we see $C^+ = 1$, $C^- = 0$, which makes the system a Chern insulator again with unit Chern number. Next we reverse $V^{\text{bottom}}$ to minus, and $V^{\text{top}}_S < 0$, $V^{\text{top}}_A = 0$, which makes the system trivial with $C^+ = C^- = 0$. Finally, we switch down again $V^{\text{top}}$, and now $V^{\text{top}}_S = -V^{\text{top}}_A < 0$, accordingly we have $N^+ = 1$, $C^- = 0$, which leaves half quantization of Hall conductivity in the system.

Totally, we see that there exist five more additional topologically distinct phases upon tuning surface magnetism of metallic QAHE, with Hall conductivities quantized into $2, 3/2, 1, 1/2$ and $0$ over quantum units, respectively. The topological properties of these additional phases can be easily verified by calculating their Hall conductivities, or reading from their effective mass pictures. The signs of Hall conductivities are inverted once we overturn magnetism at both surfaces and in the middle.

Table 3: Summation of main magnetic topological phases discussed. $C$ represents Chern number for a fully occupied band, while $N$ is the half-integer index for a metallic band. The Hall conductance $\sigma_H = -(C + N/2)(e^2/h)$ when the chemical potential lies inside the insulating gap and symmetry constrained regime of the metallic band.

| Name of phase | Magnetic structure | Responsible Dirac fermion(s) | Topological index |
|---|---|---|---|
| Half quantum mirror Hall effect | | $N/2 = 1/2 \quad N/2 = -1/2$ | $N^{\text{mirror}} = 1 - (-1) = 2$ |
| Half quantized anomalous Hall effect | | $C = 0 \quad N/2 = 1/2$ | $C = 0, N = 1$ |
| Metallic quantized anomalous Hall effect | | $N/2 = 1/2 \quad N/2 = 1/2$ | $N = 1 + 1 = 2$ |
| Chern insulator | | $C = 0 \quad C = 1$ | $C = 0 + 1 = 1$ |
| Axion insulator | | $C = 0 \quad C = 0$ | $C = 0 + 0 = 0$ |

# 7 Discussion and conclusion

It is quite remarkable and surprising that so many topologically distinct phases already emerge under such a relatively simple model describing a magnetic topological insulator film. At the core of physics, however, such a descriptive and predictive power of the frame should be estimated. Although, admittedly infinite possibilities exist to explain the phenomena, down to the ground several principles, such as symmetry, topology, emergence and conciseness, have almost fixed the formalism we are willing to adapt in addressing the problem. In our focused questions, particularly regarding the Hall conductances for different species of Dirac fermions in the system, the property of several points in the spectrum is already sufficient to solely determine the result. And to endow physical meaning to these points, we name the points to represent low-energy and anomaly. The invariance of laws of physics then suggests that, once we have grasped these key ingredients, the complexities of the more intricate components will naturally fall into place. Below we summarize key points in our paper and extend to further discussions.

The introduced local unitary transformation in $k$-space, based on the exact solution, unveils the existence of a pair of gapless Dirac fermions and a series of massive gapped Dirac fermions in a 3D topological insulator film, when viewed as 2D system effectively. This comprehensive understanding of the constitutes inside the TI film is paramount, as our derivation here is a complete extension of the previous work on projection of TI surface states [36–38], with the inclusion of the high-energy part of the surface bands in the full 2D Brillouin zone and higher massive Dirac fermions for bulk bands.

The Hall conductivity associated with the gapless and gapped Dirac fermions in the TI film are $\pm e^2/2h$ and 0, respectively. This results in a half-quantized topological phase, serving as a metallic partner to the insulating quantum spin Hall effect, namely, the half quantum mirror Hall effect in TI film itself with a mirror symmetry. The pairing feature of the gapless Dirac fermions in half quantum mirror Hall effect is summarized in Table 3. It is noteworthy that their existence here is not a result from the Nielsen-Ninomiya theorem, since they are two separable fermions in whole Brillouin zone; rather, it is the mirror symmetry along the opened direction of the TI film that requires the doubling — symmetric and antisymmetric.

The mass term of the gapless Dirac fermion in our study is a regularized one that can be directly expressed on a lattice. However, this regularization comes at the cost of introducing an explicitly parity-symmetry-breaking term away from the Dirac point. As a result, the gapless Dirac fermion remains massless at low-energy but becomes massive at high-energy. In the article, a Heaviside Theta function is utilized to grasp the feature of such a mass term, which exhibits long-range algebraic decay with the first power modified by a sinusoidal function, when Fourier transformed into real space. Specifically, it contains a hopping term proportional to $\sim \sin(\Delta l)/\Delta l$, with $\Delta l$ being the distance between sites. Not accidentally, a similar hopping term with the same algebraic decaying order has been used as one way to construct single gapless Dirac fermion on lattice, known as the SLAC fermion [35,135,136]. However, it is important to note that in our theory, the phenomenological evasion of locality by the gapless Dirac fermion, residing in effectively 2D space, is a consequence of the bulk property of the 3D TI, where locality is preserved. This phenomenon underscores the concept of bulk-boundary correspondence and suggests that a seemingly unphysical theory in lower dimensions can be attributed to a projection from a higher-dimensional theory. It is noteworthy that the procedure employed here is different from a dimensional reduction, and is not an effective field theory because the Dirac fermion naturally obtains completeness on lattice. Rather, a better similarity can be shared with the quasicrystal containing aperiodic order, which can arise from projections of higher-dimensional periodic lattices [144]. Essentially, both the gapless Dirac fermion containing surface states of a 3D TI, and the quasicrystal from tilings, are physically realizable systems.

The formalism introduced here, involving the transformation of a confined spatially $(n + 1)$D Dirac Hamiltonian into $n$D Dirac fermions through the construction of a local unitary matrix using solutions from a decomposed 1D Hamiltonian along the confined direction, can be generalized to arbitrary dimensions, with the aid of Clifford algebra. In particular, initiating from a 4D space modified Dirac equation, a unitary transformation yields a pair of gapless Dirac fermions effectively in 3D space. This extension holds the potential to enhance our comprehension of the chiral anomaly in the system [56, 57]. What is more, given that the high-energy components of the two Dirac fermions explicitly break the chiral symmetry, they are not obliged to be paired by violating conditions stipulated by the Nielsen-Ninomiya theorem. As a result, we can anticipate that when the constrained 4D Hamiltonian becomes 'semi-magnetic', a single gapless Dirac fermion will be observed, similar to that in half QAHE.

The introduced magnetism, initially presented as an out-of-plane Zeeman field at the mean-field level, undergoes the unitary transformation into two momentum-dependent matrix Higgs fields $\mathbf{I}_{S/A}(\boldsymbol{k})$, which obtain non-vanishing values along with the spontaneous symmetry break-

ing that establishes intralayer ferromagnetic order. The two fields play a pivotal role in generating mass to the Dirac fermions through Yukawa-like couplings. The nature of the magnetic structure, influencing the distribution and strength of the Zeeman field along the open direction, leads to the classification of several topologically distinct phases, including the Chern insulator, axion insulator, half-quantized anomalous Hall effect and metallic quantized anomalous Hall effect. A summary of their main features is presented in Table 3. Essentially, $\mathbf{I}_S$ predominates in the Chern insulator and metallic QAHE phases, $\mathbf{I}_A$ takes precedence in the axion insulator, while a collaborative effort between both $\mathbf{I}_S$ and $\mathbf{I}_A$ is necessary to achieve the half QAHE.

In the presence of a uniform Zeeman field, the mass of each Dirac fermion in TI film is directly modified by a Zeeman field. By tuning the strength of magnetism, sub-band inversion happens step-by-step for each Dirac fermion, whose Chern character changes correspondingly. Summing those mass-modified Dirac fermions together gives a Chern insulator that carries jumping Hall conductance among integers in $[-L_z, L_z]$ over the quantum unit $e^2/h$, with $L_z$ the total layer number.

With a relatively weak Zeeman field compared with the bulk gap, focusing solely on the $n = 1$ matrix elements that act on the two gapless Dirac fermions becomes feasible. In this scenario, only fields near the two surfaces maximally tune the topological property of the TI film by influencing the surface states. This approximation, referred to as the weak Zeeman field condition, elucidates the underlying physics behind the Chern insulator, axion insulator and half QAHE clearly, with Hall conductance showing $1 + 0$, $0 + 0$ and $1/2 + 0$ quantized nature upon quantum unit.

Under a general strong Zeeman field, the gapped series of Dirac fermions have to be involved, and the $n \neq 1$ Higgs components can play a crucial role. The most general description is conducted by a further diagonalization over mass terms $m_n$ and Higgs fields $\mathbf{I}_{S/A}$, and the procedure leads to effective masses $\tilde{m}_n$ for the Dirac fermions, which determine the topological property of the system. As discussed, the avoid-crossing between $\tilde{m}_1$ and $\tilde{m}_2$ leads to the formation of two gapless Dirac fermions with the same chirality (high-energy mass sign) in system, which bears a doublet of half quantized Hall conductivity and leads to the metallic QAHE. Interestingly, in the case, another cut-off over $n = 1, 2$ blocks can be made, since the Zeeman field applied should not alter the $n \geq 3$ states dramatically.

When $\mathbf{I}_A = 0$, the mirror symmetry is respected by the system, allowing for the separation of the total Hamiltonian by the projection operator of mirror symmetry. This separation provides valuable insights, such as the application of mirror layer Chern number in a Chern insulator with a unit Chern number.

It is certainly reasonable but lamentable that we cannot exhaustively list all relevant topological phases in magnetic topological insulators in the article. The sheer multitude of possible magnetic distributions makes it impractical to cover every potential scenario. However, our work lays down a unified framework that enables the depiction of both discovered and yet-to-be-discovered topological phases in a uniform and consistent manner, grounded in the conceptualization of the grouped Dirac fermions and the associated mass generation mechanism. We believe that the diversity and variety of different magnetic configurations can lead to even richer topological phases within our framework.

Furthermore, as elaborated in Section 4.2, our exploration is not confined solely to topological phases induced by magnetism, especially a Zeeman field in the TI film. One illustrative example, as highlighted earlier, involves the duality between the $z$-Zeeman field $\sigma_z$ and a special orbital order $\tau_y$. This duality has the potential to generate all topological phases discussed in the paper, with symmetric and antisymmetric distributions exchanged for the time-reversal-breaking $\tau_y$ field. This approach extends beyond the commonly studied ferromagnetism (or layer-by-layer antiferromagnetism, as observed in materials like $MnBi_2Te_4$) induced quantum

anomalous Hall effect (QAHE). Moreover, leveraging the superconducting effect, we can include the superconducting pairing field into the frame across all pairing symmetries. This inclusion opens avenues for exploration and determination of the possibilities and conditions necessary for realizing topological superconductors [145–148] within the solid framework we have established.

An additional intriguing aspect to consider pertains to the symmetries in the system. The modified Dirac equation model we employed for the topological insulator film encapsulates fruitful symmetries, like the standard time reversal, particle hole and chiral symmetries, together with the inversion symmetry in each dimension and the 1D mirror symmetry along each direction. Some of these symmetries play crucial roles in determining our solutions and topological phases in the system. For instance, in solving the separated 1D Hamiltonian, the utilization of one-dimensional parity and chiral symmetry is essential; the $z$-mirror symmetry becomes decisive for the manifestation of the half quantum mirror Hall effect, contributing to quantized mirror Hall conductance; despite not a protecting symmetry in the metallic quantized anomalous Hall effect, the ever existence of the same mirror symmetry helps us to cut the effective masses into two groups by their mirror labels and clarifies the mass exchange mechanism. It may prove worthwhile to contemplate a starting point Hamiltonian with lower symmetry or introduce additional symmetry-breaking fields to assess the stability of these effects. For instance, the half quantum mirror Hall effect is clearly a metallic twin partner of the quantum spin Hall effect, and it should also share a general $\mathbb{Z}_2$ classification scheme depending on the time reversal symmetry solely. Consequently, it is worthy to give a unified expression for this invariant. Moreover, as we have shortly discussed, the half-quantization of the gapless Dirac fermion is protected by the parity invariant regime around the Dirac point, and indeed, this 2D parity symmetry coexists with the time reversal in our model, which warrants further discussion regarding their individual impacts on half-quantization. This exploration can be extended to encompass broader symmetries and other kinds of metallic topological phase classes, providing a comprehensive understanding.

Besides, the exploration of disorder and interaction effects in metallic phases presents a rich avenue for investigation. As previously discussed, metallic topological phases inherently grapple with disorder effects on their metallic side, wherein mechanisms like skew-scattering and side-jump alter the transverse transport behavior [89, 92]. The stability of these phases against disorder, addressed through parameter renormalization, poses a significant question, akin to considerations in their insulating counterparts [94, 149–152]. Moreover, while the adiabatic criterion justifiably establishes a connection between a gapped interacting phase and a non-interacting one by preserving gap opening, it remains elusive in what way we can say something similar in those metallic phases. Clarifying how this linkage can be articulated in the context of these metallic phases poses an ongoing challenge.

In short, the interplay between magnetism and topology in 3D TI film is investigated under a unified frame, exploiting the Dirac fermion physics and mass generating mechanism.

# Acknowledgments

**Author contributions**  S.Q. S. conceived the project. K.-Z. B. and B. F. performed the theoretical analysis and simulation. K.-Z. B. and S.Q. S. wrote the manuscript with inputs from all authors. All authors contributed to the discussion of the results.

**Funding information**  This work was supported by the National Key R&D Program of China under Grant No. 2019YFA0308603, the Research Grants Council, University Grants Committee, Hong Kong under Grant Nos. C7012-21G and 17301823 and the Quantum Science Center of Guangdong-Hong Kong-Macao Greater Bay Area under Grant No. GDZX2301005.

# A  Derivation of Eq. (1a)

We start from solving

$$h(s) = -is\lambda_\perp \partial_z \tau_x + (m_0(\boldsymbol{k}) + t_\perp \partial_z^2)\tau_z\,, \tag{A.1}$$

with $s$ defined by eigenvalue of $\sigma_z$. All parameters are real with $m_0(\boldsymbol{k}) = m_0 - t_\parallel k^2 > 0$ to be the criterion for the region where surface states emerge. For the purpose of keeping consistence with the lattice model in 2.2, one in fact needs to substitute parameters as

$$\lambda_\perp \to c\lambda_\perp\,, \qquad \lambda_\parallel \to a\lambda_\parallel\,, \qquad t_\perp \to c^2 t_\perp\,, \qquad t_\parallel \to a^2 t_\parallel\,.$$

However, we would not write in that way explicitly for simplicity. Also, to make discussion pithy, we shall omit $s$ in wavefunction below.

The eigenproblem of $h(s)$ is a second-order differential equation and allows us to set solutions with trial function $\phi = \phi_\xi e^{i\xi z}$. Using $\partial_z \phi = i\xi\phi$, $\partial_z^2 \phi = -\xi^2\phi$, one has equation below:

$$\begin{pmatrix} m_0(\boldsymbol{k}) - t_\perp \xi^2 & s\lambda_\perp \xi \\ s\lambda_\perp \xi & -m_0(\boldsymbol{k}) + t_\perp \xi^2 \end{pmatrix}\phi = E\phi\,, \tag{A.2}$$

which readily leads to

$$E^2 - (m_0(\boldsymbol{k}) - t_\perp \xi^2)^2 - \lambda_\perp^2 \xi^2 = 0\,, \tag{A.3}$$

and gives

$$\xi_\alpha^p = p\xi_\alpha = p\sqrt{-\frac{F}{D} + (-1)^{\alpha-1}\frac{\sqrt{R}}{D}}\,, \quad p = \pm\,, \quad \alpha = 1, 2\,, \tag{A.4}$$

where

$$D = 2t_\perp^2\,, \qquad F = -2m_0(\boldsymbol{k})t_\perp + s\lambda_\perp^2\,, \qquad R = F^2 - 2D(m_0^2(\boldsymbol{k}) - E^2)\,.$$

For each $\xi_\alpha^s$, one has

$$\phi_{\alpha p} = \begin{pmatrix} s\lambda_\perp p\xi_\alpha \\ E - m_0(\boldsymbol{k}) + t_\perp \xi_\alpha^2 \end{pmatrix}\,, \tag{A.5}$$

and the general solution would be

$$\Phi = \sum_{\alpha p} C_{\alpha p}\phi_{\alpha p}e^{ip\xi_\alpha z}\,. \tag{A.6}$$

Now considering finite size along $z$ direction with top and bottom surfaces located at $\pm\dfrac{L}{2}$, respectively, one would have boundary condition

$$\Phi\left(\pm\frac{L}{2}\right) = 0\,, \tag{A.7}$$

applying which one would get four linear equations for coefficients

$$\mathbb{P}(C_{1+}, C_{1-}, C_{2+}, C_{2-})^T = 0\,, \tag{A.8}$$

and requirement $\det(\mathbb{P}) = 0$ leads to two transcendental equations

$$\frac{m_1\xi_2}{m_2\xi_1} = \frac{\tan\xi_2 L/2}{\tan\xi_1 L/2}\,, \tag{A.9a}$$

$$\frac{m_1\xi_2}{m_2\xi_1} = \frac{\tan\xi_1 L/2}{\tan\xi_2 L/2}\,, \tag{A.9b}$$

which gives two energies varying with $\boldsymbol{k}$, designated as $E_+$ and $E_-$, respectively. To be clearer,

$$E_+ = m_0(\boldsymbol{k}) - t_\perp \frac{\xi_1^2 g^+(\xi_1) - \xi_2^2 g^+(\xi_2)}{g^+(\xi_1) - g^+(\xi_2)}, \qquad g^+(\xi) = \frac{\tan(\xi L/2)}{\xi}, \tag{A.10a}$$

$$E_- = m_0(\boldsymbol{k}) - t_\perp \frac{\xi_1^2 g^-(\xi_1) - \xi_2^2 g^-(\xi_2)}{g^-(\xi_1) - g^-(\xi_2)}, \qquad g^-(\xi) = \frac{1}{\tan(\xi L/2)\xi}. \tag{A.10b}$$

In common sense, it is time taking $E_\pm$ into expressions of $\xi$s, together with the coefficients equations again and solve them. However, that not only is tricky but lacks of physical insight, and we shall change our perspective.

Notice that under parity operation $z \leftrightarrow -z$, $\tau_x \leftrightarrow -\tau_x$ and $h(s) \leftrightarrow h(s)$, then both $h(s)$ and $H_{1d}$ has parity symmetry and the general solution should contain two factors below considering the boundary condition:

$$\begin{cases} f_+(z) = \dfrac{\cos(\xi_1 z)}{\cos(\xi_1 L/2)} - \dfrac{\cos(\xi_2 z)}{\cos(\xi_2 L/2)}, \\ f_-(z) = \dfrac{\sin(\xi_1 z)}{\sin(\xi_1 L/2)} - \dfrac{\sin(\xi_2 z)}{\sin(\xi_2 L/2)}, \end{cases} \tag{A.11}$$

where the subscripts refer to even or odd parity. Now we can assume that for energy $E$, $h(s)$ has solution

$$\phi = \tilde{c} f_+ + \tilde{d} f_- = \begin{pmatrix} \tilde{c}_1 f_+ + \tilde{d}_1 f_- \\ \tilde{c}_2 f_+ + \tilde{d}_2 f_- \end{pmatrix}, \tag{A.12}$$

and the two-line eigenequation $h(s)\phi = E\phi$ gives, for the first line

$$\tilde{d}_2 = it_\perp \eta_1 \tilde{c}_1 / s\lambda_\perp, \tag{A.13a}$$

$$\tilde{c}_2 = -it_\perp \eta_2 \tilde{d}_1 / s\lambda_\perp, \tag{A.13b}$$

which leads to

$$\phi_1^+ = C_1^+ \begin{pmatrix} -is\lambda_\perp f_+ \\ t_\perp \eta_1 f_- \end{pmatrix}, \qquad E = E_+, \tag{A.14a}$$

$$\phi_1^- = C_1^- \begin{pmatrix} is\lambda_\perp f_- \\ t_\perp \eta_2 f_+ \end{pmatrix}, \qquad E = E_-, \tag{A.14b}$$

and for the second line,

$$\tilde{d}_1 = -it_\perp \eta_1 \tilde{c}_2 / s\lambda_\perp, \tag{A.15a}$$

$$\tilde{c}_1 = it_\perp \eta_2 \tilde{d}_2 / s\lambda_\perp, \tag{A.15b}$$

which leads to

$$\phi_2^+ = C_2^+ \begin{pmatrix} t_\perp \eta_1 f_- \\ is\lambda_\perp f_+ \end{pmatrix}, \qquad E = -E_+, \tag{A.16a}$$

$$\phi_2^- = C_2^- \begin{pmatrix} t_\perp \eta_2 f_+ \\ -is\lambda_\perp f_- \end{pmatrix}, \qquad E = -E_-, \tag{A.16b}$$

by defining two coefficients

$$\eta_1 = \frac{\xi_1^2 - \xi_2^2}{\xi_1 \cot(\xi_1 L/2) - \xi_2 \cot(\xi_2 L/2)}, \tag{A.17a}$$

$$\eta_2 = \frac{\xi_1^2 - \xi_2^2}{\xi_1 \tan(\xi_1 L/2) - \xi_2 \tan(\xi_2 L/2)}, \tag{A.17b}$$

with $C$ is the norm, and super and lower indices represent $E_\pm$ and line index, respectively. Clearly, $C_1^\iota = C_2^\iota$ is identified, and $\phi_1^\iota = -i\tau_y\phi_2^\iota$ as they are chiral partners ($\iota = \pm$).

Solution above seems to give four solutions, mathematical restriction, however, tells that equations from different lines for the same set of coefficients must stand simultaneously, i.e., (A.14a)$\Longleftrightarrow$(A.16b) and (A.14b)$\Longleftrightarrow$(A.16a), which gives us two relations as

$$1 = \left| \frac{it_\perp \eta_1}{s\lambda_\perp} \cdot \frac{it_\perp \eta_2}{s\lambda_\perp} \right| \Longrightarrow |\eta_1 \eta_2| = \frac{\lambda_\perp^2}{t_\perp^2}, \tag{A.18a}$$

$$E_+ = -E_-, \tag{A.18b}$$

and the latter is also a physical result from Dirac equation. Then, we only have two independent solutions for one $h(s)$ sub-block, say Eq. (A.14a) and Eq. (A.16a). Formal combination of equations for the simultaneous-standing equations from different lines again leads to

$$E^2 - (m_0(\boldsymbol{k}) - t_\perp \xi^2)^2 - \lambda_\perp^2 \xi^2 = 0. \tag{A.19}$$

Then we see that the guessing solution not only satisfies the boundary condition, but also satisfies all $E - \xi$ equations, thus it is indeed our solution.

Notice that, by Eq. (A.4), $\xi_\alpha$ are both complex or not complex for a given energy, where complex means both real and imaginary parts of $\xi$ are non-vanishing, determined by the sign of $R$. This information, combined with property of trigonometric/hyperbolic function leads to the conclusion that quadratic form $f_+^* f_-$ and $\eta$ (at certain $(\boldsymbol{k}, z, E)$) are always real. Essentially, $f_\pm$ are either real or purely imaginary.

Now, we restore $s$ explicitly and extract

$$\varphi(s) = \phi_1^{s,+}, \qquad \chi(s) = \phi_2^{s,+} \tag{A.20}$$

as two solutions for $h(s)$ for basis construction. Then by defining

$$\begin{cases} m = E_+ = m_0(\boldsymbol{k}) - t_\perp \dfrac{\xi_1^2 g(\xi_1) - \xi_2^2 g(\xi_2)}{g(\xi_1) - g(\xi_2)}, \\ g(\xi) = \dfrac{\tan(\xi L/2)}{\xi}, \\ \eta = \dfrac{\xi_1^2 - \xi_2^2}{\xi_1 \cot(\xi_1 L/2) - \xi_2 \cot(\xi_2 L/2)}, \\ C = C_1^+ = C_2^+, \end{cases} \tag{A.21}$$

one obtains four projecting basis in certain sequence as

$$\Phi_1 = \begin{pmatrix} \varphi(+) \\ 0 \end{pmatrix} = C \begin{pmatrix} -i\lambda_\perp f_+ \\ t_\perp \eta f_- \\ 0 \\ 0 \end{pmatrix},$$

$$\Phi_2 = \begin{pmatrix} 0 \\ \chi(-) \end{pmatrix} = C \begin{pmatrix} 0 \\ 0 \\ t_\perp \eta f_- \\ -i\lambda_\perp f_+ \end{pmatrix}, \tag{A.22}$$

$$\Phi_3 = \begin{pmatrix} \chi(+) \\ 0 \end{pmatrix} = C \begin{pmatrix} t_\perp \eta f_- \\ i\lambda_\perp f_+ \\ 0 \\ 0 \end{pmatrix},$$

$$\Phi_4 = \begin{pmatrix} 0 \\ \varphi(-) \end{pmatrix} = C \begin{pmatrix} 0 \\ 0 \\ i\lambda_\perp f_+ \\ t_\perp \eta f_- \end{pmatrix},$$

with energy $(m, -m, -m, m)$, respectively. Notice that $\Phi_{3,4}$ are chiral partners of $\Phi_{1,2}$ by $-i\tau_y$, respectively. To obtain $m$, a set of closed equations need to be solved

$$m = m_0(\boldsymbol{k}) - t_\perp \frac{\xi_1^2 g^+(\xi_1) - \xi_2^2 g^+(\xi_2)}{g^+(\xi_1) - g^+(\xi_2)}, \tag{A.23a}$$

$$\xi_\alpha = \sqrt{-\frac{F}{D} + (-1)^{\alpha-1}\frac{\sqrt{R}}{D}}, \quad \alpha = 1, 2, \tag{A.23b}$$

where

$$\begin{cases} g^+(\xi) = \tan(\xi L/2)/\xi, \\ D = 2t_\perp^2, \\ F = -2m_0(\boldsymbol{k})t_\perp + \lambda_\perp^2, \\ R = F^2 - 2D(m_0^2(\boldsymbol{k}) - m^2). \end{cases} \tag{A.24}$$

Basically, there are three variables $(m, \xi_1, \xi_2)$ with three equations, then they could be determined exactly.

## A.1 Symmetry analysis of solutions

Firstly, as we have stated, the chiral symmetry $\tau_y$ is respected in Eq. (A.1) since $\{h(s), \tau_y\} = 0$, and this symmetry is reflected in our solutions by $\varphi(s) = -i\tau_y \chi(s)$ with opposite energies.

Meanwhile, we have relied on the help from the 1D parity symmetry which is a reflection along $z$ direction, or simply, the $z$-parity $\mathcal{P}_z$, which acts on the basis as

$$\Phi(z) \xrightarrow{\mathcal{P}_z} \tau_z \Phi(-z), \tag{A.25}$$

with $\tau_z$ the unitary matrix related to inner degrees of freedom transformation. Now since $f_\pm(z) = \pm f_\pm(-z)$, we identify that $\Phi_{1,4}$ ($\Phi_{2,3}$) are even (odd) under $z$-parity, and correspondingly, under the representation of $\Phi$, the unitary matrix related to $z$-parity is written as $\tau_z \sigma_z$.

There exists in fact a hidden symmetry in the model, namely, the mirror symmetry about the $x$-$y$ plane. Effectively, it will also bring $z$ to $-z$ as an inversion, but with an additional operation that rotates spin angular momentum by $\pi$ phase, i.e., such a $z$-Mirror symmetry $\mathcal{M}_z$ is a combination of $\mathcal{P}_z$ and a $\mathcal{C}_{2z}$ rotation, which then acts on the basis as

$$\Phi(z) \xrightarrow{\mathcal{M}_z} \sigma_z \tau_z \Phi(-z), \tag{A.26}$$

and classifies $\Phi_{1,2}$ ($\Phi_{3,4}$) into $z$-mirror even (odd) states. Accordingly, under $\Phi$ representation this operator has form $\tau_z \sigma_0$. Then combined with the spin index $s = \pm$ appeared in $\varphi(s), \chi(s)$, we can further assign $\Phi_i$ to be $\Phi_{\chi,s}$ with $\chi, s$ labelling mirror and spin-$z$ index as

$$\begin{aligned} \Phi_{++} &= \Phi_1, & \Phi_{+-} &= \Phi_2, \\ \Phi_{-+} &= \Phi_3, & \Phi_{--} &= \Phi_4. \end{aligned} \tag{A.27}$$

The single $h(s)$ does not share time reversal symmetry, since under $\mathcal{T} = i\sigma_y \mathcal{K}$, $h(+) \leftrightarrow h(-)$, i.e., $H_{1d}$ owns this symmetry. Also given by the fact that time reversal keeps energy unconverted, one finds $\Phi_4 = e^{i\theta}\mathcal{T}\Phi_1$, $\Phi_2 = e^{i\theta}\mathcal{T}\Phi_3$, where $\theta = 0$ or $\pi$ depending on $k, E$. The essential point to get avoid of subtle $f_\pm^*$ is to notice that they are either both real or imaginary, as stated above, while $\eta$ is always real. Also notice that we did not write $k$ explicitly since $H_{1d}(\boldsymbol{k}) = H_{1d}(-\boldsymbol{k})$.

The combination of time reversal and chiral symmetries gives rise to a particle hole symmetry, which, when implanted over basis, reads $\varphi(s) = e^{i\theta}\varphi^*(\bar{s}) = e^{i\theta}[-i\tau_y\chi(\bar{s})]^*$, with $\bar{s} = -s$ identified.

Similar analysis applies for the lattice model, and the projected $\mathcal{P}_z$, $\mathcal{M}_z$ share the same matrix form above.

## A.2 Equivalent block Hamiltonian

The projection procedure works under the given basis representation $H_{TI}(\boldsymbol{k})$, which is formally $H = \langle \Phi | H_{TI}(\boldsymbol{k}) | \Phi \rangle$, with

$$(H)_{ij}^{nn'} = \int \mathrm{d}z \, (\Phi_i^n(z))^\dagger H_{TI}(\boldsymbol{k}, z)\Phi_j^{n'}(z), \tag{A.28}$$

where the integral is done from $-L/2$ to $L/2$. Clearly, projection on $H_{1d}$ would give $\mathrm{diag}(m, -m, -m, m)$, then we only need to deal with $H_\parallel(\boldsymbol{k}) = \lambda_\parallel(\boldsymbol{k}\cdot\boldsymbol{\sigma})\tau_x = \lambda_\parallel(k_x\sigma_x + k_y\sigma_y)\tau_x$ term. Since $H_\parallel(\boldsymbol{k})$ is purely off-diagonal, it is easy to conclude that

$$\langle \Phi_i^n | H_\parallel | \Phi_i^{n'} \rangle = 0, \quad i = 1, 2, 3, 4,$$
$$\langle \Phi_1^n | H_\parallel | \Phi_3^{n'} \rangle = 0 = \langle \Phi_2^n | H_\parallel | \Phi_4^{n'} \rangle.$$

Then only four terms need consideration by hermicity, among which

$$\langle \Phi_1^n | H_\parallel | \Phi_4^{n'} \rangle = \lambda_\parallel k_- |C^n C^{n'}| \int \mathrm{d}z \, i\lambda_\perp t_\perp [\eta^n(f_+^n)^* f_-^{n'} + \eta^{n'}(f_-^n)^* f_+^{n'}] = 0,$$

$$\langle \Phi_2^n | H_\parallel | \Phi_3^{n'} \rangle = \lambda_\parallel k_- |C^n C^{n'}| \int \mathrm{d}z \, i\lambda_\perp t_\perp [\eta^n(f_-^n)^* f_+^{n'} + \eta^{n'}(f_+^n)^* f_-^{n'}] = 0,$$

as $f_- f_+$ is odd to $z$. Here $k_\pm = k_x \pm i k_y$ is defined. Then, the only remaining terms are

$$\langle \Phi_1^n | H_\parallel | \Phi_2^{n'} \rangle = \int \mathrm{d}z \, \lambda_\parallel k_- \varphi^\dagger(\lambda_\perp)\tau_x\chi(-\lambda_\perp) = \lambda_\parallel k_- \delta_{nn'},$$

$$\langle \Phi_3^n | H_\parallel | \Phi_4^{n'} \rangle = \int \mathrm{d}z \, \lambda_\parallel k_- \varphi^\dagger(\lambda_\perp)\tau_x\chi(-\lambda_\perp) = \lambda_\parallel k_- \delta_{nn'},$$

where the normalization condition is used. And finally we arrive at the block Hamiltonian

$$H(\boldsymbol{k}) = \bigoplus_n \lambda_\parallel \tau_0(\boldsymbol{k}\cdot\boldsymbol{\sigma}) + m_n(\boldsymbol{k})\tau_z\sigma_z, \tag{A.29}$$

as Eq. (1a). Here, notice that the spin degree of freedom is fully preserved as $\sigma$, while the newly-defined $\tau$ owns different meaning from the original one.

To make the transformation more formal, we define the transformation matrix

$$U^c(\boldsymbol{k}, z) = (\{\{\Phi\}_i\}^n)(\boldsymbol{k}, z), \tag{A.30}$$

where the double brackets mean that we arrange $i = 1, 2, 3, 4$ index inside each $n = 1, 2, \cdots$, and by written more straightforwardly,

$$U^c = (\mathbf{\Phi}^1, \mathbf{\Phi}^2, \cdots), \qquad \mathbf{\Phi}^n = (\Phi_1^n, \Phi_2^n, \Phi_3^n, \Phi_4^n). \tag{A.31}$$

This transformation then brings the Hamiltonian of the boundary constrained topological insulator film $H_{TI}(\mathbf{k}, -i\partial_z)$ into the direct sum form of Dirac fermions by

$$H(\mathbf{k}) = \int \mathrm{d}z \, (U^c)^\dagger (\mathbf{k}, z) H_{TI}(\mathbf{k}, -i\partial_z) U^c(\mathbf{k}, z). \tag{A.32}$$

### A.3 Analytic expression for mass term

The proof has been posted separately [33], and here is a repetition. Analytic expression for effective mass $m(\mathbf{k})$ is obtained in the $L \to \infty$ case as a thick limit, however, notice that finite-size correction to $m(\mathbf{k})$ decays exponentially with thickness [38], our proof here is suitable even for a thin film. Closed $E - \xi$ equations are

$$\begin{cases} \xi_1^2 + \xi_2^2 = \dfrac{2m_0(\mathbf{k})t_\perp - \lambda_\perp^2}{t_\perp^2}, \\[2mm] \xi_1^2 \xi_2^2 = \dfrac{m_0(\mathbf{k})^2 - E^2}{t_\perp^2}, \\[2mm] E = m_0(\mathbf{k}) - t_\perp \dfrac{\xi_1^2 g^+(\xi_1) - \xi_2^2 g^+(\xi_2)}{g^+(\xi_1) - g^+(\xi_2)}, \end{cases} \tag{A.33}$$

where $g^+(\xi) = \tan(\xi L/2)/\xi$. We shall assume $\lambda_\perp > 0$, $t_\perp > 0$ in the following discussion, without losing generality, and $m_0(\mathbf{k})$ controls the expression form.

The classification on $\tan(\xi L/2)$ leads to

$$\lim_{L \to +\infty} \tan(\xi L/2) = \begin{cases} i, & \mathrm{Im}(\xi) > 0, \\ \mathrm{N.A.}, & \mathrm{Im}(\xi) = 0, \\ -i, & \mathrm{Im}(\xi) < 0. \end{cases} \tag{A.34}$$

And three basic cases are separated as

$$\begin{cases} \mathrm{Im}(\xi_1) > 0 > \mathrm{Im}(\xi_2), \\ \mathrm{Im}(\xi_{1,2}) > 0, \\ \mathrm{Im}(\xi_1) = 0, \quad \mathrm{Im}(\xi_2) > 0, \end{cases} \tag{A.35}$$

while other cases could be obtained similarly.

**Case I.** ($\mathrm{Im}(\xi_1) > 0 > \mathrm{Im}(\xi_2)$)

Now $\tan(\xi_1 L/2) = i = -\tan(\xi_2 L/2)$ ($L \to +\infty$ ignored), and

$$\begin{cases} \xi_1^2 + \xi_2^2 = \dfrac{2m_0(\mathbf{k})t_\perp - \lambda_\perp^2}{t_\perp^2}, \\[2mm] \xi_1^2 \xi_2^2 = \dfrac{m_0(\mathbf{k})^2 - E^2}{t_\perp^2}, \\[2mm] E = m_0(\mathbf{k}) - t_\perp \xi_1 \xi_2, \end{cases} \tag{A.36}$$

where the second and third equations lead to

$$m_0(\mathbf{k})^2 - E^2 = (m_0(\mathbf{k}) - E)^2, \tag{A.37}$$

which offers two possible solutions $E = 0$ or $E = m_0(\boldsymbol{k})$.

    **I.** ($E = 0$) This leads to

$$
\begin{cases}
\xi_1 \xi_2 = \dfrac{m_0(\boldsymbol{k})}{t_\perp}, \\[2mm]
\xi_1^2 + \xi_2^2 = \dfrac{2m_0(\boldsymbol{k})t_\perp - \lambda_\perp^2}{t_\perp^2}.
\end{cases}
\tag{A.38}
$$

Requiring $\mathrm{Im}(\xi_1) > 0 > \mathrm{Im}(\xi_2)$ then gives

$$
\begin{cases}
\xi_1 + \xi_2 = \begin{cases} 2u\sqrt{4\gamma - 1}, & \gamma > 1/4, \\ 2ui\sqrt{1 - 4\gamma}, & \gamma < 1/4, \end{cases} \\[4mm]
\xi_1 - \xi_2 = 2ui,
\end{cases}
\tag{A.39}
$$

$$
\begin{cases}
\gamma = m_0(\boldsymbol{k})t_\perp / \lambda_\perp^2, \\[1mm]
u = \lambda_\perp / 2t_\perp,
\end{cases}
\tag{A.40}
$$

which offers:

    • $\gamma > 1/4$:

$$
\begin{cases}
\xi_1 = u(\sqrt{4\gamma - 1} + i), \\
\xi_2 = u(\sqrt{4\gamma - 1} - i),
\end{cases}
\tag{A.41}
$$

    • $\gamma < 1/4$:

$$
\begin{cases}
\xi_1 = iu(\sqrt{1 - 4\gamma} + 1), \\
\xi_2 = iu(\sqrt{1 - 4\gamma} - 1).
\end{cases}
\tag{A.42}
$$

The latter condition stands only when $\gamma > 0$ as for $\mathrm{Im}(\xi_2) < 0$.

    **II.** ($E = m_0(\boldsymbol{k})$) This leads to

$$
\begin{cases}
\xi_1 \xi_2 = 0, \\[2mm]
\xi_1^2 + \xi_2^2 = \dfrac{2m_0(\boldsymbol{k})t_\perp - \lambda_\perp^2}{t_\perp^2},
\end{cases}
\tag{A.43}
$$

and one of $\xi_\alpha = 0$ is unavoidable, which fails the precondition and is abandoned, i.e., $E = m_0(\boldsymbol{k})$ is not a solution in the case.

**Case II.** ($\mathrm{Im}(\xi_{1,2}) > 0$)

    Now $\tan(\xi_1 L/2) = i = \tan(\xi_2 L/2)$ ($L \to +\infty$ ignored), and

$$
\begin{cases}
\xi_1^2 + \xi_2^2 = \dfrac{2m_0(\boldsymbol{k})t_\perp - \lambda_\perp^2}{t_\perp^2}, \\[3mm]
\xi_1^2 \xi_2^2 = \dfrac{m_0(\boldsymbol{k})^2 - E^2}{t_\perp^2}, \\[3mm]
E = m_0(\boldsymbol{k}) + t_\perp \xi_1 \xi_2,
\end{cases}
\tag{A.44}
$$

then the second and third equations above leads to

$$
m_0(\boldsymbol{k})^2 - E^2 = (m_0(\boldsymbol{k}) - E)^2,
\tag{A.45}
$$

which gives us two possible solutions as $E = 0$ or $E = m_0(\boldsymbol{k})$.

**I.** ($E = 0$) This condition leads to

$$\begin{cases} \xi_1\xi_2 = -\dfrac{m_0(\boldsymbol{k})}{t_\perp}, \\ \xi_1^2 + \xi_2^2 = \dfrac{2m_0(\boldsymbol{k})t_\perp - \lambda_\perp^2}{t_\perp^2}. \end{cases} \tag{A.46}$$

Requirement $\text{Im}(\xi_{1,2}) > 0$ then gives

$$\begin{cases} \xi_1 + \xi_2 = 2ui, \\ \xi_1 - \xi_2 = \begin{cases} 2u\sqrt{4\gamma - 1}, & \gamma > 1/4, \\ 2ui\sqrt{1 - 4\gamma}, & \gamma < 1/4, \end{cases} \end{cases} \tag{A.47}$$

which offers:
- $\gamma > 1/4$:

$$\begin{cases} \xi_1 = u(\sqrt{4\gamma - 1} + i), \\ \xi_2 = u(-\sqrt{4\gamma - 1} + i), \end{cases} \tag{A.48}$$

- $\gamma < 1/4$:

$$\begin{cases} \xi_1 = iu(\sqrt{1 - 4\gamma} + 1), \\ \xi_2 = iu(-\sqrt{1 - 4\gamma} + 1). \end{cases} \tag{A.49}$$

The latter condition stands only when $\gamma > 0$ as for $\text{Im}(\xi_2) > 0$.

**II.** ($E = m_0(\boldsymbol{k})$) This leads to

$$\begin{cases} \xi_1\xi_2 = 0, \\ \xi_1^2 + \xi_2^2 = \dfrac{2m_0(\boldsymbol{k})t_\perp - \lambda_\perp^2}{t_\perp^2}, \end{cases} \tag{A.50}$$

and again one of $\xi_\alpha = 0$ is unavoidable, and one concludes $E = m_0(\boldsymbol{k})$ is not a solution in the case.

**Case III.** ($\text{Im}(\xi_1) = 0, \ \text{Im}(\xi_2) > 0$)

By guessing $E = m_0(\boldsymbol{k})$, we have

$$\begin{cases} \xi_1\xi_2 = 0, \\ \xi_1^2 + \xi_2^2 = \dfrac{2m_0(\boldsymbol{k})t_\perp - \lambda_\perp^2}{t_\perp^2}, \end{cases} \tag{A.51}$$

which gives

$$\begin{cases} (\xi_1 + \xi_2)^2 = 4u^2(2\gamma - 1), \\ (\xi_1 - \xi_2)^2 = 4u^2(2\gamma - 1), \end{cases} \tag{A.52}$$

and choosing

$$\begin{cases} \xi_1 = 0, \\ \xi_2 = 2ui\sqrt{1 - 2\gamma}, \end{cases} \tag{A.53}$$

fulfills the requirement. Notice that $\gamma < 1/2$ is assumed, which should not bother the self-consistent solution. Meanwhile, since $\xi_1 = 0$ leads to degenerate eigenvalue $\pm\xi_1$, then one should generally assume another solution as

$$(A + Bz)e^{i\xi_1 z}\phi\big|_{\xi_1 = 0, E = m_0(\boldsymbol{k})},$$

which, however, only gives result that $B = 0$ while $A$ is arbitrary, which passes no additional information.

Retrospecting the definition $\gamma = m_0(\boldsymbol{k})t_\perp/\lambda_\perp^2$, the discussion above naturally leads to the conclusion that the lowest eigenenergy of $H_{1d}$ reads

$$E = \begin{cases} 0, & m_0(\boldsymbol{k}) > 0, \\ m_0(\boldsymbol{k}), & m_0(\boldsymbol{k}) < 0, \end{cases} \tag{A.54}$$

or by re-defining lowest $E(\boldsymbol{k})$ as $m_1(\boldsymbol{k})$, we write

$$m_1(\boldsymbol{k}) = \Theta(-m_0(\boldsymbol{k}))m_0(\boldsymbol{k}),$$

as result mention in Eq. (11).

## A.4 Finite-size correction to mass term

We could in fact conserve the lowest order correction to see the finite size gap when $L$ is not that large. For $\xi_1$ and $\xi_2$, one could approximately get lowest order correction for $\tan(\xi L/2)$ by treating $\beta^{\pm L/2}$ as small quantity (depend on sign of $\text{Im}(\xi)$)

$$\tan(\xi L/2) \approx \begin{cases} i(1 - 2\beta^L), & \text{Im}(\xi) > 0, \\ -i(1 - 2\beta^{-L}), & \text{Im}(\xi) < 0. \end{cases} \tag{A.55}$$

Also notice that from the original $E - \xi$ equation

$$E^2 - (m_0(\boldsymbol{k}) - t_\perp \xi^2)^2 - \lambda_\perp^2 \xi^2 = 0,$$

which could be further split into (when $E = 0$ as zeroth-order)

$$t_\perp \xi^2 \pm i\lambda_\perp \xi - m_0(k) = 0, \tag{A.56}$$

one solves

$$\xi = \frac{s_1 i\lambda_\perp + s_2\sqrt{4m_0(k)t_\perp - \lambda_\perp^2}}{2t_\perp} = u(is_1 + s_2\sqrt{4\gamma - 1}), \tag{A.57}$$

where $s_1, s_2 = \pm$ without restriction. Notice that in real calculation, one needs to specify which branch $\xi_{1,2}$ lie in, but such choice will not affect the final result as long as chosen $\xi_{1,2}$ satisfy zeroth-order solution. Now again we have two cases below:

• $\gamma > 1/4$, we choose

$$\begin{cases} \xi_1 = \xi_2^*, \\ \text{Im}(\xi_1) > 0 > \text{Im}(\xi_2), \\ \text{Re}(\xi_1) = \text{Re}(\xi_2) > 0, \end{cases} \tag{A.58}$$

as main branch condition, then

$$\begin{cases} \tan(\xi_1 L/2) \approx i(1 - 2\beta_1^L), \\ \tan(\xi_2 L/2) \approx -i(1 - 2\beta_2^{-L}), \end{cases} \tag{A.59}$$

and

$$E(k) \approx (m_0(k) - t_\perp \xi_1 \xi_2) + 2t_\perp \xi_1 \xi_2 \frac{\xi_1 - \xi_2}{\xi_1 + \xi_2}(e^{i\xi_1 L} - e^{-i\xi_2 L}).$$

Notice that first term in bracket is zeroth order as $E \approx 0$. Now, it is time to utilize four solutions in Eq. (A.57). By main branch condition above, accordingly we choose

$$\begin{cases} \xi_1 = u(\sqrt{4\gamma - 1} + i), \\ \xi_2 = u(\sqrt{4\gamma - 1} - i), \end{cases} \tag{A.60}$$

considering that $\gamma > 1/4$ in this zone. Afterwards, one obtains

$$E(k) \approx -\frac{4m_0(k)}{\sqrt{4\gamma - 1}} \sin\left(u\sqrt{4\gamma - 1}L\right)e^{-uL}. \tag{A.61}$$

Low energy surface state mass shows both exponentially decay and oscillating behavior.
• $0 < \gamma < 1/4$, we choose

$$\begin{cases} \text{Im}(\xi_1) > 0, \\ \text{Im}(\xi_2) > 0, \end{cases} \tag{A.62}$$

as main branch condition, then

$$\begin{cases} \tan(\xi_1 L/2) \approx i(1 - 2\beta_1^L), \\ \tan(\xi_2 L/2) \approx i(1 - 2\beta_2^L), \end{cases} \tag{A.63}$$

$$E(k) \approx (m_0(k) + t_\perp \xi_1 \xi_2) - 2t_\perp \xi_1 \xi_2 \frac{\xi_1 + \xi_2}{\xi_1 - \xi_2}(e^{i\xi_1 L} - e^{i\xi_2 L}),$$

where first term in bracket is again zeroth order energy approaching zero. Again, utilizing four solutions in Eq. (A.57) with main branch condition above, we choose

$$\begin{cases} \xi_1 = iu(1 + \sqrt{1 - 4\gamma}), \\ \xi_2 = iu(1 - \sqrt{1 - 4\gamma}), \end{cases} \tag{A.64}$$

considering that $0 < \gamma < 1/4$ in this zone. Again, one obtains

$$E(k) \approx -\frac{4m_0(k)}{\sqrt{1 - 4\gamma}} \sinh\left(u\sqrt{1 - 4\gamma}L\right)e^{-uL}. \tag{A.65}$$

Since $\sin(ix) = i\sinh(x)$, and by $\gamma = m_0(\boldsymbol{k})t_\perp/\lambda_\perp^2$, we may set $\gamma(k_c) = 0$ and obtain a unified expression for lowest order mass correction

$$E(k < k_c) = -\frac{4m_0(k)}{\sqrt{4\gamma - 1}} \sin\left(u\sqrt{4\gamma - 1}L\right)e^{-uL}. \tag{A.66}$$

However, as a comment, in numerical calculation, $E$ in zone $0 < \gamma < 1/4$ is suppressed into zero in a much slower manner, which is caused by exponential cancellation between sinh and exp. Nevertheless, since $\sqrt{1 - 4\gamma} < 1$ in the region, we conclude that the exponential increasing is always slower than the decaying, which finally pushes the state to zero energy for $L \to +\infty$.

# B  Derivation of Eq. (1b)

To obtain an effective model, we start from solving $\mathcal{H}_{1d}$ and notice that $[\mathcal{H}_{1d}(\boldsymbol{k}), \sigma_z] = 0$, from which we could let

$$\mathcal{H}_{1d}(\boldsymbol{k})\zeta_s \otimes |\phi^s(\boldsymbol{k})\rangle = \zeta_s \otimes \mathcal{H}_{1d}^s(\boldsymbol{k})|\phi^s(\boldsymbol{k})\rangle, \tag{B.1}$$

where $\mathcal{H}_{1d}^s(\boldsymbol{k})$ is split Hamiltonian that only acts on one subspace, and by definition

$$\sigma_z \zeta_s = s\zeta_s, \quad s = \pm. \tag{B.2}$$

Under basis of $\{\Psi_{l_z,\boldsymbol{k}}\}_{l_z}$, $\mathcal{H}^s_{1d}(\boldsymbol{k})$ is in its matrix form denoted as $H^s_{1d}(\boldsymbol{k})$, with solution defined from its eigenvalue equation

$$H^s_{1d}(\boldsymbol{k})\phi^s(\boldsymbol{k}) = E^s(\boldsymbol{k})\phi^s(\boldsymbol{k}), \qquad \phi^s(\boldsymbol{k}) = \oplus_{l_z}\phi^s_{l_z}(\boldsymbol{k}). \tag{B.3}$$

To make discussion pithy, we shall omit $s, \boldsymbol{k}$ and let $M \equiv M_0(\boldsymbol{k})$ below in the section.

Eq. (B.3) can be written in th e recurrence form as

$$\left(t_\perp \tau_z + i\frac{\lambda_\perp}{2}s\tau_x\right)\phi_{l_z-1} + M\tau_z\phi_{l_z} + \left(t_\perp \tau_z - i\frac{\lambda_\perp}{2}s\tau_x\right)\phi_{l_z+1} = E\phi_{l_z}, \tag{B.4}$$

by observing which could we set trial function as $\phi_{l_z} = e^{i\xi l_z}\phi = \beta^{l_z}\phi$ where $\beta = e^{i\xi}$. Then accordingly the equation is reduced to

$$\left[\left(t_\perp \tau_z + i\frac{\lambda_\perp}{2}s\tau_x\right)\beta^{-1} + (M\tau_z - E) + \left(t_\perp \tau_z - i\frac{\lambda_\perp}{2}s\tau_x\right)\beta\right]\phi = 0, \tag{B.5}$$

which firstly leads to

$$E^2 = (M + 2t_\perp \cos\xi)^2 + \lambda_\perp^2 \sin^2\xi, \tag{B.6}$$

requiring non-trivial $\phi$. From Eq. (B.6) one solves

$$\begin{cases} \cos\xi^p_\alpha = \dfrac{-Mt_\perp + (-1)^{\alpha-1}\sqrt{M^2t_\perp^2 - (t_\perp^2 - \lambda_\perp^2/4)(M^2 + \lambda_\perp^2 - E^2)}}{2(t_\perp^2 - \lambda_\perp^2/4)}, \\ \sin\xi^p_\alpha = p\sqrt{1 - \cos^2\xi_\alpha}, \quad p = \pm, \quad \alpha = 1,2, \end{cases} \tag{B.7}$$

which tells that

$$\beta^p_\alpha = e^{i\xi^p_\alpha} = \cos\xi_\alpha + ip\sqrt{1 - \cos^2\xi_\alpha}. \tag{B.8}$$

Here one thing to notice is that the sign change of $\sin\xi^p_\alpha$ is caused by sign change of $\xi$, rather than a phase shift like $\xi \to \xi + \pi$, since the latter will lead to the sign change of $\cos\xi$, too, and that is not our solution.

To make maximum utilization of the symmetry, we consider canonical boundary condition in which the centre of 1-d chain sits at $z = 0$, then by denoting $l = L_z + 1$, we would have

$$\phi^s\left(\pm\frac{l}{2}\right) = 0, \tag{B.9}$$

and it is essential to notice that sites $l_z = \pm\dfrac{L_z + 1}{2}$ are two fictitious points where the constraints take place, and true lattice stops at $l_z = \pm\dfrac{L_z - 1}{2}$ as we only have $L_z$ sites. What is more, for compensation of unifying expression regardless of odevity of $L_z$, $l_z$ would be forced to choose different ways to be taken out as follows

$$\begin{cases} l_z = 0, \pm1, \pm2, \ldots, \pm\dfrac{L_z + 1}{2}, & \text{for } L_z \text{ odd}, \\ l_z = \pm\dfrac{1}{2}, \pm\dfrac{3}{2}, \ldots, \pm\dfrac{L_z + 1}{2}, & \text{for } L_z \text{ even}, \end{cases} \tag{B.10}$$

which conforms mirror symmetry to $z = 0$. Afterwards, enlightened by the idea of symmetric trial functions, we also build several functions from $\beta^p_\alpha$ considering the symmetric case stated above. Denote

$$\begin{cases} E(\beta, l_z) = \dfrac{\beta^{l_z} + \beta^{-l_z}}{\beta^{(L_z+1)/2} + \beta^{-(L_z+1)/2}} = \dfrac{\cos(\xi l_z)}{\cos(\xi l/2)}, \\ O(\beta, l_z) = \dfrac{\beta^{l_z} - \beta^{-l_z}}{\beta^{(L_z+1)/2} - \beta^{-(L_z+1)/2}} = \dfrac{\sin(\xi l_z)}{\sin(\xi l/2)}, \end{cases} \tag{B.11}$$

where '$E$' and '$O$', namely even and odd, represent the parity of two functions about $z$, and one should not identify $E$ here as the energy function. From which we establish two sets of factors respecting boundary condition with even or odd parity

$$\begin{cases} f_+(l_z) = \sum_\alpha (-1)^{\alpha-1} E(\beta_\alpha^p, l_z), \\ f_-(l_z) = \sum_\alpha (-1)^{\alpha-1} O(\beta_\alpha^p, l_z), \end{cases} \tag{B.12}$$

where the summation is over $\alpha$ but without $p$ since it only changes sign of $\xi$ and thus does not influence the value of $E$ or $O$. Before proceeding, let us find some special properties about those functions or factors. Let

$$\begin{cases} a = \beta + \dfrac{1}{\beta} = 2\cos\xi, \\ b = \beta - \dfrac{1}{\beta} = 2i\sin\xi, \end{cases} \tag{B.13}$$

who weight as the *lattice differential operators* that lead to relation

$$f_+(l_z \pm 1) = \sum_\alpha (-1)^{\alpha-1} \frac{a_\alpha E(\beta_\alpha, l_z) \pm i b_\alpha \tan(\xi_\alpha l/2) O(\beta_\alpha, l_z)}{2} \equiv g_\pm, \tag{B.14a}$$

$$f_-(l_z \pm 1) = \sum_\alpha (-1)^{\alpha-1} \frac{a_\alpha O(\beta_\alpha, l_z) \mp i b_\alpha \cot(\xi_\alpha l/2) E(\beta_\alpha, l_z)}{2} \equiv h_\pm. \tag{B.14b}$$

One could again see that the iteration relation is also independent of $p$ within our expectation.

Now we are able to come back and solve the chain problem. Let

$$\phi_{l_z} = c f_+(l_z) + d f_-(l_z), \tag{B.15}$$

to be guessed general solution confined by boundary condition. Bring this trial solution into Eq. (B.4) and requiring vanishing coefficients of $E(\beta_\alpha, l_z)$ and $O(\beta_\alpha, l_z)$, one obtains, after re-organization,

$$\begin{cases} (M - E + t_\perp a_\alpha) c_1 - \dfrac{\lambda_\perp}{2} s d_2 b_\alpha \cot(\xi_\alpha l/2) = 0, \\ -\dfrac{\lambda_\perp}{2} s c_1 b_\alpha \tan(\xi_\alpha l/2) + (M + E + t_\perp a_\alpha) d_2 = 0, \end{cases} \tag{B.16a}$$

$$\begin{cases} (M - E + t_\perp a_\alpha) d_1 + \dfrac{\lambda_\perp}{2} s c_2 b_\alpha \tan(\xi_\alpha l/2) = 0, \\ \dfrac{\lambda_\perp}{2} s d_1 b_\alpha \cot(\xi_\alpha l/2) + (M + E + t_\perp a_\alpha) c_2 = 0, \end{cases} \tag{B.16b}$$

for different $\alpha$. Requiring simultaneous standing with respect to $\alpha$ leads to four solutions in pairs

$$\begin{cases} d_2 = \dfrac{i t_\perp \eta_1}{s \lambda_\perp} c_1, & E = E_+, \\ c_1 = \dfrac{i t_\perp \eta_2}{s \lambda_\perp} d_2, & E = -E_-, \end{cases} \qquad \begin{cases} c_2 = -\dfrac{i t_\perp \eta_2}{s \lambda_\perp} d_1, & E = E_-, \\ d_1 = -\dfrac{i t_\perp \eta_1}{s \lambda_\perp} c_2, & E = -E_+, \end{cases} \tag{B.17}$$

where the formal expression for energies are

$$E_\pm = M + 2 t_\perp \frac{\cos\xi_1 g^\pm(\xi_1) - \cos\xi_2 g^\pm(\xi_2)}{g^\pm(\xi_1) - g^\pm(\xi_2)}, \tag{B.18}$$

with two defined functions

$$g^\pm(\xi) = \frac{\tan^{\pm1}(\xi(L_z + 1)/2)}{\sin\xi}, \tag{B.19}$$

and two dimensionless factors

$$\begin{cases} \eta_1 = \dfrac{-2(\cos\xi_1 - \cos\xi_2)}{\sin\xi_1\cot(\xi_1 l/2) - \sin\xi_2\cot(\xi_2 l/2)}, \\ \eta_2 = \dfrac{-2(\cos\xi_1 - \cos\xi_2)}{\sin\xi_1\tan(\xi_1 l/2) - \sin\xi_2\tan(\xi_2 l/2)}, \end{cases} \tag{B.20}$$

have been introduced. From the above discussion we seemingly have four solutions, mathematical restriction, however, tells that equations in Eq. (B.16) in the same brace must stand simultaneously, which then gives us two relations as

$$\begin{cases} 1 = \left| \dfrac{it_\perp \eta_1}{s\lambda_\perp} \cdot \dfrac{it_\perp \eta_2}{s\lambda_\perp} \right| \Longrightarrow |\eta_1\eta_2| = \dfrac{\lambda_\perp^2}{t_\perp^2}, \\ m \equiv E_+ = -E_-, \end{cases} \tag{B.21}$$

and the latter one is also a physical result from Dirac equation. This reduces our four solutions to two independent ones for each $s$. The above discussion is equivalent to requiring simultaneous standing of equations in left brace of Eq. (B.16)

$$E^2 = (M + 2t_\perp \cos\xi_\alpha)^2 + \lambda_\perp^2 \sin^2\xi_\alpha,$$

which is independent of $\alpha$ and matches the result of Eq. (B.6).

Similar arguments can be made here as in the continuum model. Counting on complexity of $\xi_{1,2}$ restricted by Eq. (B.7) and the property of trigonometric/hyperbolic function leads to the conclusion that quadratic form $f_+^* f_-$ and $\eta$ (at certain $(\boldsymbol{k}, z, E)$) are always real. Essentially, $f_\pm$ are either real or purely imaginary.

In short, what we need solving to get all energy states $m$ are the simultaneous equations below

$$m = M + 2t_\perp \frac{\cos\xi_1 g(\xi_1) - \cos\xi_2 g(\xi_2)}{g(\xi_1) - g(\xi_2)}, \tag{B.22a}$$

$$\cos\xi_\alpha = \frac{-Mt_\perp + (-1)^{\alpha-1}\sqrt{M^2 t_\perp^2 - (t_\perp^2 - \lambda_\perp^2/4)(M^2 + \lambda_\perp^2 - m^2)}}{2(t_\perp^2 - \lambda_\perp^2/4)}, \tag{B.22b}$$

where

$$\begin{cases} M = M_0(\boldsymbol{k}) = m_0 - 4t_\parallel \left( \sin^2\dfrac{k_x a}{2} + \sin^2\dfrac{k_y b}{2} \right) - 2t_\perp, \\ g(\xi) = \dfrac{\tan(\xi(L_z + 1))/2}{\sin\xi}, \end{cases} \tag{B.23}$$

and sign of $\xi$ is fixed by $p = +$ so that

$$\sin\xi_\alpha = \sqrt{1 - \cos\xi_\alpha^2}, \quad \alpha = 1, 2. \tag{B.24}$$

Basically, there are three variables $\xi_1, \xi_2$ and $m$, together with three equations above, then it is in a sense some *exact system of equations* but a non-linear transcendental version. From this set of equations, one may expect $L_z$ solutions $m_n(\boldsymbol{k}), n = 1, 2, \cdots, L_z$ including one surface state and $L_z - 1$ purely trivial bulk states, if within suitable choice of parameters. And the other set of $L_z$ solutions are just chiral partners with $-m_n(\boldsymbol{k})$. Notice that these $2L_z$ solutions compose eigenvalues for one $H_{1d}^s$, then by counting $s = \pm$ there are in fact $4L_z$ solutions in total, which is expected from the matrix form of $H_{1d}$.

Here it comes to construct basis for projection, we firstly ignore lower index for $m$ since our wavefunction solution form is universal whatever $n$ takes. Then by counting $s$, we totally have four independent solutions for each $m$ as follows

$$\begin{cases} \varphi(s) = \begin{pmatrix} c_1 f_+ \\ d_2 f_- \end{pmatrix} = C \begin{pmatrix} -is\lambda_\perp f_+ \\ t_\perp \eta f_- \end{pmatrix}, & E = m, \\[4mm] \chi(s) = \begin{pmatrix} d_1 f_- \\ c_2 f_+ \end{pmatrix} = C \begin{pmatrix} t_\perp \eta f_- \\ is\lambda_\perp f_+ \end{pmatrix}, & E = -m, \end{cases} \tag{B.25}$$

where we have ignored lower index of $\eta_1$, and the norm $C$ is the same for $\varphi$ and $\chi$ states. Then restoring $n$-indices we have $4L_z$ basis in certain sequence as

$$\Phi_1^n = \zeta_+ \otimes \varphi(+) = \begin{pmatrix} \varphi^n(+) \\ 0 \end{pmatrix}, \qquad \Phi_2^n = \begin{pmatrix} 0 \\ \chi^n(-) \end{pmatrix},$$
$$\Phi_3^n = \begin{pmatrix} \chi^n(+) \\ 0 \end{pmatrix}, \qquad \Phi_4^n = \begin{pmatrix} 0 \\ \varphi^n(-) \end{pmatrix}, \tag{B.26}$$

with energies $(m_n(\boldsymbol{k}), -m_n(\boldsymbol{k}), -m_n(\boldsymbol{k}), m_n(\boldsymbol{k}))$, respectively. The $(\boldsymbol{k}, l_z)$ dependence of these basis states are inherited from functions $f_\pm^n(\boldsymbol{k}, l_z)$ and factor $\eta^n(\boldsymbol{k})$.

The basis here shares the same symmetry analysis as within the continuum model, while here the parity and mirror symmetries can be written down explicitly in the off-diagonal matrix form, with $\sigma_0 \tau_z$ and $-i\sigma_z \tau_z$ as the anti-diagonal elements, respectively. And especially, by combining the mirror and spin-$z$ index, we assign $\Phi_i^n = \Phi_{\chi,s}^n$ with

$$\begin{aligned} \Phi_{++}^n &= \Phi_1^n, & \Phi_{+-}^n &= \Phi_2^n, \\ \Phi_{-+}^n &= \Phi_3^n, & \Phi_{--}^n &= \Phi_4^n. \end{aligned} \tag{B.27}$$

Now we turn to the projection, which is formally

$$\langle \Phi | H_{\text{Film}} | \Phi \rangle = \langle \Phi | H_{1d} | \Phi \rangle + \langle \Phi | H_\parallel | \Phi \rangle, \tag{B.28}$$

where the first part, by the definition of eigenvalue equation, is just

$$\oplus_n \text{diag}(m_n, -m_n, -m_n, m_n) = \oplus_n m_n(\boldsymbol{k}) \tau_z \sigma_z,$$

while in the second part, since $H_\parallel = \lambda_\parallel (\sin(k_x a)\sigma_x \tau_x + \sin(k_y b)\sigma_y \tau_x)$ is purely off diagonal, it is easy to conclude that

$$\langle \Phi_i^n | H_\parallel | \Phi_i^{n'} \rangle = 0, \quad i = 1,2,3,4,$$
$$\langle \Phi_1^n | H_\parallel | \Phi_3^{n'} \rangle = 0 = \langle \Phi_2^n | H_\parallel | \Phi_4^{n'} \rangle.$$

Then only four terms need consideration by hermicity, among which

$$\langle \Phi_1^n | H_\parallel | \Phi_4^{n'} \rangle = \lambda_\parallel (\sin(k_x a) - i\sin(k_y b)) \sum_{l_z} |C|^2 i\lambda_\perp t_\perp [\eta^n (f_+^n)^* f_-^{n'} + \eta^{n'} (f_-^n)^* f_+^{n'}] = 0,$$

$$\langle \Phi_2^n | H_\parallel | \Phi_3^{n'} \rangle = \lambda_\parallel (\sin(k_x a) - i\sin(k_y b)) \sum_{l_z} |C|^2 i\lambda_\perp t_\perp [\eta^n (f_-^n)^* f_+^{n'} + \eta^{n'} (f_+^n)^* f_-^{n'}] = 0,$$

as $f_- f_+$ is odd to $z$. Then, the only remaining terms are

$$\langle \Phi_1^n | H_\parallel | \Phi_2^{n'} \rangle = \lambda_\parallel (\sin(k_x a) - i\sin(k_y b))\delta_{nn'} = \langle \Phi_3^n | H_\parallel | \Phi_4^{n'} \rangle,$$

where normalization condition is used. Finally we arrive at the equivalent Hamiltonian

$$H(\boldsymbol{k}) = \bigoplus_{n=1}^{L_z} \left[ \lambda_\parallel (\sin(k_x a)\sigma_x + \sin(k_y b)\sigma_y) + m_n(\boldsymbol{k})\tau_z \sigma_z \right] = \bigoplus_{n,\chi} h_{n,\chi}(\boldsymbol{k}), \tag{B.29}$$

where unspecified degrees of freedom are all identity matrix. And hereto we have successfully arrived at Eq. (1b) in the main text. Also notice that $H$ is exactly equivalent to original $H_{\text{Film}}$, since by counting all $n$, the projection we did is just a unitary basis transformation, where the unitary matrix is composed of solutions of $H_{1d}$.

The projection here is also a unitary transformation, which shares a simpler form than that in the continuum model. Since now the original Hamiltonian reads

$$\mathcal{H}_{\text{Film}}(\boldsymbol{k}) = \sum_{l_z, l_z'} \Psi_{l_z}^\dagger H_{\text{Film}}(\boldsymbol{k}, l_z, l_z') \Psi_{l_z'}, \tag{B.30}$$

then by defining $\Psi = \oplus_{l_z} \Psi_{l_z}$, we identify the unitary transformation as

$$\mathcal{H}_{\text{Film}}(\boldsymbol{k}) = (\Psi^\dagger U^l)\big[(U^l)^\dagger H_{\text{Film}}(\boldsymbol{k}) U^l\big]((U^l)^\dagger \Psi), \tag{B.31}$$

where

$$U^l = (\boldsymbol{\Phi}^1, \boldsymbol{\Phi}^2, \cdots, \boldsymbol{\Phi}^{L_z}), \qquad \boldsymbol{\Phi}^n = (\Phi_1^n, \Phi_2^n, \Phi_3^n, \Phi_4^n), \tag{B.32}$$

and we recognize $\Phi_i^n = \oplus_{l_z} \Phi_i^n(l_z)$ here so that $U^l$ is a $4L_z \times 4L_z$ unitary matrix. And here again $U^l$ is trivial in $k$-space. The core transformation on matrix form of Hamiltonian gives rise to

$$H(\boldsymbol{k}) = (U^l(\boldsymbol{k}))^\dagger H_{\text{Film}}(\boldsymbol{k}) U^l(\boldsymbol{k}), \tag{B.33}$$

while the inverse transformation $(U^l)^\dagger \Psi$ assigns composed Fermionic operators to the new basis. Essentially, the transformation to each $h_{n,\chi}$ is done by

$$h_{n,\chi} = (U_{n,\chi}^l)^\dagger H_{\text{Film}} U_{n,\chi}^l, \tag{B.34}$$

where

$$U_{n,\chi}^l = \boldsymbol{\Phi}_\chi^n = (\Phi_{\chi,s=+}^n, \Phi_{\chi,s=-}^n), \tag{B.35}$$

is a $2L_z \times 2$ matrix.

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
