# Peer review of "Dirac Fermions and Topological Phases in Magnetic Topological Insulator Films"

_SciPost Physics, doi:SciPost Phys. 17, 146 (2024)_

## Round 1 · Referee Report · Anonymous (Referee 1) · 2024-7-24

Report

In this paper, the authors explore diverse topological phases using a simple Dirac band model, incorporating symmetry, mass, and magnetization considerations. The referee finds most results reasonable but notes the analysis resembles the widely used four-band model for topological materials. The predictability of the model is questioned due to a lack of clear parameter estimation methods. To strengthen the work, the authors should distinguish their model from the four-band model and link their discussions to real materials beyond MBT. The paper is well-written but contains typos. Addressing these issues would make it suitable for publication in SciPost Physics.

Recommendation

Publish (easily meets expectations and criteria for this Journal; among top 50%)

  • validity: high
  • significance: good
  • originality: good
  • clarity: high
  • formatting: excellent
  • grammar: excellent

Author:  Kai-Zhi Bai  on 2024-08-23  [id 4714]

(in reply to Report 1 on 2024-07-24)

Reply to the report:

We agree that the manuscript lacks parameter estimation methods. The performed calculations are based on parameters from the four-band model of Bi$_2$Se$_3$ (https://doi.org/10.1038/nphys1270), since we hope to transmit a general picture of how Dirac fermions and different topological phases emerge in a strong TI with magnetization. Certainly, our calculations are a model-base analysis, but the model is extensively used in the field of study, and for a quantitative case-by-case study which needs an experimentally accessible parameter estimation, we suggest parameter fittings through a first principles calculation. We add this point after taking specific parameters in the caption of Fig. 2.

Following the suggestion, we have added discussions on the relation and difference between our model and previous four-band model. In short, starting from the minimal model for bulk 3D TI, which is a 4-by-4 modified Dirac equation model, we solve the 1D model along the $z$-direction exactly in a film geometry, and use the complete set of exact solutions to mapping the 3D model on the 2D Brillouin zone, obtaining a series of 2D Dirac fermions in the full 2D Brillouin zone. Then with magnetism, the lowest four-band model can share the same Dirac matrices with the previous four-band model describing surface states of TI film, under a change of representation. However, in our derivation, the Dirac mass has a zero plateau at low-energy and is non-vanishing at high-energy. This is distinct from all the previous models. This property plays a key role in the realization of the plateau of the half-quantized Hall effect of the gapless Dirac cone in several related phases. Please see Section 4.1.2 ‘Weak Zeeman field’, Section 7 ‘Discussion and conclusion’ for the detailed modifications.

We also agree that more materials need to be linked with the model discussion. In the revised version, we point the materials out, like Cr and V doped (Bi, Sb)$_2$Te$_3$ in realizing QAHE, axion insulator and half-QAHE. For the half quantum mirror Hall effect, we have suggested in a recent paper (https://doi.org/10.1038/s41467-024-51215-x) to utilize Bi$_2$(Se, Te)$_3$ along [100] direction or along [001] direction with a twin boundary as realistic candidates. For phases that need stronger magnetism, such as the metallic QAHE, paving the way to the real world is needed as you suggest.

In addition, we have also: - Corrected the typos found. - Improved some expressions for readability. - Interchanged captions of Fig. 14 and Fig. 15, where conditions were labelled inversely. - Updated arXiv references to published references. - Added Ref. [108][144] [155]. - Deleted discussions about weak TI in Section 2.2 the lattice model. - Moved the original section 6.1.2 to 6.1.3 and added a new 6.1.2 section discussing the lowered threshold of needed magnetic strength in realizing the metallic QAHE, by TI bulk mass decreasing in the middle caused by magnetic doping concentration increase.

---

## Round 1 · Referee Report · Anonymous (Referee 2) · 2024-8-12

Strengths

Comprehensive exposition.
Solid theoretical foundation.
Goes beyond existing models of topological insulators.

Weaknesses

Quite a few typos (also flagged by the other Referee).

Report

The study of topological phases has revolutionised condensed matter physics. At the same time, a complete and general understanding of the topic is lacking in the field. The 3D topological insulator system is one of the most studied systems but almost all studies of it rely on a simplified model that tends to miss important physics. The authors remedy this shortcoming in this work, where they show that high bands of the 3D bulk material contribute to the topological behaviour of the thin film/surface Dirac fermion states. Beyond this, the present paper also presents a complex and mathematically sophisticated analysis of the topological phases available in 3D TI systems in terms of matrix Higgs fields. It emphasises the physics behind weak and strong Zeeman regimes, which are critical for the behaviour of Dirac fermion states.

The paper is a welcome addition to a highly complex field and will be extremely useful to those wishing to learn the topic from scratch thanks to the thorough overview of the discipline inherent in the presentation. I especially appreciated the discussion of the novel material MnBiTe, which is very promising in spintronics and nonlinear applications.

I believe the paper should be published in SciPost, my only recommendation is a thorough revision of the typos and spell check.

Requested changes

  1. Thorough revision of typos.
  2. Spell check.

Recommendation

Publish (meets expectations and criteria for this Journal)

  • validity: top
  • significance: top
  • originality: high
  • clarity: top
  • formatting: perfect
  • grammar: good

Author:  Kai-Zhi Bai  on 2024-08-27  [id 4718]

(in reply to Report 2 on 2024-08-12)

Reply to the report:

We would like to express our sincere thanks for your positive feedback on our manuscript. We have
carefully addressed the typos and spellings, and made the necessary corrections in the revised version of the manuscript.

---

## Editorial Decision

published